# UNDERSTANDING AND IMPROVING HYPERBOLIC DEEP REINFORCEMENT LEARNING

**Timo Klein**[*,1,2], **Thomas Lang**[*,1,2], **Andrii Shkabrii**[1,2], **Alexander Sturm**[1,2], **Kevin Sidak**[1,2]
**Lukas Miklautz**[3], **Claudia Plant**[1,4], **Yllka Velaj**[1,4], **Sebastian Tschiatschek**[1,4]

[1] Faculty of Computer Science, University of Vienna, Vienna, Austria

[2] UniVie Doctoral School Computer Science, University of Vienna, Vienna, Austria

[3] Department of Machine Learning and Systems Biology, Max Planck Institute of Biochemistry, Martinsried, Germany

[4] ds:UniVie, University of Vienna, Vienna, Austria

[*] Joint first authors

`firstname.lastname@univie.ac.at`

## ABSTRACT

The exponential volume growth of hyperbolic geometry can embed the hierarchical relationships between states in reinforcement learning (RL) with far less distortion than Euclidean space. However, hyperbolic deep RL faces severe optimization challenges, and formal analysis of *why* optimization fails is lacking. We identify key factors that determine the success and failure of training hyperbolic deep RL agents. By analyzing the gradients of core operations in the Poincaré Ball and Hyperboloid models of hyperbolic geometry, we show that large-norm embeddings destabilize gradient-based training, leading to trust-region violations in proximal policy optimization (PPO). Based on these insights, we introduce HYPER++, a new hyperbolic deep RL agent that consists of three components: (i) feature regularization guaranteeing bounded norms while avoiding the curse of dimensionality from clipping; (ii) a categorical value loss for stable critic training; and (iii) a more optimization-friendly formulation of hyperbolic network layers. On ProcGen, we show that HYPER++ guarantees stable learning, outperforms prior hyperbolic agents, and reduces wall-clock time by approximately 30%. On Atari-5 with Double DQN, HYPER++ strongly outperforms Euclidean and hyperbolic baselines. **We release our code at https://github.com/Probabilistic-and-Interactive-ML/hyper-rl**.

## 1 INTRODUCTION

Consider a chess agent evaluating its next move: each action branches into exponentially many future states, creating a vast tree of possibilities. This same structure defines common reinforcement learning (RL) benchmarks like ProcGen BIGFISH (Cobbe et al., 2020), where an agent grows by eating smaller fish following an irreversible hierarchy. More generally, sequential decision-making produces inherently hierarchical data: each state branches into multiple potential next states, forming tree-like structures that grow *exponentially* with depth. In contrast, Euclidean volume grows only *polynomially* relative to its radius, resulting in a fundamental geometric mismatch between the exponential branching of decision processes and the polynomial capacity of Euclidean embedding spaces. This forces an agent's representation to severely distort hierarchical relationships, a structural limitation that may contribute to deep

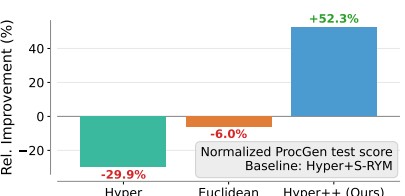

Figure 1: **Baseline improvement on ProcGen.** We compare mean test rewards for our agent (HYPER++), a Euclidean agent, and an unregularized hyperbolic agent (Hyper) with Cetin et al. (2023)'s agent (Hyper+S-RYM).

RL's notorious data inefficiency (Sarkar, 2011; Gromov, 1987).

Hyperbolic geometry offers a natural solution to these limitations. Its exponential volume growth aligns with hierarchical structures, enabling efficient, low-distortion embeddings of trees. While

applications of hyperbolic geometry in deep learning have achieved strong results in classification (Ganea et al., 2018), unsupervised representation learning (Mathieu et al., 2019), deep metric learning (Ermolov et al., 2022), and image-text alignment (Pal et al., 2025), optimization instabilities have limited broader adoption (Guo et al., 2022; Mishne et al., 2023). This is particularly evident in RL, where nonstationarity amplifies gradient instability (Cetin et al., 2023).

We study these optimization failures in proximal policy optimization (PPO) (Schulman et al., 2017) agents using hybrid Euclidean-hyperbolic encoders, a commonly used architecture in deep RL (Cetin et al., 2023; Salemohamed et al., 2023). Through formal gradient analysis, we find that *growing embedding norms destabilize training in both the Poincaré Ball and Hyperboloid models*, causing trust-region violations despite PPO's clipping mechanism. Existing stabilization techniques, such as SpectralNorm, are insufficient as they cannot mitigate gradient pathologies without severely limiting network capacity.

**HYPER++** addresses these failures with three targeted components. RMSNorm (Zhang & Sennrich, 2019) combined with a novel learned scaling layer bounds embedding norms without sacrificing capacity — eliminating SpectralNorm's stability-capacity trade-off. Switching to the Hyperboloid model removes instabilities inherent to the Poincaré ball model at their source, preventing large gradients from propagating via the chain rule. Finally, we replace MSE regression with a categorical value loss, aligning the critic's output with the hyperplane-distance geometry of hyperbolic multinomial logistic regression. This stabilizes critic learning under nonstationary targets. Compared to prior hyperbolic agents, HYPER++ **achieves better performance, is faster, and more general**: It improves test return by 52% (PPO+ProcGen), reduces forward pass time by 30%, and performance gains transfer to Double DQN (Atari-5) and Phasic Policy Gradient (Cobbe et al., 2021) (ProcGen).

---

**Our Key Contributions**

1. **Characterization of training issues.** For both the Poincaré Ball and Hyperboloid, we formally analyze key operations and link them to training instability in deep RL.

2. **Principled regularization.** We study the weaknesses of current approaches and propose improvements rooted in our insights into hyperbolic deep RL training.

3. **HYPER++, a *strong and general* hyperbolic agent.** We combine RMSNorm with a novel scaling layer, the hyperboloid model, and a categorical value loss.

---

## 2 BACKGROUND

This section first reviews Markov decision processes (MDPs) and the PPO optimization procedure (Section 2.1), then presents the mathematical foundations of hyperbolic representation learning in Section 2.2. A more thorough overview of the Poincaré Ball and Hyperboloid models can be found in Ganea et al. (2018); Shimizu et al. (2021); Bdeir et al. (2024).

### 2.1 REINFORCEMENT LEARNING

We formalize RL as a discrete MDP $M = \langle \mathcal{S}, \mathcal{A}, \mathcal{P}, \mathcal{R}, \gamma \rangle$ with state space $\mathcal{S}$ and action space $\mathcal{A}$. At each time step $t$, the agent observes a state $s \in \mathcal{S}$ and selects an action $a \in \mathcal{A}$ with its policy $\pi \colon \mathcal{S} \to [0,1]^{|\mathcal{A}|}$. The environment generates a reward via its reward function $\mathcal{R} \colon \mathcal{S} \times \mathcal{A} \to \mathbb{R}$ and transitions to the next state according to the transition kernel $\mathcal{P} \colon \mathcal{S} \times \mathcal{S} \times \mathcal{A} \to [0,1]$. The agent maximizes discounted future rewards $J(\pi) = \mathbb{E}_\pi \left[ \sum_{t=0}^{\infty} \gamma^t r(s_t, a_t) \mid \pi \right]$, where $\gamma \in [0,1)$ is a discount factor determining how much the agent values future rewards (Sutton & Barto, 2018).

**PPO** Proximal Policy Optimization (PPO) (Schulman et al., 2017) is an actor-critic algorithm directly maximizing cumulative reward via gradient ascent on a surrogate objective. It replaces the hard trust-region constraint of Trust-Region Policy Optimization (TRPO) (Schulman et al., 2015) with the clipped objective

$$J^{\text{CLIP}}(\theta) = \hat{\mathbb{E}}_t \left[ \min(r_t(\theta) A_t \ , \ \text{clamp}(r_t(\theta), 1 - \epsilon, 1 + \epsilon) \, A_t \right], \tag{1}$$

where $r_t(\theta) = \frac{\pi_\theta(a_t|s_t)}{\pi_{\theta_{\text{old}}}(a_t|s_t)}$ are importance sampling ratios of policies parameterized by $\theta$, and $\hat{\mathbb{E}}_t$ is the empirical mean with respect to the samples generated in episode $t$. The $\min$-clamping in Equation 1

truncates the incentive to move probability ratios beyond $[1 - \epsilon, 1 + \epsilon]$, acting as an unconstrained proxy for TRPO's KL-divergence trust region (see Appendix B.1).

## 2.2 Hyperbolic Representation Learning

**Hyperbolic Geometry**

In this work, we employ two common models of hyperbolic space: the *Poincaré Ball* and the *Hyperboloid*. The two isometrically equivalent (distance-preserving) models are $d$-dimensional simply-connected Riemannian submanifolds $(\mathcal{M}, g)$ with constant negative sectional curvature $-c$ (see Figure 4), with $c \in \mathbb{R}_{>0}$.

**Poincaré Ball** The $d$-dimensional *Poincaré Ball* is defined as the Riemannian submanifold $(\mathbb{P}^d_c, g_{\mathbb{P}^d_c})$, with $\mathbb{P}^d_c = \left\{ (x_1, \ldots, x_d) \in \mathbb{R}^d : \|\boldsymbol{x}\|^2 < \frac{1}{c} \right\}$. Its Riemannian metric $g_{\mathbb{P}^d_c}$ is given by the collection of inner products $\langle \boldsymbol{u}, \boldsymbol{v} \rangle_{\boldsymbol{x}} : \mathcal{T}_{\boldsymbol{x}} \mathbb{P}^d_c \times \mathcal{T}_{\boldsymbol{x}} \mathbb{P}^d_c \to \mathbb{R}, (\boldsymbol{u}, \boldsymbol{v}) \mapsto \lambda^c_{\boldsymbol{x}} \langle \boldsymbol{u}, \boldsymbol{v} \rangle$ that smoothly varies between tangent spaces $\mathcal{T}_{\boldsymbol{x}} \mathbb{P}^d_c$ with base points $\boldsymbol{x} \in \mathbb{P}^d_c$. That is, the Poincaré Ball is conformal (angle-preserving) to the Euclidean space with conformal factor $\lambda^c_x = \frac{2}{1 - c\|\boldsymbol{x}\|^2}$.

**Hyperboloid** The $d$-dimensional *Hyperboloid*, often called *Lorentz manifold*, is defined as the forward sheet $(\mathbb{H}^d_c, g_{\mathbb{H}^d_c})$ of a two-sheeted Hyperboloid, where $\mathbb{H}^d_c = \left\{ (x_0, \ldots, x_d) \in \mathbb{R}^{d+1} : \langle \boldsymbol{x}, \boldsymbol{x} \rangle_{\mathcal{L}} = -\frac{1}{c}, x_0 > 0 \right\}$ and $\langle \boldsymbol{x}, \boldsymbol{x} \rangle_{\mathcal{L}} = -x_0^2 + x_1^2 + \cdots + x_d^2$ is the Minkowski inner product. It is endowed with the Riemannian metric $g_{\mathbb{H}^d_c}$ that arises when restricting the Minkowski inner product to the tangent spaces $\mathcal{T}_{\boldsymbol{x}} \mathbb{H}^d_c$, i.e. $\langle \boldsymbol{u}, \boldsymbol{v} \rangle_{\boldsymbol{x}} : \mathcal{T}_{\boldsymbol{x}} \mathbb{H}^d_c \times \mathcal{T}_{\boldsymbol{x}} \mathbb{H}^d_c \to \mathbb{R}, (\boldsymbol{u}, \boldsymbol{v}) \mapsto \langle \boldsymbol{u}, \boldsymbol{v} \rangle_{\mathcal{L}}$. In this work, we frequently refer to the first component $\boldsymbol{x_0}$ of $\boldsymbol{x} \in \mathbb{H}^d_c$ as *time component* and to the other components $\boldsymbol{x_{1:d}}$ as *space component*.

**Hyperbolic Encoding** In our experiments, we retrieve hyperbolic latent representations by first mapping Euclidean vectors $\boldsymbol{v} \in \mathbb{R}^d$ to the tangent space at the manifold's origin $\bar{\mathbf{0}}$, followed by applying the *exponential map* at the origin $\exp_{\bar{\mathbf{0}}}$, to project it onto the manifold $\mathcal{M}$. This process can be summarized as $\mathbb{R}^d \xrightarrow{\phi} \mathcal{T}_{\bar{\mathbf{0}}} \mathcal{M} \xrightarrow{\exp_{\bar{\mathbf{0}}}} \mathcal{M}$. The *exponential map* at the origin $\exp_{\bar{\mathbf{0}}} : \mathcal{T}_{\bar{\mathbf{0}}} \mathcal{M} \to \mathcal{M}$ maps vectors $\boldsymbol{v} \in \mathcal{T}_{\bar{\mathbf{0}}} \mathcal{M}$ to the manifold $\mathcal{M}$ such that the curve $t \in [0, 1] \mapsto \exp_{\bar{\mathbf{0}}}(t\boldsymbol{v})$ is a geodesic (shortest path) joining the manifold's origin $\bar{\mathbf{0}}$ and $\exp_{\bar{\mathbf{0}}}(\boldsymbol{v})$. The specific mapping functions are:

- **Poincaré Ball:** The origin $\bar{\mathbf{0}}$ is the Euclidean origin $\mathbf{0}$, i.e. $\phi$ is the identity function and the exponential map at the origin is $\exp_{\bar{\mathbf{0}}} : \boldsymbol{v} \mapsto \frac{\tanh(\sqrt{c}\|\boldsymbol{v}\|)}{\sqrt{c}\|\boldsymbol{v}\|} \boldsymbol{v}$.

- **Hyperboloid:** The origin is $\bar{\mathbf{0}} = (1/\sqrt{c}, 0, \ldots, 0)$. The map $\phi$ projects a Euclidean vector $\boldsymbol{v} \in \mathbb{R}^d$ onto the tangent space $\mathcal{T}_{\bar{\mathbf{0}}} \mathbb{H}^d_c = \left\{ \boldsymbol{v} \in \mathbb{R}^{d+1} : \langle \boldsymbol{v}, \bar{\mathbf{0}} \rangle_{\mathcal{L}} = 0 \right\}$ by setting its first coordinate to zero, i.e. $\phi : \boldsymbol{v} \mapsto (0, \boldsymbol{v})$. The exponential map at the origin is $\exp_{\bar{\mathbf{0}}} : \boldsymbol{v} \mapsto \cosh\left( \sqrt{c \langle \boldsymbol{v}, \boldsymbol{v} \rangle_{\mathcal{L}}} \right) \bar{\mathbf{0}} + \sinh\left( \sqrt{c \langle \boldsymbol{v}, \boldsymbol{v} \rangle_{\mathcal{L}}} \right) \frac{\boldsymbol{v}}{\sqrt{c \langle \boldsymbol{v}, \boldsymbol{v} \rangle_{\mathcal{L}}}}$.

**Hyperbolic Multinomial Logistic Regression** For the policy and value function of our PPO agent, we compute the Multinomial Logistic Regression (MLR) (Lebanon & Lafferty, 2004; Shimizu et al., 2021; Bdeir et al., 2024) in hyperbolic space. The method computes the probability $p(\boldsymbol{y} = k \mid \boldsymbol{x})$ of an input $\boldsymbol{x} \in \mathcal{M} \simeq \mathbb{R}^d$ belonging to a specific class $k \in \{1, \ldots, K\}$:

$$p(\boldsymbol{y} = k \mid \boldsymbol{x}) \propto \exp(v_{\boldsymbol{z}_k, r_k}(\boldsymbol{x})), \quad v_{\boldsymbol{z}_k, r_k}(\boldsymbol{x}) = \|\boldsymbol{z}_k\|_{\mathcal{T}_{\boldsymbol{p}_k} \mathcal{M}} \, d_{\mathcal{M}}(\boldsymbol{x}, \mathcal{H}_{\boldsymbol{z}_k, r_k}). \tag{2}$$

Here, $\exp(v_{\boldsymbol{z}_k, r_k}(\boldsymbol{x}))$ is the logit for class $k$ and $v_{\boldsymbol{z}_k, r_k}(\boldsymbol{x})$ the signed distance to the margin hyperplane $\mathcal{H}_{\boldsymbol{z}_k, r_k}$ with learnable parameters $\boldsymbol{z}_k \in \mathbb{R}^d$, $r_k \in \mathbb{R}$ specifying the normal and shift vector $\boldsymbol{p_k}$, respectively. The specific definitions for these parameters and the hyperplane itself depend on the hyperbolic model. We expand on this further in Appendix B.3.

## 3 Diagnosing Issues With Hyperbolic PPO Agents

In this section, we analyze training issues of hyperbolic PPO agents (Section 3.1). We link these issues to the gradients of common hybrid neural network architectures as used in Cetin et al. (2023) in Sections 3.2, 3.3, and 3.4. These networks consist of a shared Euclidean encoder with only the

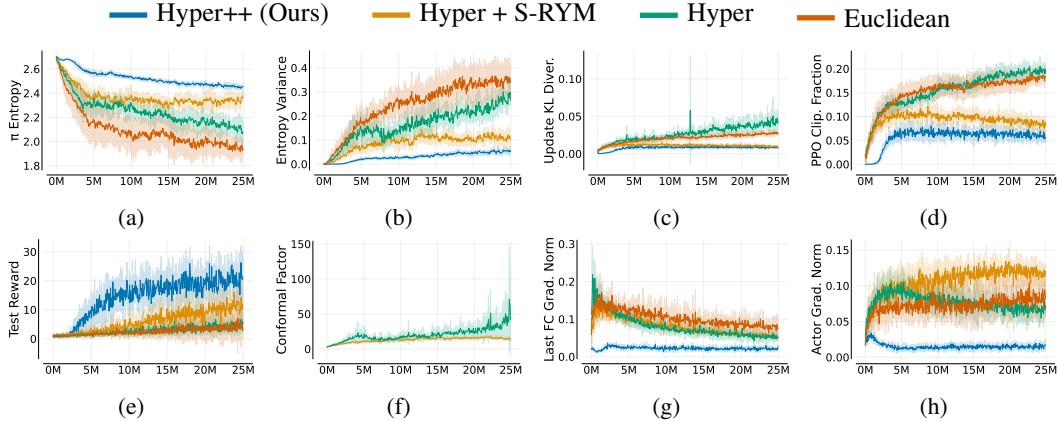

Figure 2: **PPO training metrics.** Unregularized agents (Hyper, Euclidean) lose entropy and show unstable updates (higher update KL and clip fraction), with lower returns and larger gradients (BigFish). Hyper's conformal factor explodes. In contrast, HYPER++ uses the Hyperboloid, which has no conformal factor. Metrics are means over six seeds with one standard deviation.

last layers for the actor and the critic being hyperbolic (cf. Figure 11 in the Appendix). Appendices B.3, B.3.1, and B.3.2 contain additional background on the MLR formulations of the Poincaré Ball and the Hyperboloid.

## 3.1 PPO OPTIMIZATION

PPO's clipped surrogate objective (Eq. 1) restricts the per-sample importance sampling ratios and acts as a heuristic trust region (Schulman et al., 2017). A high clipping fraction indicates many samples are at the trust region boundary. Crucially, PPO constrains ratios only on the sampled states in a batch. Gradient steps leading to large policy changes on unseen states remain unconstrained, so the heuristic trust region can fail. This cross-state interference can produce large unintended policy shifts beyond the sampled states in the batch (Moalla et al., 2024).

Figure 2 shows key training metrics for hyperbolic PPO training in the BIGFISH environment (top row). As noted by Cetin et al. (2023), unregularized hyperbolic PPO is prone to early entropy collapse in Figure 2a. This coincides with a rapid rise in entropy variance across batch states, producing large policy updates that potentially interfere (Figure 2b). Figures 2c and 2d confirm: unregularized agents experience larger update KL-divergence and more trust-region violations. Cetin et al. (2023) propose to mitigate this with S-RYM, a combination of Euclidean embeddings scaled by $1/\sqrt{d}$ and SpectralNorm to bound the encoder's Lipschitz constant (Hyper+S-RYM in Fig. 2). In comparison, our method (Section 4) achieves lower update KL and markedly less clipping while avoiding the overhead of SpectralNorm and instabilities from the conformal factor.

## 3.2 GRADIENT ANALYSIS PRELIMINARIES

To explain the trust-region instability of the hyperbolic agent, we follow Cetin et al. (2023) and analyze the gradients of the last encoder layer (Fig. 2g). Figure 2f shows that the conformal factor of the Poincaré Ball $\lambda_x^c = \frac{2}{1-c\|\boldsymbol{x}\|^2}$ is a key driver for inducing instability. In the following, we derive closed-form, curvature-aware gradients for core hyperbolic layers and maps to study optimization failure points, extending Guo et al. (2022); Mishne et al. (2023) with new expressions for PPO. Below, we present the gradient with respect to the last Euclidean layer weights $\mathbf{W}^{\mathrm{E}}$ for a generic loss $L$.

$$\frac{\partial L}{\partial \boldsymbol{W}^{\mathrm{E}}} = \frac{\partial L}{\partial v_{\boldsymbol{z},r}(\boldsymbol{x}_{\mathrm{H}})} \quad \frac{\partial v_{\boldsymbol{z},r}(\boldsymbol{x}_{\mathrm{H}})}{\partial \boldsymbol{x}_{\mathrm{H}}} \quad \frac{\partial \boldsymbol{x}_{\mathrm{H}}}{\partial \boldsymbol{x}_{\mathrm{E}}} \quad \frac{\partial \boldsymbol{x}_{\mathrm{E}}}{\partial \boldsymbol{W}^{\mathrm{E}}} , \tag{3}$$

where $v_{\boldsymbol{z},r}$ denotes the score function of any hyperbolic multinomial regression (MLR) layer, $\boldsymbol{x}_{\mathrm{E}}$ are the Euclidean embeddings from the encoder, and $\boldsymbol{x}_{\mathrm{H}} = \exp_{\bar{0}}(\boldsymbol{x}_{\mathrm{E}})$ are the embeddings represented as tangent vectors mapped to hyperbolic space. For the Poincaré MLR layer used by Cetin et al. (2023), Guo et al. (2022) have shown that backpropagating through the exponential map yields vanishing

gradients near the boundary because the Riemannian gradient scales with the inverse conformal factor gradient:

$$\nabla_{\boldsymbol{x}_{\mathrm{H}}} \lambda_{\boldsymbol{x}_{\mathrm{H}}}^c = \frac{4c\,\boldsymbol{x}_{\mathrm{H}}}{\left(1 - c\|\boldsymbol{x}_{\mathrm{H}}\|^2\right)^2} \ . \tag{4}$$

### 3.3 Gradient analysis for Hyperbolic Network++ MLR

The derivative (Appendix A.2) of the HNN++ MLR formulation (Shimizu et al., 2021) with respect to its input $\boldsymbol{x}_{\mathrm{H}}$ is:

$$\frac{\partial}{\partial \boldsymbol{x}_{\mathrm{H}}} v_{\boldsymbol{z},r}^{\mathrm{HNN++}}(\boldsymbol{x}_{\mathrm{H}}) = \frac{2\|\boldsymbol{z}\|}{\sqrt{c}} \frac{1}{\sqrt{1 + F(\boldsymbol{x}_{\mathrm{H}})^2}} \frac{\partial}{\partial \boldsymbol{x}_{\mathrm{H}}} F(\boldsymbol{x}_{\mathrm{H}}) \ , \quad \text{where} \tag{5}$$

$$\frac{\partial}{\partial \boldsymbol{x}_{\mathrm{H}}} F(\boldsymbol{x}_{\mathrm{H}}) = \frac{2\sqrt{c}\cosh(2\sqrt{c}r)}{1 - c\|\boldsymbol{x}_{\mathrm{H}}\|^2}\hat{\boldsymbol{z}} + \frac{4\,c\,\boldsymbol{x}_{\mathrm{H}}\left(-\sinh(2\sqrt{c}r) + \sqrt{c}\cosh(2\sqrt{c}r)\langle\hat{\boldsymbol{z}}, \boldsymbol{x}_{\mathrm{H}}\rangle\right)}{(1 - c\|\boldsymbol{x}_{\mathrm{H}}\|^2)^2} \ .$$

where $\hat{\boldsymbol{z}} = \boldsymbol{z}/\|\boldsymbol{z}\|$ is the (normalized) Euclidean weight vector of the layer and $r$ is a scalar bias term. The problematic term is the denominator $(1 - c\|\boldsymbol{x}_{\mathrm{H}}\|^2)^2$ in $\partial F(\boldsymbol{x}_{\mathrm{H}})/\partial\boldsymbol{x}_{\mathrm{H}}$ stemming from Equation 4: it causes gradient explosion near the Poincaré Ball boundary as $\|\boldsymbol{x}_{\mathrm{H}}\| \to 1/\sqrt{c}$. Clipping $\lambda_{\boldsymbol{x}_{\mathrm{H}}}^c$ is undesirable because HNN++ MLR logits depend on $\lambda_{\boldsymbol{x}_{\mathrm{H}}}^c$ and alter the hyperbolic geometry by shifting decision boundaries, leading to performance plateaus. Hence, while HNN++ removes over-parameterization (Shimizu et al., 2021), it does not, by itself, resolve PPO training instabilities.

Next, we analyze the Jacobian of the Poincaré Ball exponential map $\frac{\partial\boldsymbol{x}_{\mathrm{H}}}{\partial\boldsymbol{x}_{\mathrm{E}}}$ similar to Guo et al. (2022) (Appendix A.1):

$$\frac{\partial\boldsymbol{x}_{\mathrm{H}}}{\partial\boldsymbol{x}_{\mathrm{E}}} = \frac{\partial}{\partial\boldsymbol{x}_{\mathrm{E}}}\exp_0(\boldsymbol{x}_{\mathrm{E}}) = \frac{\tanh(\sqrt{c}\|\boldsymbol{x}_{\mathrm{E}}\|)}{\sqrt{c}\|\boldsymbol{x}_{\mathrm{E}}\|}\boldsymbol{I} + \left(\frac{\mathrm{sech}^2(\sqrt{c}\|\boldsymbol{x}_{\mathrm{E}}\|)}{\|\boldsymbol{x}_{\mathrm{E}}\|} - \frac{\tanh(\sqrt{c}\|\boldsymbol{x}_{\mathrm{E}}\|)}{\sqrt{c}\|\boldsymbol{x}_{\mathrm{E}}\|^2}\right)\frac{\boldsymbol{x}_{\mathrm{E}}\boldsymbol{x}_{\mathrm{E}}^\top}{\|\boldsymbol{x}_{\mathrm{E}}\|} .$$

Although the exponential map Jacobian decays like $O(\|\boldsymbol{x}_{\mathrm{E}}\|^{-1})$, the directional term (second summand) is highly sensitive to growing $\|\boldsymbol{x}_{\mathrm{E}}\|$. Figures 2g and 2h show how volatile layer-wise gradients can get during training without proper handling. Cetin et al. (2023)'s S-RYM scaling factor $\boldsymbol{x}_{\mathrm{E}} \mapsto \boldsymbol{x}_{\mathrm{E}}/\sqrt{d}$ keeps $\|\boldsymbol{x}_{\mathrm{E}}\|$ moderate, preventing $\partial\exp_{\bar{0}}(\boldsymbol{x}_{\mathrm{E}})/\partial\boldsymbol{x}_{\mathrm{E}}$ from destabilizing the learning signal fed back to the encoder (Eq. 3) while reducing directional variability. Hence, *regularizing Euclidean embeddings before the hyperbolic layers is a necessity for stable hyperbolic PPO agents*.

### 3.4 Gradient analysis for Hyperboloid MLR

Prior work establishes that the Hyperboloid trains more stably than the Poincaré Ball (Mettes et al., 2024; Mishne et al., 2023; Bdeir et al., 2024) for two reasons. First, the Hyperboloid MLR score (Eq. 26) contains no conformal factor as it is not conformal to Euclidean space. Second, it neither multiplies nor divides by the Euclidean feature norm. As a result, its gradients avoid the instabilities of the Poincaré Ball. However, we will show in the following that the Jacobian $\frac{\partial\boldsymbol{x}_{\mathrm{H}}}{\partial\boldsymbol{v}} = \frac{\partial}{\partial\boldsymbol{v}}\exp_{\bar{0}}^c(\boldsymbol{v})$ of the Hyperboloid's exponential may still destabilize training. We denote $\boldsymbol{v} = [0, \boldsymbol{x}_{\mathrm{E}}] \in \mathcal{T}_{\bar{0}}\mathcal{M}$ as the Euclidean embeddings mapped into the tangent space of the Hyperboloid (cf. Section 2):

$$\frac{\partial\boldsymbol{x}_{\mathrm{H}}}{\partial\boldsymbol{v}} = \begin{bmatrix} 0 & \sinh(\sqrt{c}\|\boldsymbol{x}_{\mathrm{E}}\|)\frac{\boldsymbol{x}_{\mathrm{E}}^\top}{\|\boldsymbol{x}_{\mathrm{E}}\|} \\ \boldsymbol{0} & \frac{\sinh(\sqrt{c}\|\boldsymbol{x}_{\mathrm{E}}\|)}{\sqrt{c}\|\boldsymbol{x}_{\mathrm{E}}\|}\mathbf{I}_d + \frac{\sqrt{c}\|\boldsymbol{x}_{\mathrm{E}}\|\cosh(\sqrt{c}\|\boldsymbol{x}_{\mathrm{E}}\|)-\sinh(\sqrt{c}\|\boldsymbol{x}_{\mathrm{E}}\|)}{\sqrt{c}\|\boldsymbol{x}_{\mathrm{E}}\|^3}\boldsymbol{x}_{\mathrm{E}}\boldsymbol{x}_{\mathrm{E}}^\top \end{bmatrix} . \tag{6}$$

Equation 6 is a $(1+d)\times(1+d)$ matrix, where the first column is zero. For large $\|\boldsymbol{x}_{\mathrm{E}}\|$, $\sinh(\sqrt{c}\|\boldsymbol{x}_{\mathrm{E}}\|)$ and $\cosh(\sqrt{c}\|\boldsymbol{x}_{\mathrm{E}}\|)$ grow exponentially, i.e., a faster rate than $\sqrt{c}\|\boldsymbol{x}_{\mathrm{E}}\|$. *Thus, the Hyperboloid exponential map can destabilize gradients when Euclidean feature norms grow*, requiring regularization of $\|\boldsymbol{x}_{\mathrm{E}}\|$.

Summarizing the findings in this section, we arrive at a more nuanced understanding of the training issues of hyperbolic deep RL agents: Policy breakdown and large-norm gradients in the encoder are a function of the hyperbolic layers used in the actor and the critic. The conformal factor, in particular, is a source of numerical instability in Riemannian optimization methods (Guo et al., 2022; Mishne et al., 2023). This numerical instability gets exacerbated by noisy gradients in actor-critic training, particularly from the critic's side (Sutton & Barto, 2018; Nauman et al., 2024a). In the next section, we will show how our method HYPER++ deals with these issues.

## 4 STABILIZING HYPERBOLIC DEEP RL

In this part, we establish the components of our agent HYPER++: Section 4.1 proposes RM-SNorm (Zhang & Sennrich, 2019) as an alternative to SpectralNorm. Section 4.2 introduces a novel feature scaling layer. Section 4.3 discusses how these components relate to the Hyperboloid. Beyond these design choices, we use a categorical loss to stabilize critic gradients (Imani & White, 2018; Farebrother et al., 2024) and to resolve an architectural mismatch in hyperbolic value learning. While Euclidean linear layers naturally support MSE regression over continuous values, hyperbolic MLR layers output classification-oriented hyperplane distances, making the categorical loss over discrete bins a better geometrical fit. **Collectively, our components target complementary sources of instability in Equation 3**: the categorical loss stabilizes the loss derivative (first term), Hyperboloid MLR stabilizes the hyperbolic layer Jacobian (second term), and RMSNorm with feature scaling stabilizes the Jacobian of the exponential map (third term). Figure 11 illustrates the underlying hybrid network architecture (Guo et al., 2022; Cetin et al., 2023) analyzed in the following.

### 4.1 REGULARIZATION

Here, we study how SpectralNorm (Miyato et al., 2018) affects the Euclidean embeddings produced by the encoder (cf. Figure 11). To this end, consider Lemma 4.1 which provides a bound on the norm of the embeddings computed by a single layer, depending on the input norm:

**Lemma 4.1.** *Let $\boldsymbol{x} \in \mathbb{R}^n$, $\boldsymbol{W} \in \mathbb{R}^{d \times n}$ and $\boldsymbol{b} \in \mathbb{R}^d$. Then, for any function $f : \mathbb{R}^d \to \mathbb{R}^d$ with Lipschitz constant L, it holds that*

$$\|f(\boldsymbol{W}\boldsymbol{x} + \boldsymbol{b})\|_2 \le \|f(\boldsymbol{0})\|_2 + L\|\boldsymbol{W}\|_2\|\boldsymbol{x}\|_2 + L\|\boldsymbol{b}\|_2 . \tag{7}$$

*In particular, for ReLU activation functions and any normalized weight matrix $\hat{\boldsymbol{W}}$, we have*

$$\left\|\text{ReLU}(\hat{\boldsymbol{W}}\boldsymbol{x} + \boldsymbol{b})\right\|_2 \le \|\boldsymbol{x}\|_2 + \|\boldsymbol{b}\|_2 . \tag{8}$$

Lemma 4.1 shows that for multi-layer encoders such as the one used by Cetin et al. (2023), applying SpectralNorm only to the last (linear) layer of the encoder is not sufficient to prevent the Euclidean embedding norms from growing via the preceding layers. To tangibly affect these norms, SpectralNorm must be applied to *every* layer of the encoder (Cetin et al., 2023). This constrains the Lipschitz constant of all layers and reduces expressivity by globally enforcing smoothness (Rosca et al., 2020; Cetin et al., 2023). Additionally, SpectralNorm incurs computational overhead from the power-iteration steps needed at each forward pass.

Ideally, we want to use regularization via spectral normalization only where needed and such that we can guarantee stable training, without limiting the expressivity of the entire Euclidean encoder. Proposition 4.2 shows that applying RMSNorm (Zhang & Sennrich, 2019) before the activation of the encoder's last linear layer achieves stability without overly restricting its representational capacity (if the other layers are not regularized, their expressivity is not limited).

**Proposition 4.2.** *Let $\boldsymbol{x} \in \mathbb{R}^d$ and $f : \mathbb{R}^d \to \mathbb{R}^d$ with Lipschitz constant L. Then, for $\hat{\boldsymbol{x}} = \frac{1}{\sqrt{d}}f(\text{RMS}(\boldsymbol{x}))$, it holds that:*

$$\|\hat{\boldsymbol{x}}\|_2 < \frac{1}{\sqrt{d}}\|f(\boldsymbol{0})\|_2 + L, \qquad \lambda_{\exp_{\bar{0}}(\hat{\boldsymbol{x}})} < 2\cosh^2\left(\sqrt{c}\left(\frac{1}{\sqrt{d}}\|f(\boldsymbol{0})\|_2 + L\right)\right). \tag{9}$$

Proposition 4.2 ensures stable hyperbolic operations for a broad class of activation functions. For common 1-Lipschitz activations such as TanH and ReLU, the bounds reduce to $\|\hat{\boldsymbol{x}}\|_2 < 1$ and $\|\exp_{\bar{0}}(\hat{\boldsymbol{x}})\| < \frac{1}{\sqrt{c}}\tanh(\sqrt{c})$. Unlike SpectralNorm, which constrains every encoder layer, we only require applying RMSNorm to the pre-activation output embeddings of the final linear layer. This retains the expressivity of each encoder layer. We use RMSNorm (Zhang & Sennrich, 2019) rather than LayerNorm (Ba et al., 2016) because we do not want the mean-centering in LayerNorm to distort the hierarchical structure of the hyperbolic embeddings. Additionally, RMSNorm brings three further advantages: it smoothes gradients, prevents dead ReLU or saturated TanH units (Zhang & Sennrich, 2019; Xu et al., 2019; Lyle et al., 2024), and supports arbitrary embedding dimensions $d$, since the bound in Proposition 4.2 is dimension-independent for activation functions with fixed point 0.

## 4.2 LEARNED EUCLIDEAN FEATURE SCALING

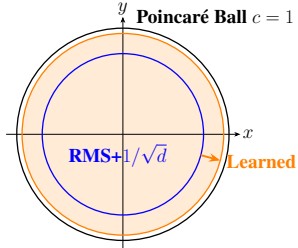

Proposition 4.2 guarantees stability by bounding both Euclidean embedding norms and the conformal factor. However, this may still affect representational capacity in the hyperbolic layers of the agent. For example, with ReLU as the last encoder layer's activation function and curvature $c = 1$, the bound restricts the Poincaré Ball radius to $\|\boldsymbol{x}_{\mathrm{H}}\|_2 \leq 0.76$ (see the proof of Proposition 4.2). Since the volume of a $d$-ball scales as $V_d(r) = \frac{\pi^{d/2}}{\Gamma(\frac{d}{2}+1)} r^d \propto r^d$,

Figure 3: **Learned scaling effect.**

even a modest restriction of the radius causes an exponential loss of available volume in $d$. To mitigate this, we rescale the Euclidean tangent embeddings obtained after application of Proposition 4.2 $\hat{\boldsymbol{x}}_E$ by a learnable scalar $\xi_\theta$:

$$\hat{\boldsymbol{x}}_E^{\mathrm{rescale}} = \rho_{\max}\,\sigma(\xi_\theta)\,\hat{\boldsymbol{x}}_E, \qquad \rho_{\max} = \frac{\mathrm{atanh}(\alpha)}{\sqrt{c}}\,, \tag{10}$$

where $\sigma$ denotes the sigmoid function. By choosing this particular form for $\rho_{\max}$, we have that $\|\exp_{\bar{0}}(\hat{\boldsymbol{x}}_E^{\mathrm{rescale}})\|_2 \leq \alpha/\sqrt{c}$ since $\tanh(\sqrt{c}\rho_{\max}) = \alpha$. Setting $\alpha = 0.95$ (and $c = 1$) expands the usable ball radius from 0.76 to 0.95, i.e., a volume gain of $(0.95/0.76)^d$. For $d = 32$, this is approximately $1.2 \times 10^3$ more volume while still preventing the explosion of the conformal factor according to Proposition 4.2. Figure 3 illustrates the effect in 2D.

## 4.3 HYPERBOLOID MODEL

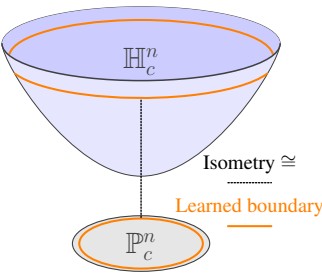

Figure 4: **Isometry between Poincaré Ball and Hyperboloid.**

Section 3.4 shows that the Hyperboloid avoids conformal factor instabilities and is therefore more robust against large norms. Yet, operations can become ill-conditioned far from the origin, i.e., when the sheet approaches the asymptotic null cone, and the Jacobian of the exponential map in Equation 6 gets more sensitive to large Euclidean norms. Since the Poincaré Ball and the Hyperboloid are isometric (Fig. 4) models, our stabilization strategy transfers: instead of capping the Poincaré Ball radius, we propose to apply RMSNorm and feature scaling before the last Euclidean activation to bound the Hyperboloid through its time component $x_0$. Corollary 4.3 formalizes this insight by combining Proposition 4.2 with the Poincaré Ball-Hyperboloid isometry (Chami et al., 2021; Mishne et al., 2023).

**Corollary 4.3.** *Let $\hat{\boldsymbol{x}}_E \in \mathbb{R}^n$ be a point regularized by RMSNorm with learnable scaling, and $\boldsymbol{x}_{\mathrm{H}} = \exp_{\bar{0}}(\hat{\boldsymbol{x}}_E) \in \mathbb{P}^n$. Then, the maximum value of the time component $x_0$ of that point on the Hyperboloid is*

$$x_0^{\max} = \frac{1 + c\|\boldsymbol{x}_{\mathrm{H}}\|^2}{\sqrt{c}\,(1 - c\|\boldsymbol{x}_{\mathrm{H}}\|^2)} = \frac{1 + \tanh^2(\sqrt{c}\|\hat{\boldsymbol{x}}_E\|)}{\sqrt{c}\,(1 - \tanh^2(\sqrt{c}\|\hat{\boldsymbol{x}}_E\|))}\,.$$

Since the time and space components are dependent (cf. Section 2.2), bounding the maximum norm of $x_0^{\max}$ also ensures that the space component $x_s$ remains bounded. Therefore, we also apply regularization with RMSNorm and learned scaling when training agents using the Hyperboloid. In Section 5.2, we show that this approach works well empirically. Our proposed HYPER++ architecture is visualized in Figure 11 (Appendix) and consists of the following components:

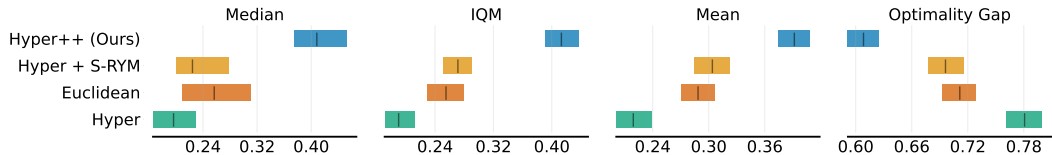

Figure 5: **Normalized test rewards on ProcGen for PPO.** HYPER++ outperforms baselines for all aggregation methods without increasing variance (as measured by the bootstrap confidence interval). We report median, interquartile mean (IQM), mean, and optimality gap, which is $1 - \text{IQM}$.

---

**HYPER++**

HYPER++ tackles optimization issues in hyperbolic deep RL through **formal and empirical analysis of training dynamics**:

1. RL nonstationarity $\implies$ **Categorical value function**.
2. Growing Euclidean feature norms $\implies$ **RMSNorm + Feature scaling**.
3. Conformal factor instability $\implies$ **Hyperboloid model**.

---

## 5 EXPERIMENTS

We evaluate HYPER++ on ProcGen (Cobbe et al., 2020) with PPO (Schulman et al., 2017) and PPG (Cobbe et al., 2021) in Section 5.1. Section 5.2 provides ablation studies for PPO. We test performance with the off-policy algorithm DDQN (van Hasselt et al., 2016) on a subset of Atari games (Bellemare et al., 2015; Towers et al., 2024; Aitchison et al., 2023). Unless stated otherwise, error bands show one standard deviation. Wall-clock times are reported in Appendix D.1.

### 5.1 PROCGEN

Figure 5 shows normalized aggregate test rewards for 25M time steps on all 16 ProcGen environments with PPO. We normalize using random performance as the minimum and either a theoretical or empirically determined maximum (Cobbe et al., 2020). We use the rliable library (Agarwal et al., 2021) to compute aggregate metrics such as the interquartile-mean (IQM) and the optimality gap with bootstrap confidence intervals.

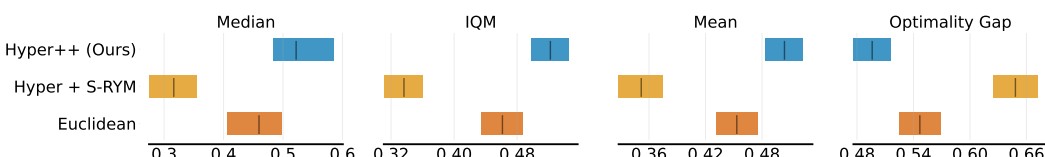

Figure 6: **Normalized test rewards on ProcGen for PPG.** HYPER++ outperforms both Euclidean and existing hyperbolic agents. Hyper+S-RYM is substantially worse than the Euclidean baseline, performing even worse compared to HYPER++ with PPO.

With PPO, HYPER++ outperforms Poincaré agents with and without S-RYM, as well as the Euclidean baseline. Tables 7 and 8 show HYPER++ winning head-to-head vs. Hyper+S-RYM in $8/16$ games on the train set and $11/16$ games on the test set. Training curves for all runs are in Appendix E.1. We further test the performance of HYPER++ by evaluating it with a more recent baseline, PPG (Cobbe et al., 2021). Figure 6 shows the results: Our method outperforms a strong Euclidean baseline in all metrics, whereas the Hyper+S-RYM agent is substantially worse than Euclidean. Notably, HYPER++ with PPO achieves a higher test IQM than Hyper+S-RYM with PPG. We provide full PPG results in Appendix E.8. **In summary, our ProcGen experiments with PPO and PPG highlight the strong performance and generality of HYPER++ compared to previous hyperbolic approaches**.

## 5.2 ABLATION STUDIES

Figure 7 presents ablations of HYPER++'s components using test IQM with bootstrapped confidence intervals. We begin with the most critical component: normalization. Removing RMSNorm (Zhang & Sennrich, 2019) and $1/\sqrt{d}$ feature scaling causes complete learning failure (-RMSNorm), confirming the predictions of Proposition 4.2.

This failure manifests as large embedding norms and near-zero gradients in the encoder's final layer (Figure 14), providing empirical support for the theoretical analysis in Section 3 and Proposition 4.2. The next most important architectural choice is learned scaling (-Scaling), which we attribute to its synergy with RMSNorm. Among the loss function variants, replacing the categorical HL-Gauss loss (Imani & White, 2018) with MSE (+MSE) degrades performance, though not uniformly across all games. This aligns with the findings of Farebrother et al. (2024), who similarly observe that HL-Gauss does not consistently improve performance on all environments. Interestingly, substituting the C51 (Bellemare et al., 2017) distributional loss for HL-Gauss performs even worse than MSE. Using the Poincaré ball instead of the hyperboloid model (+Poincaré) leads to a modest drop in performance, which is expected given their isometry.

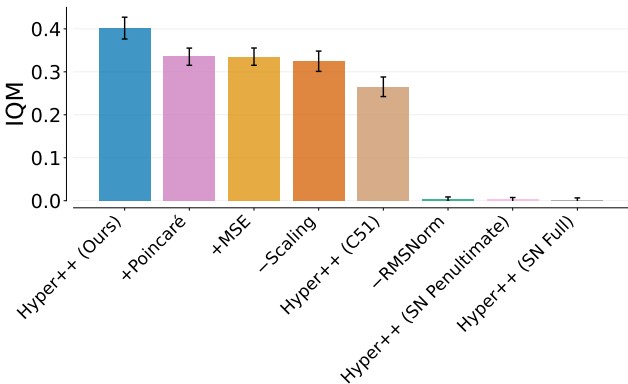

Figure 7: **Ablation studies on ProcGen with Hyperbolic geometry.** We report the test interquartile mean (IQM) across six seeds with bootstrap confidence intervals. − indicates that a component is removed from HYPER++, + indicates a component replacing its analog.

We further validate Lemma 4.1 by testing SpectralNorm (Miyato et al., 2018) as an alternative to RMSNorm in two configurations: applying SpectralNorm to the complete Euclidean encoder (HYPER++ (SN Full)) and applying it only to the penultimate layer (HYPER++ (SN Penultimate)). In both cases, the agent fails to learn entirely. This underscores the critical importance of RMSNorm for obtaining the bounded feature norms guaranteed by Proposition 4.2.

Finally, Figure 8 isolates the contribution of hyperbolic representations by evaluating Euclidean agents equipped with HL-Gauss, RMSNorm, and our full regularization combination. For Euclidean representations, the HL-Gauss loss (Euclidean+Categorical) performs worse than MSE. Adding RMSNorm to Euclidean agents improves performance, and equipping Euclidean agents with our full method yields an IQM of 0.35, which is slightly better than HYPER++ with the Poincaré ball (IQM=0.34). However, HYPER++ with the Hyperboloid achieves the best overall performance (IQM=0.40). The underperformance of Euclidean+HL-Gauss relative to Euclidean+MSE indicates that categorical losses are particularly well-suited for hyperbolic agents due to the geometric alignment between the loss and the critic's architecture. Overall,

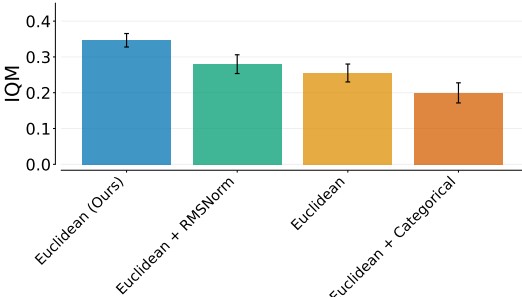

Figure 8: **Ablation studies on ProcGen with Euclidean geometry.** We report the test interquartile mean (IQM) across six seeds with bootstrap confidence intervals.

our results demonstrate that hyperbolic representations can benefit deep RL agents, but require an optimization-friendly approach to hyperbolic geometry to realize these benefits. We present complete ablation results in Tables 9 and 10. **In summary, every ablation underperforms HYPER++ with the Hyperboloid, confirming the synergistic interactions between hyperbolic geometry and our method's components**.

Figure 9: **Human-normalized performance for DDQN on Atari-5.** All agents are trained for $10M$ steps and five seeds. HYPER++ strongly improves over the baselines.

## 5.3 ATARI DDQN

We evaluate our algorithm using the value-based off-policy algorithm DDQN (van Hasselt et al., 2016) (Appendix B.2). We focus on the Atari-5 subset of games (Aitchison et al., 2023), which consists of NAMETHISGAME, PHOENIX, BATTLEZONE, DOUBLE DUNK, and Q\*BERT. This subset has been shown to be the most predictive of overall performance across all Atari environments (Bellemare et al., 2015; Towers et al., 2024). We train each agent for 10M steps and five random seeds. Figure 9 shows the final episode rewards achieved by each method. **HYPER++ substantially outperforms the baselines across all five games in all metrics**. Appendix E.4 provides the full learning curves for each individual game. We find that performance varies across games: our method achieves its strongest gains on NAMETHISGAME and Q\*BERT. On PHOENIX, HYPER++ exhibits strong initial performance but subsequently plateaus, mirroring the behavior of the baseline agents. This plateauing is consistent with plasticity loss being a confounding factor on this particular game (Klein et al., 2024). We ablate the choice of Polyak vs. hard target updates on NAMETHISGAME, the most representative Atari-5 game (Aitchison et al., 2023). Figure 17 (Appendix) shows only minor performance differences, suggesting robustness to this design choice.

## 6 RELATED WORK

**Hyperbolic deep learning** has progressed quickly from early hyperbolic neural networks and Riemannian optimization (Ganea et al., 2018; Bécigneul & Ganea, 2019), which Cetin et al. (2023) adapt to RL. Parameter redundancy in Poincaré Ball MLR was reduced by Shimizu et al. (2021). Fully hyperbolic architectures on the Hyperboloid now include transformers and convolutional networks, as well as a Hyperboloid MLR layer (Chen et al., 2022; Bdeir et al., 2024). Mettes et al. (2024) survey this literature from a vision perspective. Optimization and numerical stability have been analyzed independently (Mishne et al., 2023; Guo et al., 2022). We study optimization problems of hyperbolic networks within RL and propose a principled solution.

**Reinforcement learning.** We focus on PPO (Schulman et al., 2017), which remains under active study (Andrychowicz et al., 2021; Moalla et al., 2024) because of its strong performance. Several works show that regularization can improve deep RL training; with LayerNorm being widely adopted in the deep RL (Henderson et al., 2018; Ba et al., 2016; Lyle et al., 2023; Nauman et al., 2024b; Lee et al., 2025; Gallici et al., 2025). Instead, we regularize our agent using RMSNorm (Zhang & Sennrich, 2019), preventing interference with hyperbolic representations. A separate line of research is stabilizing value function learning via categorical objectives (Bellemare et al., 2017; Schrittwieser et al., 2020; Imani & White, 2018; Farebrother et al., 2024), which we extend to hyperbolic PPO.

## 7 LIMITATIONS AND CONCLUSION

**Limitations and Future Work.** Our analysis takes an optimization-centric view, focusing on training dynamics and the question of *how* hyperbolic deep RL learns, rather than *what* structures their representations capture. We also do not address which environments are most suited to hyperbolic representations. Moreover, the interaction between geometric choices and the design of different deep RL algorithms, remains unexplored. Each of these directions is an exciting avenue for future work.

**Conclusion.** Our work analyzes gradients in the Poincaré Ball and Hyperboloid, linking large-norm embeddings to PPO trust-region breakdowns. Based on these insights, we introduce HYPER++, which combines RMSNorm with learned feature scaling and a categorical value loss to stabilize hyperbolic deep RL. On ProcGen, HYPER++ improves performance and substantially reduces wall-clock time compared to existing hyperbolic PPO agents. Our findings transfer to Atari and DDQN with strong gains, indicating broader applicability beyond PPO.

REPRODUCIBILITY

Our code is publicly available at https://github.com/Probabilistic-and-Interactive-ML/hyper-rl. Appendix A contains the derivations for our gradient analysis and proofs for our theoretical results. In Appendix D, we state agent architecture, hyperparameters, relevant implementation details, and hardware used. In Appendix C.3 we discuss differences in results to existing works.

USAGE OF LARGE LANGUAGE MODELS (LLMS)

During this project, we used LLMs as an assistive tool. In the early stages of our project, we used LLMs for literature search and paper summarization. During the implementation phase, we used code assistants to support repetitive coding tasks such as Matplotlib figure generation. For the paper, LLMs were used as a tool to iterate on our writing. An example use case is paragraph shortening with "Shorten this paragraph." All LLM outputs used in this paper were thoroughly reviewed to ensure accuracy. LLMs were not used for idea generation, experimental design, or for proofs. Mathematical expressions were derived independently.

ETHICS STATEMENT

Our work advances the fundamental capabilities of hyperbolic deep RL agents and has no direct ethical implications by itself. We cannot rule out that unethical uses could occur in downstream applications because RL and PPO, in particular, are used to train LLMs, and hyperbolic embeddings are well-suited for text data. However, such uses would require significant extensions and modifications beyond the work submitted here.

ACKNOWLEDGMENTS

This work has been funded in parts by the Vienna Science and Technology Fund (WWTF) [10.47379/ICT20058]. We acknowledge EuroHPC JU for awarding the project ID EHPC-DEV-2025D08-024 access to the Luxembourg national supercomputer, MeluXina, and the Spanish supercomputer, MareNostrum. The authors gratefully acknowledge the LuxProvide and Barcelona Supercomputing Center for their expert support. Without Nikolaus Süß' tireless work maintaining our research group's computing infrastructure, this work would not have been possible. We are deeply grateful. Lastly, we want to thank the open source community, particularly the developers of PyTorch (Paszke et al., 2019) and Matplotlib (Hunter, 2007).

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

# Appendix

Table 1 summarizes the contents of our Appendix:

Table 1: Structure of our appendix.

| Appendix Section | Content |
|---|---|
| Appendix A | Proofs & Derivations |
| Appendix B | Additional Background for RL and hyperbolic MLR Layers |
| Appendix C | Environment descriptions |
| Appendix D | Compute details and hyperparameters |
| Appendix E | Complete results and additional metrics |

## A DERIVATIONS AND PROOFS

This section contains derivations and proofs for the results in

### A.1 POINCARÉ EXPONENTIAL MAP GRADIENTS

We build on the analysis by Guo et al. (2022) and derive the Jacobian of the Poincaré Ball exponential map at the origin.

$$
\begin{aligned}
D\boldsymbol{x}_{\mathrm{H}} &= \frac{\partial}{\partial \boldsymbol{x}_{\mathrm{E}}} \exp_0(\boldsymbol{x}_{\mathrm{E}}) \\
&= \frac{\partial}{\partial \boldsymbol{x}_{\mathrm{E}}} \frac{\tanh(\sqrt{c}\|\boldsymbol{x}_{\mathrm{E}}\|)}{\sqrt{c}\|\boldsymbol{x}_{\mathrm{E}}\|} \boldsymbol{x}_{\mathrm{E}} \\
&= \frac{\tanh(\sqrt{c}\|\boldsymbol{x}_{\mathrm{E}}\|)}{\sqrt{c}\|\boldsymbol{x}_{\mathrm{E}}\|} \boldsymbol{I}_d + \left( \frac{\partial}{\partial \boldsymbol{x}_{\mathrm{E}}} \frac{\tanh(\sqrt{c}\|\boldsymbol{x}_{\mathrm{E}}\|)}{\sqrt{c}\|\boldsymbol{x}_{\mathrm{E}}\|} \right) \boldsymbol{x}_{\mathrm{E}} \,,
\end{aligned}
\tag{11}
$$

where $\boldsymbol{I}_d$ denotes the $d \times d$ identity matrix.

Deriving the second term yields:

$$
\begin{aligned}
\frac{\partial}{\partial \boldsymbol{x}_{\mathrm{E}}} \frac{\tanh(\sqrt{c}\|\boldsymbol{x}_{\mathrm{E}}\|)}{\sqrt{c}\|\boldsymbol{x}_{\mathrm{E}}\|} &= \frac{\frac{\partial}{\partial \boldsymbol{x}_{\mathrm{E}}} \tanh(\sqrt{c}\|\boldsymbol{x}_{\mathrm{E}}\|) \cdot \sqrt{c}\|\boldsymbol{x}_{\mathrm{E}}\| - \tanh(\sqrt{c}\|\boldsymbol{x}_{\mathrm{E}}\|) \cdot \frac{\partial}{\partial \boldsymbol{x}_{\mathrm{E}}} \sqrt{c}\|\boldsymbol{x}_{\mathrm{E}}\|}{(\sqrt{c}\|\boldsymbol{x}_{\mathrm{E}}\|)^2} \\
&\overset{(i)}{=} \frac{\operatorname{sech}^2(\sqrt{c}\|\boldsymbol{x}_{\mathrm{E}}\|)\sqrt{c}\hat{\boldsymbol{x}}_E \cdot \sqrt{c}\|\boldsymbol{x}_{\mathrm{E}}\| - \tanh(\sqrt{c}\|\boldsymbol{x}_{\mathrm{E}}\|) \cdot \sqrt{c}\hat{\boldsymbol{x}}_E}{c\|\boldsymbol{x}_{\mathrm{E}}\|^2} \\
&= \frac{\operatorname{sech}^2(\sqrt{c}\|\boldsymbol{x}_{\mathrm{E}}\|)\, c\boldsymbol{x}_{\mathrm{E}} - \tanh(\sqrt{c}\|\boldsymbol{x}_{\mathrm{E}}\|) \cdot \sqrt{c}\hat{\boldsymbol{x}}_E}{c\|\boldsymbol{x}_{\mathrm{E}}\|^2} \\
&= \frac{\operatorname{sech}^2(\sqrt{c}\|\boldsymbol{x}_{\mathrm{E}}\|)\, \boldsymbol{x}_{\mathrm{E}}}{\|\boldsymbol{x}_{\mathrm{E}}\|^2} - \frac{\tanh(\sqrt{c}\|\boldsymbol{x}_{\mathrm{E}}\|)\, \boldsymbol{x}_{\mathrm{E}}}{\sqrt{c}\|\boldsymbol{x}_{\mathrm{E}}\|^3} \\
&= \left( \frac{\operatorname{sech}^2(\sqrt{c}\|\boldsymbol{x}_{\mathrm{E}}\|)}{\|\boldsymbol{x}_{\mathrm{E}}\|^2} - \frac{\tanh(\sqrt{c}\|\boldsymbol{x}_{\mathrm{E}}\|)}{\sqrt{c}\|\boldsymbol{x}_{\mathrm{E}}\|^3} \right) \boldsymbol{x}_{\mathrm{E}} \,,
\end{aligned}
\tag{12}
$$

where we use $\hat{\boldsymbol{x}}_E = \frac{\partial}{\partial \boldsymbol{x}_{\mathrm{E}}} \|\boldsymbol{x}_{\mathrm{E}}\| = \frac{\boldsymbol{x}_{\mathrm{E}}}{\|\boldsymbol{x}_{\mathrm{E}}\|}$ in $(i)$.

Putting Equation 11 and 12 together yields:

$$
\frac{\partial}{\partial \boldsymbol{x}_{\mathrm{E}}} \exp_0(\boldsymbol{x}_{\mathrm{E}}) = \frac{\tanh(\sqrt{c}\|\boldsymbol{x}_{\mathrm{E}}\|)}{\sqrt{c}\|\boldsymbol{x}_{\mathrm{E}}\|} \boldsymbol{I} + \left( \frac{\operatorname{sech}^2(\sqrt{c}\|\boldsymbol{x}_{\mathrm{E}}\|)}{\|\boldsymbol{x}_{\mathrm{E}}\|} - \frac{\tanh(\sqrt{c}\|\boldsymbol{x}_{\mathrm{E}}\|)}{\sqrt{c}\|\boldsymbol{x}_{\mathrm{E}}\|^2} \right) \frac{\boldsymbol{x}_{\mathrm{E}}\boldsymbol{x}_{\mathrm{E}}^\top}{\|\boldsymbol{x}_{\mathrm{E}}\|}.
\tag{13}
$$

We can see that the Jacobian of the exponential map decays with $O\big(\|\boldsymbol{x}_{\mathrm{E}}\|^{-1}\big)$, although the important directional term (second summand) can vanish faster with $O\big(\|\boldsymbol{x}_{\mathrm{E}}\|^{-2}\big)$.

### A.2 HYPERBOLIC NETWORKS++ GRADIENTS

Let us first re-state the forward pass for the hyperbolic networks++ formulation (Shimizu et al., 2021) of the Poincaré multinomial logistic regression (MLR) layer:

$$
v_{\boldsymbol{z},r}^{\mathrm{HNN++}}(\boldsymbol{x}_{\mathrm{H}}) = \frac{2\,\|\boldsymbol{z}\|}{\sqrt{c}} \sinh^{-1}\left( (1 - \lambda_{\boldsymbol{x}_{\mathrm{H}}}^c) \sinh(2\sqrt{c}\,r) + \sqrt{c}\,\lambda_{\boldsymbol{x}_{\mathrm{H}}}^c \cosh(2\sqrt{c}\,r)\, \langle \hat{\boldsymbol{z}}, \boldsymbol{x}_{\mathrm{H}} \rangle \right) .
\tag{14}
$$

Here $\boldsymbol{x}_{\mathrm{H}} = \exp_{\bar{0}}(\boldsymbol{x}_{\mathrm{E}})$, $\lambda_{\boldsymbol{x}_{\mathrm{H}}}^c = \frac{2}{1 - c\|\boldsymbol{x}_{\mathrm{H}}\|^2}$ is the conformal factor of the Poincaré Ball, $\boldsymbol{z}$ is the weight vector of the layer, $\hat{\boldsymbol{z}} = \frac{\boldsymbol{z}}{\|\boldsymbol{z}\|}$ are the weights normalized to unit length, and $r$ a scalar bias term. We omit the class index $k$ as in the main paper to simplify the notation.

We can re-state Equation 14 as

$$
v_{\boldsymbol{z},r}^{\mathrm{HNN++}}(\boldsymbol{x}_{\mathrm{H}}) = \frac{2\|\boldsymbol{z}\|}{\sqrt{c}} \sinh^{-1}\big( F(\boldsymbol{x}_{\mathrm{H}}) \big) \,,
\tag{15}
$$

with

$$F(\boldsymbol{x}_{\mathrm{H}}) = (1 - \lambda^c_{\boldsymbol{x}_{\mathrm{H}}}) \sinh(2\sqrt{c}\,r) + \sqrt{c}\,\lambda^c_{\boldsymbol{x}_{\mathrm{H}}} \cosh(2\sqrt{c}\,r)\,\langle \hat{\boldsymbol{z}}, \boldsymbol{x}_{\mathrm{H}}\rangle \ . \tag{16}$$

We first calculate the outer derivative (Equation 15):

$$\nabla_{\boldsymbol{x}_{\mathrm{H}}} v^{\mathrm{HNN++}}_{\boldsymbol{z},r}(\boldsymbol{x}_{\mathrm{H}}) = \frac{2\|\boldsymbol{z}\|}{\sqrt{c}}\frac{1}{\sqrt{1 + F(\boldsymbol{x}_{\mathrm{H}})^2}}\nabla_{\boldsymbol{x}_{\mathrm{H}}} F(\boldsymbol{x}_{\mathrm{H}}) \ . \tag{17}$$

The gradient of Equation 16 $\nabla_{\boldsymbol{x}_{\mathrm{H}}} F(\boldsymbol{x}_{\mathrm{H}})$ is:

$$\begin{aligned}
\nabla_{\boldsymbol{x}_{\mathrm{H}}} F(\boldsymbol{x}_{\mathrm{H}}) &= -\sinh(2\sqrt{c}r)\nabla_{\boldsymbol{x}_{\mathrm{H}}}\lambda^c_{\boldsymbol{x}_{\mathrm{H}}} + \nabla_{\boldsymbol{x}_{\mathrm{H}}}\Big[\sqrt{c}\lambda^c_{\boldsymbol{x}_{\mathrm{H}}}\cosh(2\sqrt{c}r)\langle\hat{\boldsymbol{z}},\boldsymbol{x}_{\mathrm{H}}\rangle\Big] \\
&= \sqrt{c}\lambda^c_{\boldsymbol{x}_{\mathrm{H}}}\cosh(2\sqrt{c}r)\hat{\boldsymbol{z}} + \Big(-\sinh(2\sqrt{c}r) + \sqrt{c}\cosh(2\sqrt{c}r)\langle\hat{\boldsymbol{z}},\boldsymbol{x}_{\mathrm{H}}\rangle\Big)\nabla_{\boldsymbol{x}_{\mathrm{H}}}\lambda^c_{\boldsymbol{x}_{\mathrm{H}}} \\
&= \frac{2\sqrt{c}\cosh(2\sqrt{c}r)}{1 - c\|\boldsymbol{x}_{\mathrm{H}}\|^2}\hat{\boldsymbol{z}} + \frac{4c\,\boldsymbol{x}_{\mathrm{H}}\Big(-\sinh(2\sqrt{c}r) + \sqrt{c}\cosh(2\sqrt{c}r)\langle\hat{\boldsymbol{z}},\boldsymbol{x}_{\mathrm{H}}\rangle\Big)}{(1 - c\|\boldsymbol{x}_{\mathrm{H}}\|^2)^2} \ . \tag{18}
\end{aligned}$$

We plug in the definition of the conformal factor $\lambda^c_{\boldsymbol{x}_{\mathrm{H}}} = \frac{2}{1-c\|\boldsymbol{x}_{\mathrm{H}}\|^2}$ and its derivative in the last step.

The term $\frac{1}{\sqrt{1+F(\boldsymbol{x}_{\mathrm{H}})^2}} \leq 1$ in Equation 17 cannot blow up. However, the gradients in Equation 18 grow with $O\big((1 - c\|\boldsymbol{x}_{\mathrm{H}}\|^2)^{-2}\big)$ for samples $\boldsymbol{x}_{\mathrm{H}}$ close to the boundary of the Poincaré Ball.

### A.3 Hyperboloid Exponential Map Gradients

The exponential map of the Hyperboloid at the origin $\bar{\boldsymbol{0}} = (1/\sqrt{c}, 0, \ldots, 0)$ maps a tangent vector $\boldsymbol{v} = [0, \boldsymbol{x}_{\mathrm{E}}] \in \mathcal{T}_{\bar{\boldsymbol{0}}}\mathcal{M}$ to the Hyperboloid (Bdeir et al., 2024):

$$\exp_{\bar{\boldsymbol{0}}}(\boldsymbol{v}) = \frac{1}{\sqrt{c}}\left[\cosh(\sqrt{c}\|\boldsymbol{x}_{\mathrm{E}}\|),\ \sinh(\sqrt{c}\|\boldsymbol{x}_{\mathrm{E}}\|)\frac{\boldsymbol{x}_{\mathrm{E}}}{\|\boldsymbol{x}_{\mathrm{E}}\|}\right]^{\top} , \tag{19}$$

where the first element is a scalar **time component** and the remaining elements constitute the **space component**.

The derivative of the **time component** is a $d + 1$-dimensional vector whose first element is zero:

$$\frac{\partial}{\partial\boldsymbol{v}}\frac{\cosh(\sqrt{c}\|\boldsymbol{x}_{\mathrm{E}}\|)}{\sqrt{c}} = \left[0, \sinh(\sqrt{c}\|\boldsymbol{x}_{\mathrm{E}}\|)\frac{\boldsymbol{x}_{\mathrm{E}}}{\|\boldsymbol{x}_{\mathrm{E}}\|}\right]^{\top} . \tag{20}$$

For the derivative of the **space component**, we start by reformulating it as:

$$\frac{\partial}{\partial\boldsymbol{v}}\frac{\sinh(\sqrt{c}\|\boldsymbol{x}_{\mathrm{E}}\|)\boldsymbol{x}_{\mathrm{E}}}{\sqrt{c}\|\boldsymbol{x}_{\mathrm{E}}\|} = \left[\boldsymbol{0}, \frac{\partial}{\partial\boldsymbol{v}}f(\|\boldsymbol{x}_{\mathrm{E}}\|)\boldsymbol{x}_{\mathrm{E}}\right] = \left[\boldsymbol{0}, f(\|\boldsymbol{x}_{\mathrm{E}}\|)\boldsymbol{I}_d + \frac{f'(\|\boldsymbol{x}_{\mathrm{E}}\|)}{\|\boldsymbol{x}_{\mathrm{E}}\|}\boldsymbol{x}_{\mathrm{E}}\boldsymbol{x}_{\mathrm{E}}^{\top}\right] , \tag{21}$$

where $f(\|\boldsymbol{x}_{\mathrm{E}}\|) = \frac{\sinh(\sqrt{c}\|\boldsymbol{x}_{\mathrm{E}}\|)}{\sqrt{c}\|\boldsymbol{x}_{\mathrm{E}}\|}$, $\boldsymbol{I}_d$ is the $d \times d$ identity matrix, and $\boldsymbol{0} \in \mathbb{R}^d$.

For $f'(\|\boldsymbol{x}_{\mathrm{E}}\|)$, we have:

$$f'(\|\boldsymbol{x}_{\mathrm{E}}\|) = \frac{\sqrt{c}\|\boldsymbol{x}_{\mathrm{E}}\|\cosh(\sqrt{c}\|\boldsymbol{x}_{\mathrm{E}}\|) - \sinh(\sqrt{c}\|\boldsymbol{x}_{\mathrm{E}}\|)}{\sqrt{c}\|\boldsymbol{x}_{\mathrm{E}}\|^2} . \tag{22}$$

Plugging Equation 22 into Equation 21 yields:

$$\frac{\sinh(\sqrt{c}\|\boldsymbol{x}_{\mathrm{E}}\|)}{\sqrt{c}\|\boldsymbol{x}_{\mathrm{E}}\|}\boldsymbol{I}_d + \frac{\sqrt{c}\|\boldsymbol{x}_{\mathrm{E}}\|\cosh(\sqrt{c}\|\boldsymbol{x}_{\mathrm{E}}\|) - \sinh(\sqrt{c}\|\boldsymbol{x}_{\mathrm{E}}\|)}{\sqrt{c}\|\boldsymbol{x}_{\mathrm{E}}\|^3}\boldsymbol{x}_{\mathrm{E}}\boldsymbol{x}_{\mathrm{E}}^{\top} \tag{23}$$

We arrive at the Jacobian of the exponential map by putting Equation 20 and Equation 23 together:

$$\frac{\partial\boldsymbol{x}_{\mathrm{H}}}{\partial\boldsymbol{v}} = \frac{\partial}{\partial\boldsymbol{v}}\exp_{\bar{\boldsymbol{0}}}(\boldsymbol{v}) = \begin{bmatrix} 0 & \sinh(\sqrt{c}\|\boldsymbol{x}_{\mathrm{E}}\|)\frac{\boldsymbol{x}_{\mathrm{E}}^{\top}}{\|\boldsymbol{x}_{\mathrm{E}}\|} \\ \boldsymbol{0} & \frac{\sinh(\sqrt{c}\|\boldsymbol{x}_{\mathrm{E}}\|)}{\sqrt{c}\|\boldsymbol{x}_{\mathrm{E}}\|}\boldsymbol{I}_d + \frac{\sqrt{c}\|\boldsymbol{x}_{\mathrm{E}}\|\cosh(\sqrt{c}\|\boldsymbol{x}_{\mathrm{E}}\|)-\sinh(\sqrt{c}\|\boldsymbol{x}_{\mathrm{E}}\|)}{\sqrt{c}\|\boldsymbol{x}_{\mathrm{E}}\|^3}\boldsymbol{x}_{\mathrm{E}}\boldsymbol{x}_{\mathrm{E}}^{\top} \end{bmatrix} . \tag{24}$$

## A.4 PROOFS

*Lemma 4.1.*

$$\|f(\boldsymbol{W}\boldsymbol{x} + \boldsymbol{b})\|_2 - \|f(0)\|_2 \leq \|f(\boldsymbol{W}\boldsymbol{x} + \boldsymbol{b}) - f(0)\|_2$$
$$\overset{(i)}{\leq} L\|\boldsymbol{W}\boldsymbol{x} + \boldsymbol{b}\|_2$$
$$\overset{(ii)}{\leq} L\|\boldsymbol{W}\|_2\|\boldsymbol{x}\|_2 + L\|\boldsymbol{b}\|_2,$$

where $(i)$ uses the Lipschitz property of $f$ and $(ii)$ follows from the definition of the induced matrix norm. The special case follows directly by observing that $\mathrm{ReLU}$ is 1-Lipschitz with $\mathrm{ReLU}(0) = 0$. $\square$

*Proposition 4.2.* First, we bound the norm of the normalized feature vector. Recall that $\mathrm{RMS}(\boldsymbol{x}) = \boldsymbol{x}/\mu(\boldsymbol{x})$ with $\mu(\boldsymbol{x}) = \sqrt{\varepsilon + \frac{1}{d}\|\boldsymbol{x}\|_2^2}$. Then,

$$\|\mathrm{RMS}(\boldsymbol{x})\|_2^2 = \frac{\|\boldsymbol{x}\|_2^2}{\varepsilon + \frac{1}{d}\|\boldsymbol{x}\|_2^2} < \frac{\|\boldsymbol{x}\|_2^2}{\frac{1}{d}\|\boldsymbol{x}\|_2^2} = d,$$

and

$$\|\hat{\boldsymbol{x}}\|_2 = \left\|\frac{1}{\sqrt{d}}f\big(\mathrm{RMS}(\boldsymbol{x})\big)\right\|_2 \overset{(i)}{\leq} \frac{1}{\sqrt{d}}(\|f(\boldsymbol{0})\|_2 + L\|\mathrm{RMS}(\boldsymbol{x})\|_2) < \frac{1}{\sqrt{d}}\|f(\boldsymbol{0})\|_2 + L,$$

where $(i)$ follows from Lemma 4.1, conclude the first part.
Second, we bound the conformal factor. Let $\boldsymbol{v} = \exp_{\bar{0}}(\hat{\boldsymbol{x}}) = \tanh(\sqrt{c}\|\hat{\boldsymbol{x}}\|)\frac{\hat{\boldsymbol{x}}}{\sqrt{c}\|\hat{\boldsymbol{x}}\|}$. We have:

$$\|\boldsymbol{v}\|_2 = \left\|\tanh(\sqrt{c}\|\hat{\boldsymbol{x}}\|_2)\frac{\hat{\boldsymbol{x}}}{\sqrt{c}\|\hat{\boldsymbol{x}}\|_2}\right\|_2 = \frac{\tanh(\sqrt{c}\|\hat{\boldsymbol{x}}\|_2)}{\sqrt{c}\|\hat{\boldsymbol{x}}\|_2}\|\hat{\boldsymbol{x}}\|_2 = \frac{\tanh(\sqrt{c}\|\hat{\boldsymbol{x}}\|_2)}{\sqrt{c}}.$$

Applying the previous equality gives

$$\lambda_{\boldsymbol{v}}^c = \frac{2}{1 - c\|\boldsymbol{v}\|_2^2} = \frac{2}{1 - \tanh(\sqrt{c}\|\hat{\boldsymbol{x}}\|_2)^2}$$

which can be further bounded by combining it with the bound on $\|\hat{x}\|$ and using the fact that $\cosh$ is a monotonically increasing function on $\mathbb{R}_{>0}$:

$$\lambda_{\boldsymbol{v}}^c = \frac{2}{1 - \tanh(\sqrt{c}\|\hat{\boldsymbol{x}}\|_2)^2} = 2\cosh^2\big(\sqrt{c}\,\|\hat{x}\|\big) \leq 2\cosh^2\left(\sqrt{c}\left(\frac{1}{\sqrt{d}}\|f(0)\|_2 + L\right)\right).$$

$\square$

## B ADDITIONAL BACKGROUND

### B.1 TRUST-REGION POLICY OPTIMIZATION (TRPO)

TRPO maximizes cumulative reward through gradient ascent on a surrogate objective (Equation equation 25);

$$\max_{\theta} \quad \mathbb{E}_t\left[\frac{\pi_\theta(a_t \mid s_t)}{\pi_{\theta_{\mathrm{old}}}(a_t \mid s_t)}A_t^{\pi_{\theta_{\mathrm{old}}}}\right] \tag{25}$$
$$\text{subject to} \quad \mathbb{E}_t\Big[\mathrm{D}_{\mathrm{KL}}[\pi_\theta(a_t \mid s_t) \parallel \theta_{\mathrm{old}}]\Big] \leq \delta\,.$$

Additionally, it enforces an average KL-divergence constraint to keep the new policy close to the data-generating policy. Theoretically, optimizing this objective guarantees monotonic improvement (Schulman et al., 2015). In practice, several approximations are used for deep neural networks. Nevertheless, TRPO tends to retain the monotonic improvement of its theory.

## B.2 DOUBLE DEEP Q-NETWORK

Deep Q Network (DQN) (Mnih et al., 2015) learns the optimal Q-function for discrete action spaces by minimizing a mean-squared error loss against an off-policy bootstrap target while reusing replayed transitions. The standard target is $Q^\pi_{\text{tar:DQN}}(s, a) = r + \gamma \max_{a'} Q^\pi(s', a')$, which, together with experience replay and a periodically updated target network, stabilizes training. However, because the same function approximator effectively selects both the maximizing action and evaluates its value under noise and approximation error, the max operator induces systematic overestimation bias (Sutton & Barto, 2018).

Double DQN (DDQN) (van Hasselt et al., 2016) reduces the overestimation bias that arises when the same network both selects and evaluates the maximized next-state value. DDQN uses the online network with parameters $\theta$ to select the greedy next action, and a target network with parameters $\varphi$ to evaluate that action when calculating the TD target: $Q^\pi_{\text{tar:DDQN}}(s, a) = r + \gamma Q^\pi_\varphi(s', \arg\max_{a'} Q^\pi_\theta(s', a'))$. This decorrelates action selection from evaluation, effectively mitigating overestimation bias.

## B.3 HYPERBOLIC MULTINOMIAL LOGISTIC REGRESSION

Multinomial Logistic Regression (MLR) in hyperbolic space $\mathcal{M}$ is defined as the log-linear model with parameters $\boldsymbol{z}_k \in \mathbb{R}^d$, $r_k \in \mathbb{R}$ that predicts the probability $p(\boldsymbol{y} = k \mid \boldsymbol{x})$ of an input $\boldsymbol{x} \in \mathcal{M} \simeq \mathbb{R}^d$ belonging to a specific class $k \in \{1, \dots, K\}$:

$$p(\boldsymbol{y} = k \mid \boldsymbol{x}) \propto \exp(v_{\boldsymbol{z}_k, r_k}(\boldsymbol{x})), \quad v_{\boldsymbol{z}_k, r_k}(\boldsymbol{x}) = \|\boldsymbol{z}_k\|_{\mathcal{T}_{\boldsymbol{p}_k}\mathcal{M}} \, d_{\mathcal{M}}(\boldsymbol{x}, \mathcal{H}_{\boldsymbol{z}_k, r_k}).$$

Here, $\exp(v_{\boldsymbol{z}_k, r_k}(\boldsymbol{x}))$ is the logit for class $k$ and $v_{\boldsymbol{z}_k, r_k}(\boldsymbol{x})$ the signed distance to the margin hyperplane $\mathcal{H}_{\boldsymbol{z}_k, r_k}$. To prevent over-parametrization, each hyperplane is characterized by aligning its normal vector $\boldsymbol{a}_k$ and shift $\boldsymbol{p}_k$, requiring only $d + 1$-parameters per hyperplane (Shimizu et al., 2021; Bdeir et al., 2024). To leverage established Euclidean optimization algorithms, all parameters are maintained in Euclidean space and mapped to their hyperbolic counterparts. The normal vector $\boldsymbol{a}_k \in \mathcal{T}_{\boldsymbol{p}_k}\mathcal{M}$ is obtained by parallel transporting the Euclidean parameter $\boldsymbol{z}_k \in \mathbb{R}^d$ to the origin $\bar{\boldsymbol{0}}$: $\boldsymbol{a}_k = PT_{\bar{\boldsymbol{0}} \to \boldsymbol{p}}(\boldsymbol{z}_k)$ with $\boldsymbol{z}_k \in \mathcal{T}_{\bar{\boldsymbol{0}}}\mathcal{M}$. The hyperplane's shift $\boldsymbol{p}_k \in \mathcal{M}$ is defined as scalar multiple of the same unit tangent vector $\boldsymbol{p}_k = \exp_{\bar{0}}\left(\frac{r_k}{||\boldsymbol{z}_k||} \boldsymbol{z}_k\right)$ with $r_k \in \mathbb{R}$.

### B.3.1 POINCARÉ BALL MLR

For the Poincaré Ball we have:

$$\boldsymbol{p}_k = \exp_{\bar{0}}\left(\frac{r_k}{||\boldsymbol{z}_k||} \boldsymbol{z}_k\right) = \frac{\tanh\left(\sqrt{c}\, r_k\right)}{\sqrt{c}\, ||\boldsymbol{z}_k||} \boldsymbol{z}_k,$$

$$\boldsymbol{a}_k = PT_{\bar{\boldsymbol{0}} \to \boldsymbol{p}}(\boldsymbol{z}_k) = \left(1 - \tanh^2\left(\sqrt{c}\, r_k\right)\right) \boldsymbol{z}_k,$$

$$\mathcal{H}_{\boldsymbol{z}_k, r_k} = \mathcal{H}_{\boldsymbol{a}_k, \boldsymbol{p}_k} = \left\{\boldsymbol{x} \in \mathbb{P}^d_c : \langle \boldsymbol{a}_k, \ominus \boldsymbol{p}_k \oplus \boldsymbol{x} \rangle = 0\right\},$$

$$d_{\mathbb{P}^d_c}(\boldsymbol{x}, \mathcal{H}_{\boldsymbol{z}_k, r_k}) = d_{\mathbb{P}}(\boldsymbol{x}, \mathcal{H}_{\boldsymbol{a}_k, \boldsymbol{p}_k}) = \frac{1}{\sqrt{c}} \sinh^{-1}\left(\frac{2\sqrt{c}\, |\langle \boldsymbol{a}_k, \ominus \boldsymbol{p}_k \oplus \boldsymbol{x} \rangle|}{(1 - c||\ominus \boldsymbol{p}_k \oplus \boldsymbol{x}||^2)||\boldsymbol{a}_k||}\right),$$

where $\ominus$ and $\oplus$ denote the Mobius addition and subtraction (Ganea et al., 2018). The Poincaré Ball MLR layer (Shimizu et al., 2021) can then be summarized as

$$v^{\text{HNN++}}_{\boldsymbol{z}_k, r_k}(\boldsymbol{x}) = \frac{2\, ||\boldsymbol{z}_k||}{\sqrt{c}} \sinh^{-1}\left((1 - \lambda^c_{\boldsymbol{x}}) \sinh(2\sqrt{c}\, r_k) + \sqrt{c}\, \lambda^c_{\boldsymbol{x}} \cosh(2\sqrt{c}\, r_k) \left\langle \frac{\boldsymbol{z}_k}{||\boldsymbol{z}_k||}, \boldsymbol{x} \right\rangle\right).$$

### B.3.2 HYPERBOLOID MLR

For the Hyperboloid, the tangent vectors $\boldsymbol{z}_k$ are represented in $\mathbb{R}^d$ rather than the full $\mathbb{R}^{d+1}$ space that would typically characterize tangent vectors at the origin of the Hyperboloid. However, to preserve the correct number of degrees of freedom, we omit the time component, which is constrained to be

zero. That said, we have:

$$\boldsymbol{p}_k = \exp_{\bar{\boldsymbol{0}}}\left(\frac{r_k}{||\boldsymbol{z}_k||}\,\boldsymbol{z}_k\right) = \frac{1}{\sqrt{c}}\left[\cosh(\sqrt{c}\,r_k),\ \sinh(\sqrt{c}\,r_k)\frac{\boldsymbol{z}_k}{||\boldsymbol{z}_k||}\right]^{\top}$$

$$\boldsymbol{a}_k = PT_{\bar{\boldsymbol{0}}\to\boldsymbol{p}}(\boldsymbol{z}_k) = \left[\sinh(\sqrt{c}\,r_k)||\boldsymbol{z}_k||,\ \cosh(\sqrt{c}\,r_k)\boldsymbol{z}_k\right]^{\top},$$

$$\mathcal{H}_{\boldsymbol{z}_k,r_k} = \mathcal{H}_{\boldsymbol{a}_k,\boldsymbol{p}_k} = \left\{\boldsymbol{x}\in\mathbb{H}_c^d:\ \langle\boldsymbol{a}_k,\boldsymbol{x}\rangle_{\mathcal{L}}=0\right\},$$

$$d_{\mathbb{H}_c^d}(\boldsymbol{x},\mathcal{H}_{\boldsymbol{z}_k,r_k}) = \frac{1}{\sqrt{c}}\sinh^{-1}\left(\frac{\sqrt{c}}{||\boldsymbol{z}_k||}\left(-x_0\,\sinh(\sqrt{c}\,r_k)\,||\boldsymbol{z}_k|| + \cosh(\sqrt{c}\,r_k)\langle\boldsymbol{z}_k,\boldsymbol{x}_s\rangle\right)\right).$$

The Hyperboloid MLR layer (Bdeir et al., 2024) can then be summarized as

$$v_{\boldsymbol{z}_k,r_k}^{\text{HB}}(\boldsymbol{x}) = \frac{||\boldsymbol{z}_k||}{\sqrt{c}}\sinh^{-1}\left(\frac{\sqrt{c}}{||\boldsymbol{z}_k||}\left(-x_0\,\sinh(\sqrt{c}\,r_k)\,||\boldsymbol{z}_k|| + \cosh(\sqrt{c}\,r_k)\langle\boldsymbol{z}_k,\boldsymbol{x}_s\rangle\right)\right), \qquad (26)$$

where $\boldsymbol{x}_s = (x_1,\dots,x_d)$ denotes the *space component* and $x_0$ the *time* component.

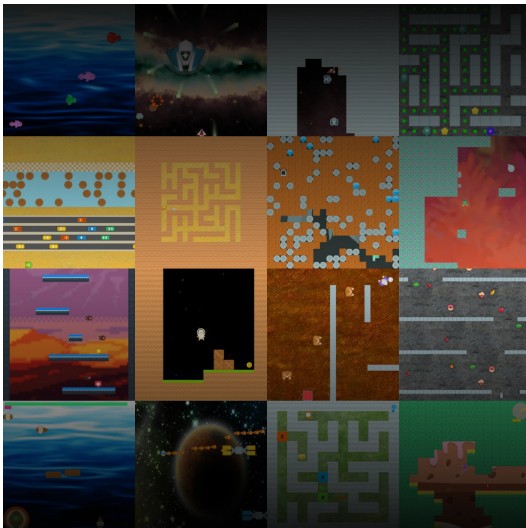

Figure 10: Visualization of all ProcGen environments.

## C    ENVIRONMENTS

In this Section, we review the environments used in our paper and discuss evaluation differences to existing methods.

### C.1    PROCGEN

ProcGen (Cobbe et al., 2020) uses RGB frames of size $64 \times 64 \times 3$ as observations. We visualize the 16 games in Figure 10. The action space is discrete with 15 actions. For training, we follow the protocol by (Cetin et al., 2023): fix difficulty to "easy" and train on the first 200 levels (seeds 0–199). For testing, we evaluate on all levels of the easy distribution. For the table, we run a single end-of-training evaluation on 100 parallel environments sampled from the train and test distribution, respectively. We then normalize the scores for each individual run before aggregating.

### C.2    ATARI

The Arcade Learning Environment (Bellemare et al., 2015; Towers et al., 2024) provides a standardized interface for deep RL research based on dozens of Atari 2600 games. Of these, 57 are commonly used in evaluation. The observations are RGB frames $210 \times 160 \times 3$, which are preprocessed via grayscaling, downsampling to $84 \times 84$, and frame stacking. The action space consists of up to 18 discrete joystick/button combinations, with most games using a subset and action repeat (frame skipping) to help with jittering. The rewards are clipped to $\{-1, 0, +1\}$. As the game dynamics are naturally deterministic, the benchmark uses randomized no-op resets as outlined in the original DQN paper (Mnih et al., 2015) and cleanRL (Huang et al., 2022).

### C.3    EVALUATION DIFFERENCES WITH EXISTING WORKS

Our paper builds on the seminal work by Cetin et al. (2023). However, we struggled to reproduce their results, which we believe is mainly due to three reasons. First, their source code does not use seeding. As deep RL is notoriously seed-dependent (Henderson et al., 2018; Agarwal et al., 2021), we find exact reproduction impossible. Second, we use a different implementation for the mathematical operations of hyperbolic geometry, which possibly affects the results. This issue is known within the hyperbolic deep learning community (Katsman & Gilbert, 2025). Third, we use different versions of PyTorch and Python. Additionally, our evaluation follows a slightly different protocol (see Appendix C.1), and we use Pytorch's evaluation mode before generating results for our agents. We hope that by releasing our complete code, we can take a step towards more reproducible research in hyperbolic deep RL.

Table 2: **Wall-clock results.**

|  | **ProcGen forward** | **NameThisGame** |
|---|---|---|
| **Euclidean** | 14ms | 17h52m |
| **Hyper+S-RYM** (Cetin et al., 2023) | 19.3ms | 58h21m |
| **HYPER++** | 14.7ms | 35h25m |

## D    EXPERIMENTAL DETAILS

### D.1    HARDWARE & RUNTIME

For our experiments, we used Nvidia A100 GPUs. For ProcGen, we can train up to four agents in parallel on a single GPU with 40GB. We report wall-clock times for forward passes on ProcGen and for full runs on NameThisGame in Table 2. Note that agent performance can be a confounding factor for results when timing on full experiments because agent performance can either be positively or negatively correlated with episode length. For NameThisGame, e.g., better agents generate longer episodes. We average over 100 passes for the forward pass results and five seeds for the NameThisGame results.

### D.2    NETWORK ARCHITECTURE

For ProcGen, we use the same Impala-ResNet (Espeholt et al., 2018) as (Cetin et al., 2023), which we visualize in Figure 11. Our modifications are shaded in purple. They consist of

1. using RMSNorm (Zhang & Sennrich, 2019) before scaling the Euclidean features,
2. using TanH instead of ReLU as penultimate activation,
3. applying learned feature scaling before the exponential map, and
4. using the Hyperboloid instead of the Poincaré Ball.

For Atari, we use the NatureCNN (Mnih et al., 2015) architecture with the same modifications applied as for ProcGen.

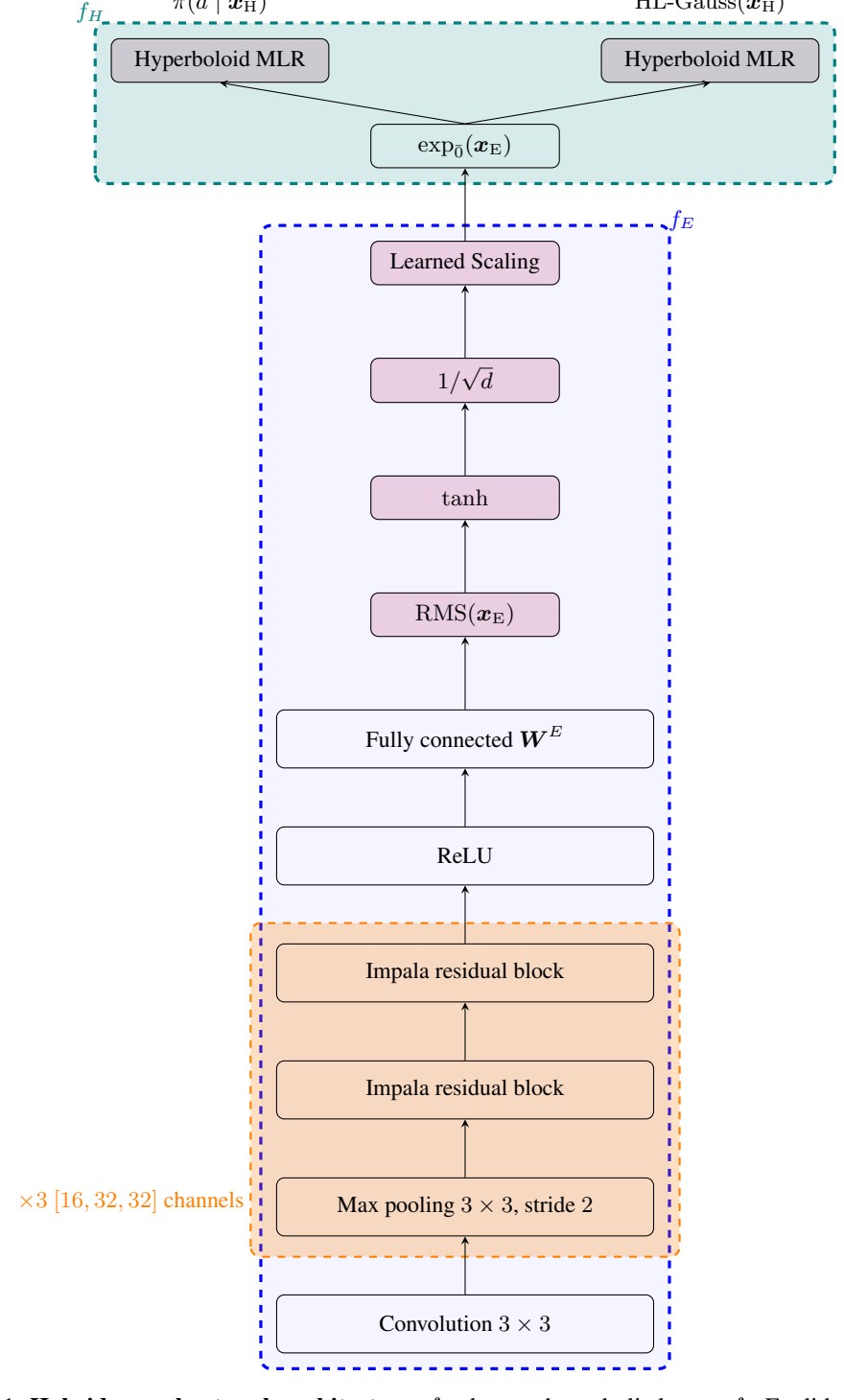

Figure 11: **Hybrid neural network architecture.** $f_H$ denotes hyperbolic layers, $f_E$ Euclidean layers. Components that are specific to HYPER++ are shaded in purple.

Table 4: PPO hyperparameters for ProcGen.

| PPO hyperparameters | |
| --- | --- |
| Parallel environments | 64 |
| Stacked input frames | 1 |
| Steps per rollout | 16384 |
| Training epochs per rollout | 3 |
| Batch size | 2048 |
| Normalize rewards | True |
| Discount $\gamma$ | 0.999 |
| GAE $\lambda$ (Schulman et al., 2015) | 0.95 |
| PPO clipping | 0.2 |
| Entropy coefficient | 0.01 |
| Value coefficient | 0.5 |
| Shared network | True |
| Impala stack filter sizes | 16, 32, 32 |
| Default latent representation size | 32 |
| Optimizer | Adam (Kingma & Ba, 2015) |
| Optimizer learning rate | $5 \times 10^{-4}$ |
| Optimizer stabilization constant ($\epsilon$) | $1 \times 10^{-5}$ |
| Maximum gradient norm. | 0.5 |

Table 5: DDQN hyperparameters for Atari.

| Atari hyperparameters | |
| --- | --- |
| Environment steps | 10M |
| Discount $\gamma$ | 0.99 |
| $\epsilon$ start | 1 |
| $\epsilon$ end | 0.01 |
| Exploration fraction | 10% of steps |
| Replay buffer size | 1M |
| Target network update frequency | 1000 |
| Default latent representation size | 512 |
| Batch size | 32 |
| Training frequency | 4 |
| Optimizer | Adam (Kingma & Ba, 2015) |
| Optimizer learning rate | $1 \times 10^{-4}$ |
| Optimizer stabilization constant ($\epsilon$) | $2.5 \times 10^{-5}$ |

## D.3 HYPERPARAMETERS

We specify the hyperparameters for all PPO agents in Table 4, for the DDQN agents in Table 5, and for PPG in Table 6. For DDQN and PPG, we use the hyperparameters and preprocessing steps from cleanRL (Huang et al., 2022). Additional parameters for our method are in Table 3. On ProcGen, our agent uses a latent dimension $d = 64$ compared to $d = 32$ for Hyper+S-RYM. For HL-Gauss (Imani & White, 2018), we use the default parameters specified by Farebrother et al. (2024). We set the learned scaling hyperparameter to $\alpha = 0.95$ without tuning for all experiments. When using RMSNorm or LayerNorm, we do not use affine parameters, because they can overfit (Xu et al., 2019) to the training set.

Table 3: HYPER++ hyperparameters.

| HYPER++ | |
| --- | --- |
| Loss: Number of bins | 51 |
| Loss: Min clip | -10.0 |
| Loss: Max clip | +10.0 |
| Last Euclidean Activation | TanH |
| Learned scaling $\alpha$ | 0.95 |

Table 6: PPG hyperparameters for ProcGen.

| PPG hyperparameters | |
| --- | --- |
| Policy iterations ($N_\pi$) | 32 |
| Policy phase epochs ($E_\pi$) | 1 |
| Value phase epochs ($E_V$) | 1 |
| Auxiliary phase epochs ($E_{\text{aux}}$) | 6 |
| Behavior cloning coefficient ($\beta_{\text{clone}}$) | 1.0 |
| Auxiliary phase minibatches | 4 |
| Gradient accumulation steps | 1 |

# E   ADDITIONAL RESULTS

## E.1   FULL PPO RESULTS

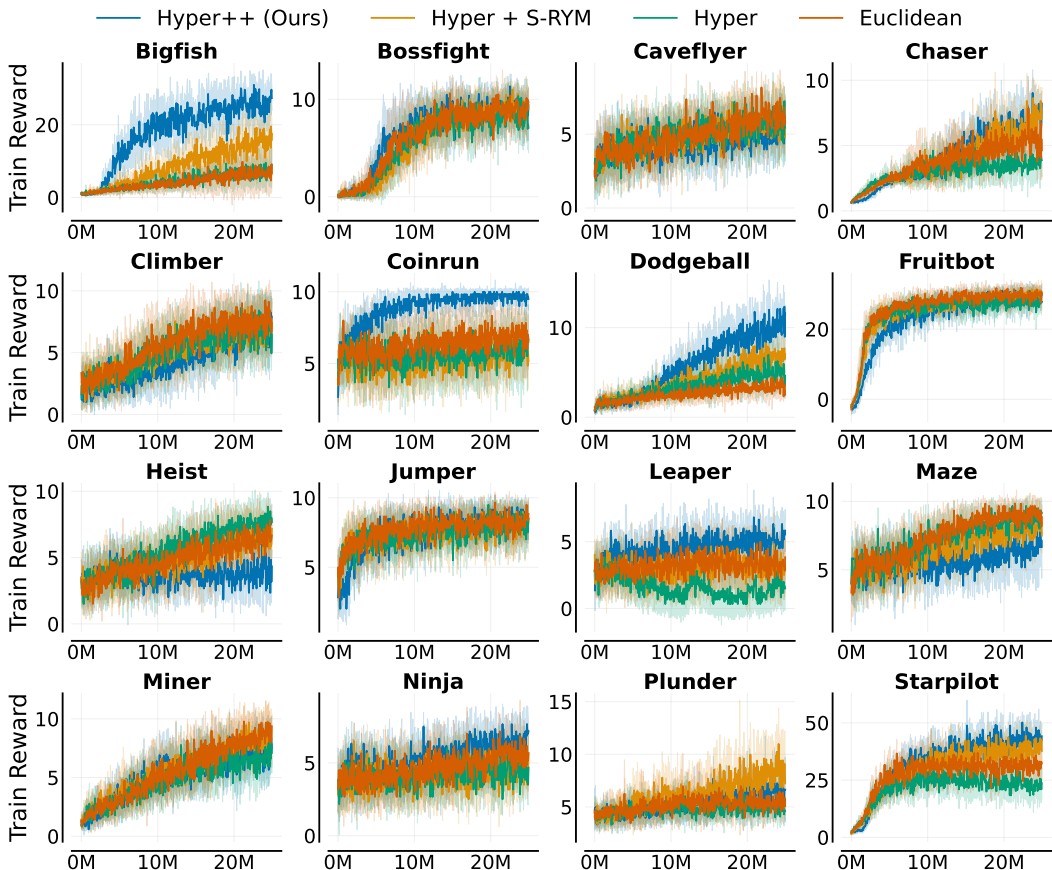

Figure 12: **PPO ProcGen Train Learning curves.**

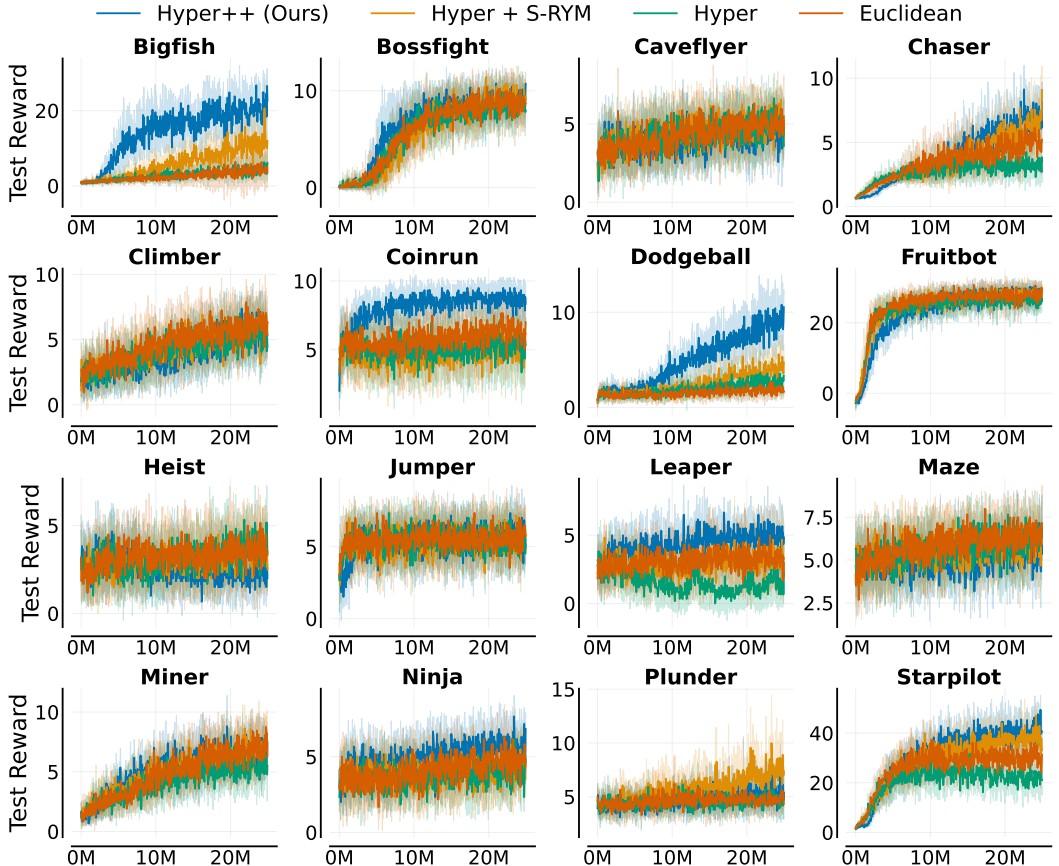

Figure 13: **PPO ProcGen Test Learning curves.**

Table 7: **ProcGen Train Results (mean ± std).**

|  | **Hyper** | **Euclidean** | **Hyper + S-RYM** | **Hyper++ (Ours)** |
|---|---|---|---|---|
| **bigfish** | 7.12±1.7 | 7.81±5.0 | 17.38±3.3 | **25.66±2.8** |
| **starpilot** | 20.43±1.5 | 31.14±4.4 | 38.92±1.9 | **43.43±3.7** |
| **dodgeball** | 5.42±0.9 | 3.28±0.7 | 6.26±0.7 | **10.28±1.0** |
| **coinrun** | 6.25±1.4 | 7.37±1.2 | 5.53±1.0 | **9.67±0.2** |
| **leaper** | 1.65±1.5 | 3.35±1.0 | 3.23±2.0 | **5.25±0.9** |
| **ninja** | 4.63±0.8 | 5.65±0.9 | 4.37±0.5 | **6.67±0.6** |
| **fruitbot** | 27.09±0.9 | 29.40±0.8 | 29.36±0.7 | **29.89±0.4** |
| **jumper** | 8.27±0.3 | 8.17±0.6 | 8.17±0.7 | **8.53±0.3** |
| **bossfight** | 8.49±0.7 | **9.40±0.5** | 9.38±0.7 | 9.27±0.7 |
| **miner** | 6.86±0.5 | 8.52±0.8 | **8.53±1.0** | 8.10±1.4 |
| **chaser** | 3.85±0.5 | 5.04±0.9 | **7.60±0.8** | 7.12±0.9 |
| **climber** | 6.72±0.7 | **7.79±0.5** | 7.43±0.8 | 7.27±0.7 |
| **caveflyer** | 5.81±0.5 | **6.41±0.3** | 6.13±0.6 | 5.04±0.6 |
| **maze** | **8.88±0.5** | 8.88±0.7 | 7.80±0.8 | 6.90±1.4 |
| **plunder** | 4.99±0.9 | 5.30±0.5 | **8.81±1.5** | 6.23±0.9 |
| **heist** | **7.57±0.6** | 6.78±1.2 | 6.68±0.8 | 3.95±1.5 |
| **IQM (normalized)** | 0.37 | 0.45 | 0.46 | **0.55** |

Table 8: **ProcGen Test Results (mean ± std).**

|  | **Hyper** | **Euclidean** | **Hyper + S-RYM** | **Hyper++ (Ours)** |
|---|---|---|---|---|
| **bigfish** | 4.80±1.9 | 3.60±3.4 | 11.88±2.7 | **20.01±4.8** |
| **starpilot** | 22.21±4.0 | 30.44±3.1 | 35.59±2.8 | **42.49±2.6** |
| **dodgeball** | 2.38±0.2 | 1.79±0.3 | 3.95±0.8 | **9.41±1.2** |
| **coinrun** | 5.07±1.3 | 6.13±1.2 | 5.57±1.1 | **8.72±0.4** |
| **leaper** | 1.38±1.4 | 3.58±1.3 | 3.33±2.0 | **5.13±1.0** |
| **ninja** | 4.08±0.5 | 4.65±0.5 | 4.27±0.4 | **5.68±0.4** |
| **fruitbot** | 26.37±2.0 | 27.88±0.7 | **28.44±0.5** | 28.28±0.9 |
| **jumper** | 5.28±0.6 | 5.08±0.5 | 5.28±0.4 | **5.30±0.4** |
| **bossfight** | 8.50±0.7 | **9.50±0.8** | 9.03±0.8 | 9.40±0.7 |
| **miner** | 5.78±1.1 | 7.15±0.6 | 7.07±1.1 | **7.21±0.7** |
| **chaser** | 3.34±0.6 | 4.55±1.0 | 6.67±0.7 | **7.27±0.9** |
| **climber** | 5.41±0.7 | **5.89±0.7** | 5.52±0.9 | 5.86±1.2 |
| **caveflyer** | 4.91±0.5 | **5.22±0.3** | 4.83±0.5 | 4.12±0.4 |
| **maze** | 6.38±0.6 | **6.43±0.5** | 5.85±0.3 | 5.17±0.8 |
| **plunder** | 4.58±0.7 | 4.71±0.5 | **7.45±0.7** | 5.82±1.6 |
| **heist** | 3.58±0.5 | **3.68±0.5** | 3.38±0.5 | 2.28±1.0 |
| **IQM (normalized)** | 0.19 | 0.26 | 0.27 | **0.41** |

Table 9: **ProcGen ablation Train results (mean ± std).**

| | Ours | Ours+Poinc. | Ours w/o Scaling | Ours+MSE | Ours+C51 | Ours w/o RMS | Ours+SN (Full) | Ours+SN (Penult.) | Euc.+RMS | Euc.+HL-Gauss | Ours+Euc. |
|---|---|---|---|---|---|---|---|---|---|---|---|
| bigfish | **27.02±1.9** | 17.35±4.0 | 20.28±3.1 | 21.72±4.3 | 19.93±3.3 | 1.04±0.2 | 0.94±0.1 | 1.04±0.2 | 5.95±4.7 | 11.59±3.8 | 24.87±1.6 |
| starpilot | 41.80±3.8 | 40.22±2.0 | **42.97±2.0** | 40.58±4.6 | 37.71±4.0 | 2.71±0.4 | 2.78±0.4 | 2.86±0.3 | 25.22±3.5 | 32.95±7.4 | 38.78±3.3 |
| dodgeball | **10.46±1.0** | 8.15±1.0 | 8.12±0.5 | 8.06±1.5 | 7.95±1.4 | 1.86±0.3 | 1.64±0.3 | 1.73±0.1 | 5.08±1.0 | 4.63±1.5 | 9.72±1.3 |
| coinrun | **9.68±0.2** | 9.52±0.1 | 9.48±0.3 | 8.22±0.3 | 9.55±0.2 | 5.68±0.2 | 5.78±0.4 | 6.07±0.2 | 8.45±1.4 | 9.27±0.2 | 9.55±0.2 |
| leaper | **4.87±0.7** | 3.47±0.9 | 3.83±0.8 | 2.80±0.5 | 4.22±1.2 | 3.57±0.6 | 3.57±0.4 | 3.35±0.3 | 3.33±1.1 | 4.13±0.9 | 4.38±1.0 |
| ninja | 5.90±0.7 | **6.45±0.9** | 5.62±0.7 | 5.27±0.6 | 5.13±0.9 | 3.43±0.4 | 3.17±0.6 | 3.32±0.5 | 5.95±0.3 | 4.32±0.3 | 5.07±0.4 |
| fruitbot | **30.13±1.2** | 29.89±0.6 | 30.04±0.8 | 29.83±1.3 | 27.23±0.9 | -1.49±0.3 | -1.82±0.3 | -1.54±0.2 | 28.08±1.1 | 29.70±1.7 | 27.49±0.9 |
| jumper | 8.20±0.3 | 8.68±0.4 | 8.57±0.3 | **8.87±0.3** | 7.75±0.4 | 3.08±0.6 | 3.05±0.7 | 3.10±0.3 | 8.57±0.3 | 7.55±1.4 | 8.37±0.6 |
| bossfight | 9.55±0.9 | 9.36±1.2 | 8.86±0.7 | **9.94±0.4** | 8.62±0.6 | 0.48±0.2 | 0.48±0.2 | 0.43±0.3 | 9.14±0.5 | 6.46±1.2 | 8.29±0.6 |
| miner | 7.49±0.6 | **7.93±0.8** | 7.22±1.2 | 7.32±1.2 | 4.53±1.5 | 1.54±0.4 | 1.43±0.3 | 1.27±0.2 | 6.09±2.5 | 4.65±2.0 | 6.68±1.0 |
| chaser | 6.66±1.0 | 7.51±0.9 | **8.03±0.8** | 6.23±0.9 | 4.02±0.6 | 0.63±0.0 | 0.69±0.0 | 0.66±0.0 | 4.73±2.0 | 4.70±1.4 | 4.87±0.7 |
| climber | 6.52±1.6 | 6.22±1.4 | 5.00±1.4 | **7.65±0.2** | 3.39±0.8 | 2.63±0.6 | 1.87±0.2 | 2.16±0.4 | 7.64±0.8 | 6.11±1.4 | 6.58±0.6 |
| caveflyer | 4.85±0.5 | 5.05±0.4 | 5.29±0.3 | **5.97±0.8** | 4.27±0.4 | 3.45±0.5 | 3.87±0.4 | 3.97±0.5 | 5.71±0.8 | 4.44±0.3 | 4.81±0.3 |
| maze | 6.78±2.1 | 8.48±1.4 | 9.28±0.5 | **9.72±0.1** | 5.60±0.2 | 5.17±0.3 | 5.42±0.7 | 5.12±0.3 | 9.38±0.3 | 8.52±1.3 | 6.95±2.0 |
| plunder | 6.10±1.7 | 7.77±1.7 | 6.89±0.5 | **8.47±1.1** | 6.37±0.6 | 4.61±0.3 | 4.74±0.2 | 4.58±0.5 | 5.46±0.5 | 5.26±1.1 | 7.68±2.1 |
| heist | 3.65±0.5 | 3.70±0.5 | 4.22±1.1 | **8.38±0.8** | 3.25±0.7 | 3.73±0.4 | 3.15±0.4 | 3.33±0.7 | 7.62±1.7 | 2.88±0.6 | 4.12±1.5 |
| IQM (normalized) | 0.51 | 0.50 | 0.50 | **0.55** | 0.33 | 0.01 | 0.01 | 0.01 | 0.45 | 0.32 | 0.45 |

Table 10: **ProcGen ablation Test results (mean ± std).**

| | Ours | Ours+Poinc. | Ours w/o Scaling | Ours+MSE | Ours+C51 | Ours w/o RMS | Ours+SN (Full) | Ours+SN (Penult.) | Euc.+RMS | Euc.+HL-Gauss | Ours+Euc. |
|---|---|---|---|---|---|---|---|---|---|---|---|
| bigfish | **21.56±3.5** | 12.52±3.8 | 15.02±2.3 | 16.47±4.6 | 14.77±4.7 | 1.05±0.1 | 1.01±0.1 | 0.96±0.1 | 3.71±2.9 | 7.60±3.6 | 19.31±2.7 |
| starpilot | 39.85±2.5 | 37.34±2.4 | 39.66±3.5 | **40.14±2.8** | 33.68±4.7 | 2.76±0.3 | 2.80±0.3 | 2.58±0.3 | 25.61±2.9 | 32.12±7.1 | 35.35±1.7 |
| dodgeball | 8.91±0.7 | 5.61±1.1 | 5.28±1.0 | 5.42±1.3 | 5.70±1.3 | 1.59±0.3 | 1.54±0.2 | 1.61±0.2 | 2.74±0.9 | 2.92±1.3 | **9.21±1.3** |
| coinrun | **8.75±0.4** | 8.58±0.4 | 8.75±0.5 | 7.75±0.7 | 8.67±0.3 | 5.25±0.6 | 5.35±0.3 | 5.40±0.5 | 7.63±1.2 | 8.05±0.2 | 8.38±0.3 |
| leaper | 4.62±1.0 | 3.58±0.9 | 4.42±1.1 | 2.47±0.2 | 4.20±1.4 | 3.30±0.5 | 3.18±0.5 | 3.27±0.3 | 3.37±1.1 | 3.88±0.8 | **4.67±0.7** |
| ninja | **5.57±0.6** | 5.33±0.8 | 4.70±0.5 | 4.78±0.2 | 5.35±0.4 | 3.28±0.5 | 3.37±0.6 | 3.40±0.2 | 5.13±0.4 | 3.20±1.1 | 4.77±0.6 |
| fruitbot | **28.62±0.9** | 28.04±1.3 | 27.76±1.3 | 28.25±1.2 | 27.21±1.3 | -1.69±0.2 | -1.74±0.3 | -1.65±0.3 | 26.82±1.3 | 28.17±0.9 | 27.04±1.0 |
| jumper | 5.37±0.3 | **5.73±0.2** | 5.38±0.5 | 5.52±0.4 | 5.23±0.6 | 2.50±0.3 | 2.40±0.4 | 2.60±0.4 | 5.37±0.1 | 5.03±0.6 | 5.22±0.7 |
| bossfight | **9.51±0.7** | 8.60±1.0 | 8.46±0.8 | 9.50±0.7 | 8.68±0.9 | 0.78±0.3 | 0.62±0.2 | 0.59±0.3 | 9.19±0.8 | 6.10±1.3 | 7.73±0.7 |
| miner | **7.42±1.2** | 7.02±0.5 | 6.44±1.4 | 6.23±1.0 | 4.40±1.4 | 1.62±0.4 | 1.48±0.3 | 1.33±0.2 | 5.25±2.0 | 4.10±2.1 | 6.18±0.7 |
| chaser | 6.75±0.8 | **7.08±1.2** | 6.94±0.6 | 5.84±0.8 | 3.99±0.5 | 0.65±0.0 | 0.65±0.0 | 0.64±0.0 | 4.67±2.1 | 4.24±1.1 | 4.84±0.5 |
| climber | 4.95±1.1 | 5.14±1.2 | 4.13±1.3 | 5.79±0.3 | 3.46±0.5 | 2.47±0.4 | 2.44±0.3 | 2.39±0.8 | **6.43±0.4** | 4.23±1.3 | 5.48±0.9 |
| caveflyer | 3.91±0.2 | 4.05±0.7 | 4.85±0.7 | **5.41±0.7** | 3.87±0.3 | 3.62±0.7 | 3.57±0.5 | 3.95±0.5 | 4.45±0.5 | 3.99±0.5 | 4.32±0.4 |
| maze | 5.58±1.5 | 5.40±0.7 | 5.87±0.6 | 6.18±0.8 | 5.22±0.9 | 5.22±0.4 | 5.08±0.7 | 5.27±0.5 | **6.92±0.3** | 5.35±1.0 | 5.55±0.4 |
| plunder | 5.08±0.7 | 6.88±1.6 | 6.16±1.0 | **7.59±1.3** | 5.94±0.6 | 4.44±0.3 | 4.42±0.3 | 4.45±0.3 | 4.83±0.5 | 4.33±1.1 | 6.39±1.3 |
| heist | 2.43±0.4 | 2.35±0.7 | 2.00±0.6 | **4.23±0.7** | 2.55±0.9 | 2.80±0.4 | 2.95±0.4 | 3.27±0.3 | 3.92±0.9 | 2.28±0.6 | 2.68±0.4 |
| IQM (normalized) | **0.40** | 0.34 | 0.33 | 0.33 | 0.27 | 0.00 | 0.00 | 0.00 | 0.28 | 0.20 | 0.35 |

## E.2 ADDITIONAL ABLATION METRICS

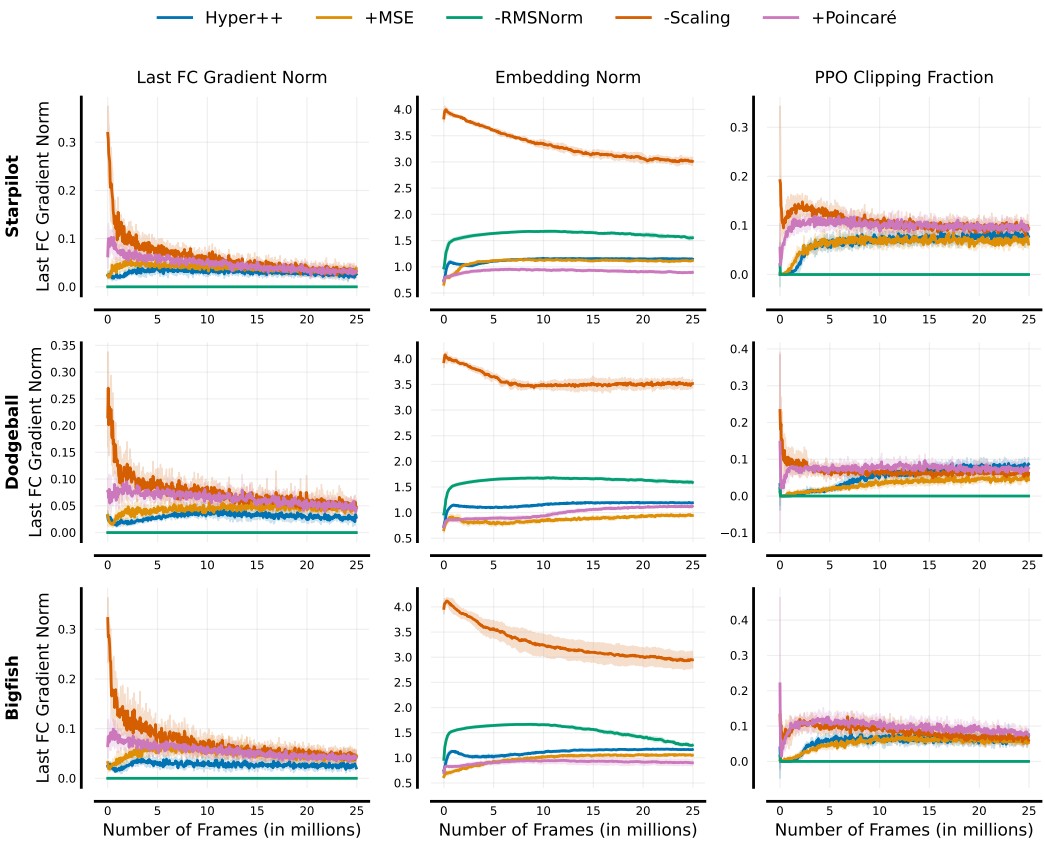

Figure 14: **Additional training metrics of hyperbolic deep RL agents.** Agents w/o RM-SNorm (Zhang & Sennrich, 2019) (−RMSNorm) suffer from growing embedding norms and vanishing gradients in the encoder. Using MSE instead of HL-Gauss (Imani & White, 2018) (+MSE) leads to larger initial encoder gradients due to gradients scaling proportional to the loss for MSE. Not using learned feature scaling (−Scaling) has the largest embedding norms and gradients, which are quickly compensated by RMSNorm's gradient variance normalization (Zhang & Sennrich, 2019).

## E.3 GRADIENT AND LOSS VARIANCE ANALYSIS

Figure 15 shows the evolution of the loss and the loss variance over the course of the training, averaged over all runs. Compared to MSE, the categorical loss is both higher and has higher variance. However, our paper argues that *not the loss values matter, but the gradients instead*. Table 11 shows the L1 and L2 gradient norms of the penultimate layer (last FC layer in the encoder) using the final 25% of training steps. Despite higher loss values and variance, using the HL-Gauss loss yields smaller, lower-variance gradients.

Table 11: **Penultimate layer layer gradient statistics.**

| Agent | L2 Grad Norm | L1 Grad Norm | N_runs |
|---|---|---|---|
| Euclidean | $0.0788 \pm 0.0276$ | $0.0506 \pm 0.0256$ | 96 |
| Euclidean+Categorical | $0.0695 \pm 0.0228$ | $0.0460 \pm 0.0240$ | 96 |
| Hyper++ | $0.0258 \pm 0.0099$ | $0.0294 \pm 0.0155$ | 96 |
| Hyper++-mse | $0.0327 \pm 0.0102$ | $0.0346 \pm 0.0134$ | 96 |

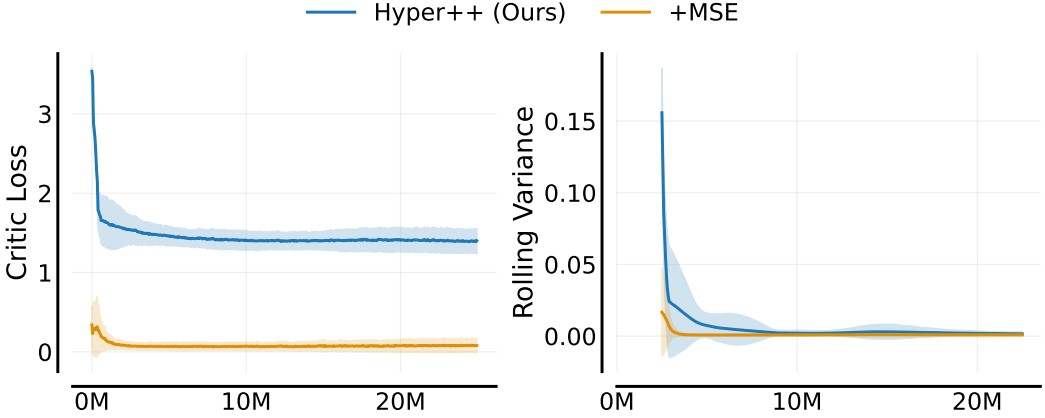

Figure 15: **Critic loss and variance.** We plot the critic loss and variance for our method when using MSE and the categorical loss, averaged over all runs and environments. The categorical HL-Gauss loss (Imani & White, 2018) has higher loss values and variance than MSE.

## E.4 LEARNING CURVES FOR ATARI GAMES

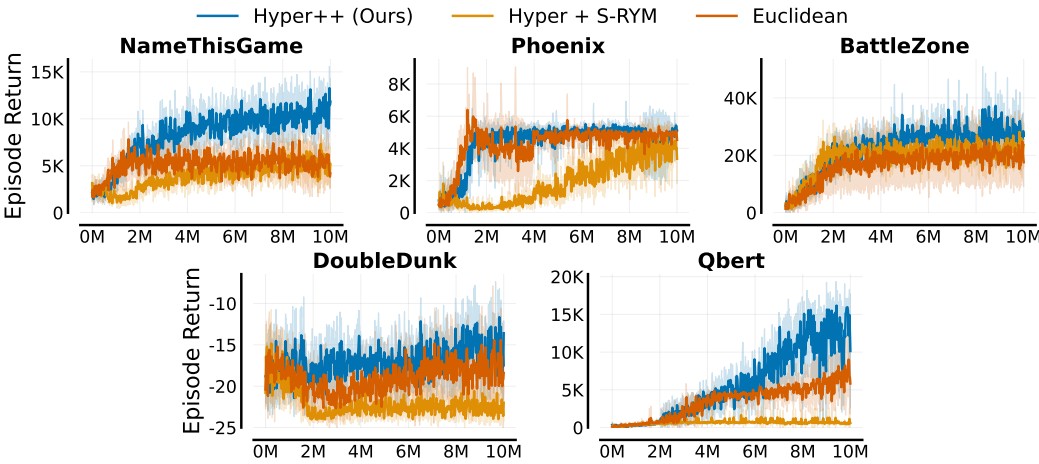

Figure 16: **Atari-5 learning curves.** HYPER++ outperforms the baselines on all environments with particularly strong gains in NAMETHISGAME and QBERT. RESULTS ARE AVERAGED OVER FIVE SEEDS.

## E.5 POLYAK AVERAGING EXPERIMENT

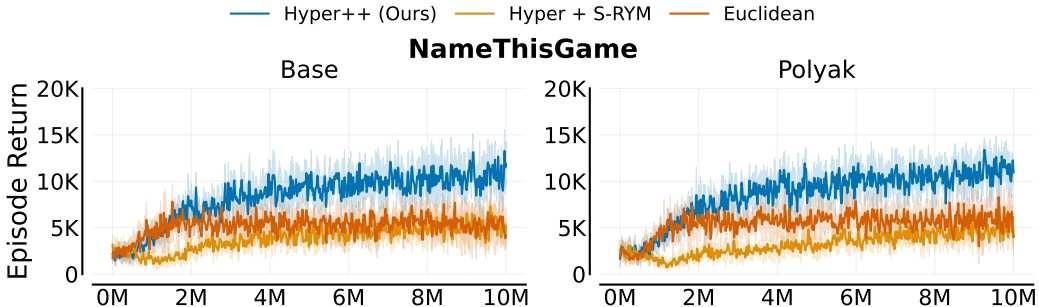

Figure 17: **Polyak averaging (NAMETHISGAME).** Polyak averaging refers to exponential moving average updates for the target network instead of hard replacement updates. Algorithm performance is not meaningfully affected by the form of the target network update. Runs are averaged over five seeds, with one standard deviation as error.

## E.6 OFF-BATCH PPO METRICS

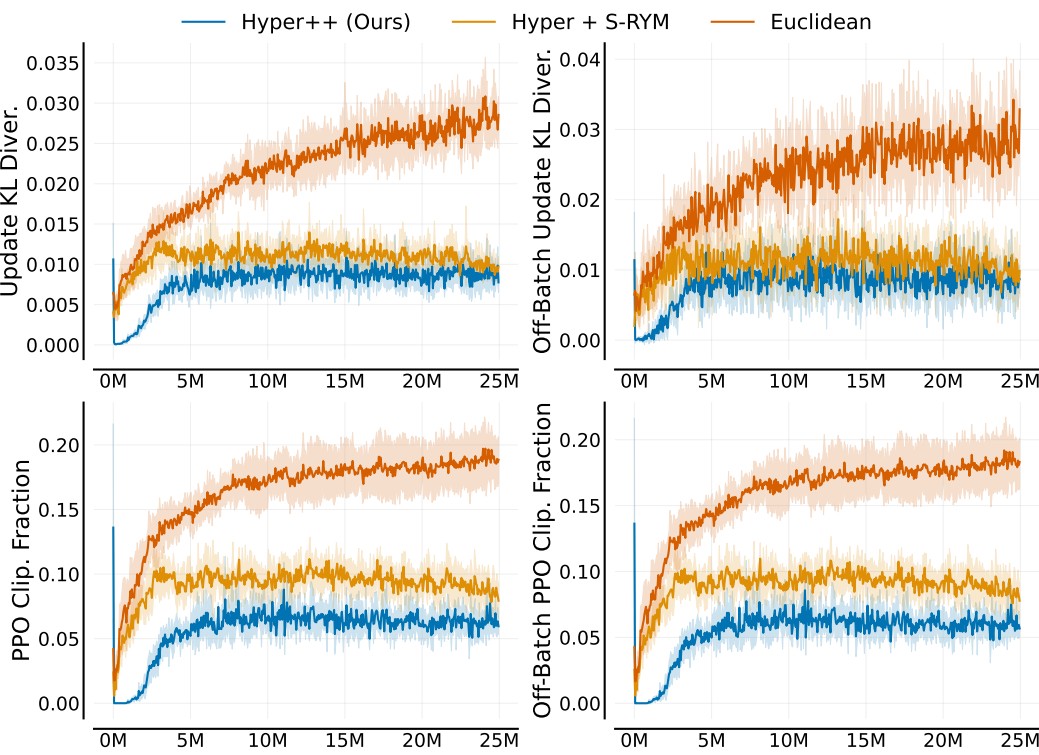

Figure 18: **Off-batch PPO stability metrics.** We track the update KL divergence and PPO clipping fraction for the batch that has currently been updated (left column) and for a batch of randomly sampled on-policy states (right column). The figures show a high level of similarity for the evolution of both metrics. For off-batch data, the update KL divergence has a noticeably higher variance.

## E.7 ARCHITECTURE ABLATIONS

Table 12: **Latent dimension and learned scaling $\alpha$ ablation studies.** Test results with PPO on ProcGen, averaged over six seeds.

| Environment | $d = 32$ | $d = 64$ | $d = 128$ | $d = 512$ | $d = 64, \alpha = 0.9$ | $d = 64, \alpha = 0.85$ |
|---|---|---|---|---|---|---|
| **bigfish** | $20.77 \pm 2.0$ | $20.01 \pm 4.8$ | $\mathbf{20.96 \pm 2.4}$ | $19.84 \pm 4.2$ | $19.55 \pm 2.9$ | $20.89 \pm 3.3$ |
| **starpilot** | $40.07 \pm 4.1$ | $\mathbf{42.49 \pm 2.6}$ | $41.49 \pm 2.0$ | $39.07 \pm 2.9$ | $39.55 \pm 4.1$ | $35.61 \pm 2.7$ |
| **dodgeball** | $9.02 \pm 1.0$ | $9.41 \pm 1.2$ | $9.15 \pm 1.9$ | $7.16 \pm 2.3$ | $\mathbf{9.62 \pm 1.4}$ | $9.07 \pm 1.5$ |
| **coinrun** | $8.60 \pm 0.4$ | $8.72 \pm 0.4$ | $8.68 \pm 0.4$ | $\mathbf{8.82 \pm 0.3}$ | $8.67 \pm 0.2$ | $8.67 \pm 0.4$ |
| **leaper** | $4.10 \pm 1.6$ | $\mathbf{5.13 \pm 1.0}$ | $4.27 \pm 1.4$ | $4.35 \pm 1.6$ | $4.75 \pm 1.3$ | $4.90 \pm 1.0$ |
| **ninja** | $5.07 \pm 0.4$ | $\mathbf{5.68 \pm 0.4}$ | $5.68 \pm 0.4$ | $5.40 \pm 0.3$ | $5.45 \pm 0.4$ | $5.68 \pm 0.9$ |
| **fruitbot** | $27.65 \pm 1.8$ | $28.28 \pm 0.9$ | $\mathbf{28.81 \pm 0.8}$ | $28.75 \pm 0.8$ | $27.87 \pm 0.5$ | $28.01 \pm 1.5$ |
| **jumper** | $5.15 \pm 0.4$ | $5.30 \pm 0.4$ | $5.22 \pm 0.6$ | $5.90 \pm 0.4$ | $5.32 \pm 0.7$ | $\mathbf{5.98 \pm 0.4}$ |
| **bossfight** | $9.23 \pm 1.2$ | $9.40 \pm 0.7$ | $8.94 \pm 1.0$ | $\mathbf{9.55 \pm 0.7}$ | $9.00 \pm 0.7$ | $9.43 \pm 0.9$ |
| **miner** | $\mathbf{7.76 \pm 0.6}$ | $7.21 \pm 0.7$ | $6.63 \pm 0.9$ | $6.46 \pm 0.7$ | $7.11 \pm 1.1$ | $7.44 \pm 0.4$ |
| **chaser** | $6.42 \pm 0.7$ | $7.27 \pm 0.9$ | $\mathbf{7.30 \pm 0.7}$ | $6.63 \pm 0.6$ | $6.74 \pm 0.9$ | $6.40 \pm 1.2$ |
| **climber** | $5.10 \pm 0.9$ | $5.86 \pm 1.2$ | $\mathbf{6.54 \pm 1.0}$ | $6.42 \pm 1.1$ | $6.20 \pm 0.9$ | $5.70 \pm 0.6$ |
| **caveflyer** | $\mathbf{4.33 \pm 0.9}$ | $4.12 \pm 0.4$ | $3.98 \pm 0.5$ | $4.08 \pm 0.5$ | $3.93 \pm 0.6$ | $3.96 \pm 0.6$ |
| **maze** | $5.68 \pm 0.4$ | $5.17 \pm 0.8$ | $\mathbf{5.70 \pm 1.2}$ | $5.57 \pm 0.4$ | $5.40 \pm 0.7$ | $5.48 \pm 0.7$ |
| **plunder** | $6.40 \pm 1.0$ | $5.82 \pm 1.6$ | $6.17 \pm 1.2$ | $5.85 \pm 0.7$ | $\mathbf{6.71 \pm 0.6}$ | $6.01 \pm 0.7$ |
| **heist** | $2.67 \pm 1.1$ | $2.28 \pm 1.0$ | $2.42 \pm 0.5$ | $2.60 \pm 0.6$ | $\mathbf{2.73 \pm 0.7}$ | $2.20 \pm 0.4$ |
| **IQM (normalized)** | $0.38$ | $0.41$ | $\mathbf{0.42}$ | $0.39$ | $0.40$ | $0.41$ |

### E.8 Phasic Policy Gradient

Tables 13 and 14 show ProcGen results using Phasic Policy Gradient (PPG) (Cobbe et al., 2021) averaged over six seeds. We use the default cleanRL hyperparameters for all runs (Huang et al., 2022). HYPER++ outperforms both baselines in terms of train and test performance. HYPER++ outperforms Hyper+S-RYM by 53% in terms of IQM test reward. Notably, Hyper+S-RYM fails to outperform the Euclidean baseline, **highlighting the generality of our method where previous hyperbolic agents fail**.

Table 13: **PPG ProcGen Train Results (mean $\pm$ std).**

|  | Euclidean | Hyper + S-RYM | Ours |
|---|---|---|---|
| **bigfish** | 26.20±4.9 | 20.84±4.4 | **27.42±4.0** |
| **starpilot** | 43.39±3.7 | 43.23±3.8 | **45.33±2.4** |
| **dodgeball** | 9.19±1.2 | 5.27±1.7 | **13.17±2.3** |
| **coinrun** | **9.87±0.1** | 9.28±0.3 | 9.80±0.2 |
| **leaper** | 5.45±2.7 | 3.30±2.8 | **6.82±2.1** |
| **ninja** | 9.07±0.4 | 5.73±0.8 | **9.13±0.3** |
| **fruitbot** | 30.58±1.6 | 31.55±0.8 | **31.63±0.7** |
| **jumper** | **8.70±0.4** | 8.32±0.3 | 8.62±0.4 |
| **bossfight** | 10.64±0.5 | 8.91±2.0 | **11.31±0.4** |
| **miner** | **9.15±0.9** | 7.68±0.6 | 6.77±1.8 |
| **chaser** | 6.94±3.1 | 7.94±0.5 | **8.86±0.5** |
| **climber** | **8.71±0.6** | 6.33±0.7 | 8.02±0.9 |
| **caveflyer** | **7.24±0.8** | 5.12±0.6 | 6.03±0.3 |
| **maze** | 9.07±0.5 | 6.67±0.5 | **9.08±0.5** |
| **plunder** | **13.78±1.2** | 10.27±3.2 | 13.67±0.5 |
| **heist** | **5.95±0.8** | 5.50±0.9 | 5.37±1.0 |
| **IQM (normalized)** | 0.67 | 0.47 | **0.69** |

Table 14: **PPG ProcGen Test Results (mean $\pm$ std).**

|  | Euclidean | Hyper + S-RYM | Ours |
|---|---|---|---|
| **bigfish** | 21.35±3.7 | 15.42±2.9 | **23.15±2.1** |
| **starpilot** | 40.06±2.9 | 41.84±4.7 | **43.70±3.4** |
| **dodgeball** | 5.56±0.9 | 3.14±0.9 | **11.63±2.2** |
| **coinrun** | 8.78±0.1 | 8.15±0.3 | **9.02±0.2** |
| **leaper** | 4.93±2.2 | 3.13±2.5 | **6.33±1.8** |
| **ninja** | 7.38±0.3 | 4.82±0.4 | **7.45±0.3** |
| **fruitbot** | 29.46±1.2 | 29.50±1.0 | **29.72±0.9** |
| **jumper** | **5.93±0.6** | 5.25±0.6 | 5.22±0.4 |
| **bossfight** | 10.20±0.6 | 8.10±2.4 | **10.89±0.6** |
| **miner** | **7.35±0.6** | 7.25±0.8 | 6.62±1.9 |
| **chaser** | 6.60±3.0 | 7.57±0.8 | **8.31±0.4** |
| **climber** | 6.57±0.5 | 5.30±0.9 | **6.88±0.8** |
| **caveflyer** | **5.70±0.6** | 4.50±1.1 | 5.43±0.9 |
| **maze** | 5.87±0.3 | 5.83±0.8 | **6.03±0.9** |
| **plunder** | 12.29±2.9 | 7.89±2.2 | **12.58±1.4** |
| **heist** | 3.38±0.8 | **3.73±0.6** | 3.55±0.7 |
| **IQM (normalized)** | 0.47 | 0.34 | **0.52** |

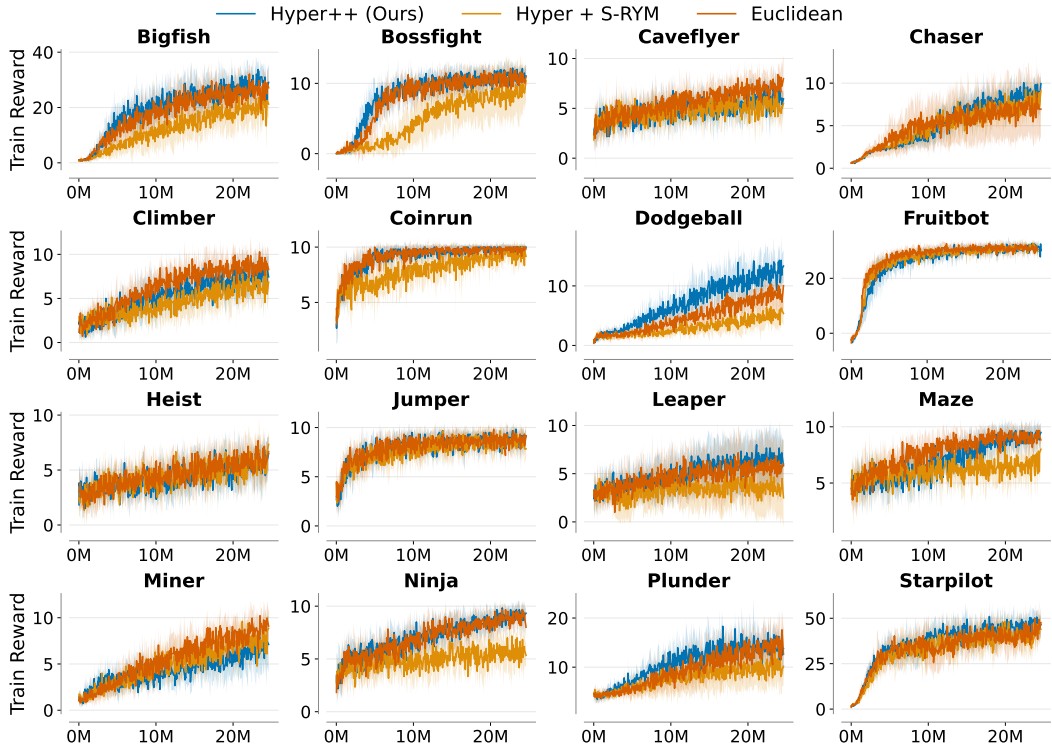

Figure 19: **PPG ProcGen Train Learning curves.**

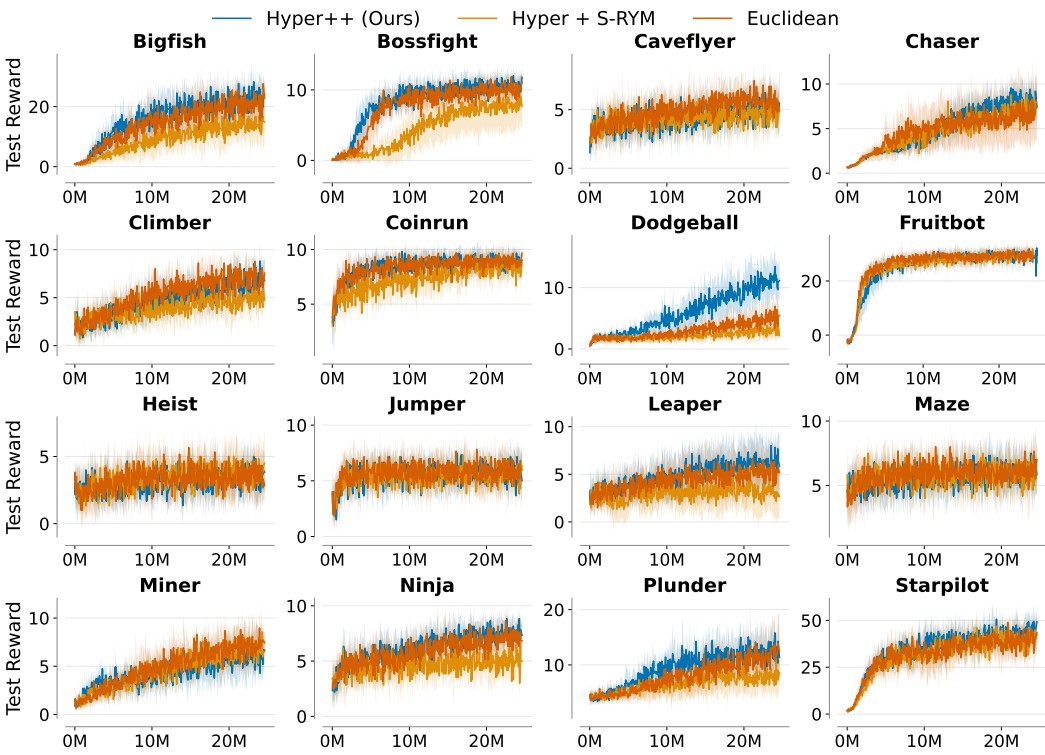

Figure 20: **PPG ProcGen Test Learning curves.**

