# OpenReview forum: "Understanding and Improving Hyperbolic Deep Reinforcement Learning"
_ICLR.cc/2026/Conference — ICLR 2026 Poster_

### Official Review · Reviewer_Pwnf · 2025-10-22

**Soundness:** 3
**Presentation:** 3
**Contribution:** 1
**Rating:** 4
**Confidence:** 4

**Summary:**

The paper proposes an extension of the Hyperbolic Deep RL (HyperRL) framework from [1]. In particular, the paper analyzes the instabilities in prior hyperRL methods and proposes several improvements to counteract them. These improvements come together in the  Hyper++ algorithm, which extends the implementation from [1] with a categorical value function, a feature-scaled RMSNorm layer, and switching from the Poincaré to the Hyperboloid model as the space for the latent features. Empirically, the proposed method is validated on the Procgen benchmark [2], and a subset of three environments from the Atari benchmark [3].


[1] Cetin, Edoardo, et al. "Hyperbolic deep reinforcement learning." arXiv preprint arXiv:2210.01542 (2022).

[2] Cobbe, Karl, et al. "Leveraging procedural generation to benchmark reinforcement learning." International conference on machine learning. PMLR, 2020.

[3] Bellemare, Marc G., et al. "The arcade learning environment: An evaluation platform for general agents." Journal of artificial intelligence research 47 (2013): 253-279.

**Strengths:**

- I found the overall exposition of the paper to be clear and intuitive. Clear text boxes are used to highlight the main methodology aspects, which are introduced naturally within the text.  While hyperbolic geometry for ML could be a challenging topic for some readers, I believe the paper does a good job at summarizing past results and providing analysis without overdoing the theoretical complexity.
- Each component of the proposed extensions seems logical and introduced from grounded considerations on the challenges of applying hyporbolic ML to reinforcement learning.
- The paper does a good job of introducing itself within the surrounding literature of RL and hyperbolic deep learning.

**Weaknesses:**

**Main:**
My main area of concern is the empirical validation of the paper. The proposed Hyper++ algorithm has only been properly examined on Procgen on top of vanilla PPO. The generality of the method is yet to be validated:
- Since its inception, several improvements upon PPO have been proposed for Procgen. It would have been nice to test if Hyper++ could also be applied to improve the performance of any more recent baselines from the last couple of years.
- The evaluation for DDQN seems to be constrained to only three handpicked environments from the whole suite. Not only is this too limited to provide evidence of generality, but it is also using a very outdated baseline in DDQN (9+ years old), which is far from state-of-the-art in sample efficiency.
Without empirical validation to potentially establish the methodology, I am unsure how much the proposed simple extension to the existing hyperRL framework would be relevant to the broader ICLR RL community.

**Other:**
1) The ablation study is only focused on three Procgen environments, rather than looking at average normalized performance across the Procgen suite, providing limited insights. Moreover, I think the ablation for RMSNorm should have added back the SpectralNorm introduced in the original HyperRL work.
2) The authors recollect their own numbers for the HyperRL baseline, which appear consistent but slightly inferior to the original results reported in this prior work. Yet, even with these baseline numbers, performance improvements of the proposed method on Procgen appear far from consistent (improving in only 8/16 and 11/16 train/test settings, respectively).
3) The paper lacks any analysis on what is the actual effect of each of their proposed addition and their effect on the learned latent representations, which is the critical main aspect of adding hyperbolic space to RL.
4) While it is good that some code was shared with the submission, it is currently far from allowing/facilitating reproducibility for the reviewers and the public. For instance, the README is empty without installation/execution instructions. I quickly did some checks trying to install the dependencies with uv and launching the ."sh" files, but was not able to get experiments started. I think for the code to be actually useful to reviewers, a simple README with instructions should have been provided at the time of submission.

**Questions:**

I have raised what I believe to be the main areas for improvement of the paper in the Weaknesses Section above. I would encourage the authors to address these aspects, and have no further questions at this stage.

---

> ### Author Response · Authors · 2025-11-21
> **Author's rebuttal (1)**
>
> Dear Pwnf,
>
> Thank you for your thoughtful and detailed feedback. We appreciate that you found the paper clear and intuitive in its exposition, recognized that each component is logically grounded in the challenges of hyperbolic RL, and noted our effective positioning within the surrounding literature. Your comments and remarks have been valuable in strengthening our evaluation. Below, we address each of your raised concerns. All changes based on your feedback are marked in **PURPLE** in the revised manuscript; changes according to multiple reviewers are in **GRAY**.
>
> **Concrete steps we took based on your feedback**:
>
> - Ran longer experiments on a larger subset of Atari games (Section 5.3, Figure 9, gray coloring)
> - Expanded our existing ablation studies to include all ProcGen environments (Section 5.2, Figure 7, Appendix E.8, gray coloring)
> - Added an ablation study with our method and SpectralNorm on the Euclidean subnetwork (Section 5.2, Figure 7, Appendix E.8)
> - Updated the README of our supplementary material with instructions to install and execute our code

---

> ### Author Response · Authors · 2025-11-21
> **Author's rebuttal (2)**
>
> ## Main W1: Validity of PPO
>
> > Since its inception, several improvements upon PPO have been proposed for Procgen. It would have been nice to test if Hyper++ could also be applied to improve the performance of any more recent baselines from the last couple of years.
>
> We appreciate the remark about validating the generality of Hyper++. However, we wanted to focus our analysis on validating gradient stabilization for hyperbolic representations while avoiding potentially confounding factors. The goal of our evaluation is to validate the effects of gradient stabilization across distinct Deep RL algorithms. We selected PPO and DDQN because they test Hyper++ across two fundamentally different RL paradigms, namely on-policy actor-critic (PPO) and off-policy value-based (DDQN). Both methods remain the canonical foundation for studying representation learning and training dynamics in recent top-tier literature.
>
> For example, [1] (NeurIps 2024) use PPO to diagnose representation collapse and plasticity loss, while [2] (NeurIps 2025 Oral) use PPO to establish a data attribution framework for on-policy RL. These works demonstrate that testing on standard PPO is the community standard for isolating the effects of algorithmic contributions. While improvements like Phasic Policy Gradient (PPG) [3] exist, they introduce orthogonal improvements that would confound our analysis. By strictly adhering to the standard PPO and DDQN formulations, we can isolate the specific contribution of hyperbolic geometry and Hyper++.
>
>
> ## Main W2: DDQN Atari experiments
>
> > The evaluation for DDQN seems to be constrained to only three handpicked environments from the whole suite. Not only is this too limited to provide evidence of generality, but it is also using a very outdated baseline in DDQN (9+ years old), which is far from state-of-the-art in sample efficiency. Without empirical validation to potentially establish the methodology, I am unsure how much the proposed simple extension to the existing hyperRL framework would be relevant to the broader ICLR RL community.
>
> Running experiments on the full Atari-57 benchmark (200M frames) is computationally infeasible for our lab. In fact, the computational demands for the full ALE benchmark are so high that it is almost exclusively used by Google DeepMind (“*Of the 17 algorithms listed on paperswithcode.com with full Atari-57 results, only one was from a research group outside Google or DeepMind.*”) [4]. Our initial three-game subset represents the most predictive subset for median human-normalized score [4], not a handpicked selection. We will explicitly mention this in the updated version of our paper.
>
> We have expanded our Atari evaluation to the Atari-5 benchmark for 10M steps/40M frames (a common reduction from 200M [5,6,7]). Results are below:
>
> | Game         |     Ours | Theirs | Euclidean |
> | ------------ | -------: | -----: | --------: |
> | Qbert        | **1.02** |   0.50 |      0.03 |
> | NameThisGame | **5.12** |   1.50 |      1.52 |
> | Phoenix      | **0.68** |   0.47 |      0.62 |
> | DoubleDunk   | **1.21** |  -1.93 |      0.12 |
> | BattleZone   | **0.74** |   0.58 |      0.52 |
> | **IQM**      | **1.07** |   0.32 |      0.45 |
>
> **On the relevance of DDQN as baseline**: Recent high-impact works continue to use DQN/DDQN as primary evaluation: [7 & 10] (ICML 2023 oral, 100+ citations each), [8] (NeurIPS 2023 spotlight, DDQN only), and [9] (ICML 2024 spotlight using only value-based methods for main Atari results, with PPO showing no gains). Our goal is to demonstrate that Hyper++ stabilizes gradient dynamics across algorithmic paradigms and architectures, not achieving SOTA Atari scores. SOTA methods, such as Agent-57 [11] and MuZero [12], lack official implementations and incorporate orthogonal components (model-based planning, custom exploration, LSTMs) that obscure method-specific contributions.
>
> Our evaluation demonstrates performance gains across on-policy (ProcGen) and off-policy (Atari) settings with different architectures (NatureCNN for DDQN, Impala-ResNet for PPO), establishing the generality of our gradient stabilization approach.

---

> ### Author Response · Authors · 2025-11-21
> **Author's rebuttal (3)**
>
> ## Other W1: Ablation completeness
>
> > The ablation study is only focused on three Procgen environments, rather than looking at average normalized performance across the Procgen suite, providing limited insights. Moreover, I think the ablation for RMSNorm should have added back the SpectralNorm introduced in the original HyperRL work.
>
> We conducted ablation studies on all 16 ProcGen environments, including a variant where we replaced RMSNorm with SpectralNorm across the entire network (following [13]). The table below shows interquartile mean (IQM) scores across the full suite. Our method achieves the highest aggregate performance (IQM = 0.40), outperforming all ablated variants, including the SpectralNorm variant (IQM = 0.00).
>
> The empirical results validate our theoretical analysis: SpectralNorm is insufficient to bound embedding norm growth (Lemma 4.1). Using SpectralNorm instead of RMSNorm leads to embeddings with approximately 50% larger norms. This leads to extremely small gradients in the Euclidean encoder, exactly as Section 3 predicted. We also observe that on some environments (e.g., heist and plunder), using an MSE loss outperforms the categorical loss. This is consistent with the Atari findings of [14, Figure C.1], who also find that MSE outperforms all categorical losses on some games (e.g., Phoenix or Alien). We hypothesize that these differences may be caused by as of now unknown properties of these specific environments.
>
> | Environment          | Ours                     | Ours+SpectralNorm (Full) | Ours+Poincaré           | Ours w/o Scaling | Ours+MSE Loss            | Ours w/o RMSNorm |
> | -------------------- | ------------------------ | ------------------------ | ----------------------- | ---------------- | ------------------------ | ---------------- |
> | bigfish              | $\mathbf{21.56 \pm 3.5}$ | $1.01 \pm 0.1$           | $12.52 \pm 3.8$         | $15.02 \pm 2.3$  | $16.47 \pm 4.6$          | $1.05 \pm 0.1$   |
> | dodgeball            | $\mathbf{8.91 \pm 0.7}$  | $1.54 \pm 0.2$           | $5.61 \pm 1.1$          | $5.28 \pm 1.0$   | $5.42 \pm 1.3$           | $1.59 \pm 0.3$   |
> | miner                | $\mathbf{7.42 \pm 1.2}$  | $1.48 \pm 0.3$           | $7.02 \pm 0.5$          | $6.44 \pm 1.4$   | $6.23 \pm 1.0$           | $1.62 \pm 0.4$   |
> | fruitbot             | $\mathbf{28.62 \pm 0.9}$ | $-1.74 \pm 0.3$          | $28.04 \pm 1.3$         | $27.76 \pm 1.3$  | $28.25 \pm 1.2$          | $-1.69 \pm 0.2$  |
> | ninja                | $\mathbf{5.57 \pm 0.6}$  | $3.37 \pm 0.6$           | $5.32 \pm 0.9$          | $4.70 \pm 0.5$   | $4.78 \pm 0.2$           | $3.28 \pm 0.5$   |
> | leaper               | $\mathbf{4.62 \pm 1.0}$  | $3.18 \pm 0.5$           | $3.58 \pm 0.9$          | $4.42 \pm 1.1$   | $2.47 \pm 0.2$           | $3.30 \pm 0.5$   |
> | bossfight            | $\mathbf{9.51 \pm 0.7}$  | $0.62 \pm 0.2$           | $8.60 \pm 1.0$          | $8.46 \pm 0.8$   | $9.50 \pm 0.7$           | $0.78 \pm 0.3$   |
> | coinrun              | $\mathbf{8.75 \pm 0.4}$  | $5.35 \pm 0.3$           | $8.58 \pm 0.4$          | $8.75 \pm 0.5$   | $7.75 \pm 0.7$           | $5.25 \pm 0.6$   |
> | starpilot            | $39.85 \pm 2.5$          | $2.80 \pm 0.3$           | $37.34 \pm 2.4$         | $39.66 \pm 3.5$  | $\mathbf{40.14 \pm 2.8}$ | $2.76 \pm 0.3$   |
> | chaser               | $6.75 \pm 0.8$           | $0.65 \pm 0.0$           | $\mathbf{7.08 \pm 1.2}$ | $6.94 \pm 0.6$   | $5.84 \pm 0.8$           | $0.65 \pm 0.0$   |
> | jumper               | $5.37 \pm 0.3$           | $2.40 \pm 0.4$           | $\mathbf{5.73 \pm 0.2}$ | $5.38 \pm 0.5$   | $5.52 \pm 0.4$           | $2.50 \pm 0.3$   |
> | maze                 | $5.58 \pm 1.5$           | $5.08 \pm 0.7$           | $5.40 \pm 0.7$          | $5.87 \pm 0.6$   | $\mathbf{6.18 \pm 0.8}$  | $5.22 \pm 0.4$   |
> | climber              | $4.95 \pm 1.1$           | $2.44 \pm 0.3$           | $5.14 \pm 1.2$          | $4.13 \pm 1.3$   | $\mathbf{5.79 \pm 0.3}$  | $2.47 \pm 0.4$   |
> | caveflyer            | $3.91 \pm 0.2$           | $3.57 \pm 0.5$           | $4.05 \pm 0.7$          | $4.85 \pm 0.7$   | $\mathbf{5.41 \pm 0.7}$  | $3.62 \pm 0.7$   |
> | heist                | $2.43 \pm 0.4$           | $2.95 \pm 0.4$           | $2.35 \pm 0.7$          | $2.00 \pm 0.6$   | $\mathbf{4.23 \pm 0.7}$  | $2.80 \pm 0.4$   |
> | plunder              | $5.08 \pm 0.7$           | $4.42 \pm 0.3$           | $6.88 \pm 1.6$          | $6.16 \pm 1.0$   | $\mathbf{7.59 \pm 1.3}$  | $4.44 \pm 0.3$   |
> | **IQM (normalized)** | $\mathbf{0.40}$          | $0.00$                   | $0.34$                  | $0.33$           | $0.33$                   | $0.00$           |

---

> ### Author Response · Authors · 2025-11-21
> **Author's rebuttal (4)**
>
> ## Other W2: Consistency of results
>
> > The authors recollect their own numbers for the HyperRL baseline, which appear consistent but slightly inferior to the original results reported in this prior work. Yet, even with these baseline numbers, performance improvements of the proposed method on Procgen appear far from consistent (improving in only 8/16 and 11/16 train/test settings, respectively).
>
> **Regarding performance consistency on ProcGen**: Figure 5 demonstrates statistically robust improvements across ProcGen. Hyper++ outperforms baselines in 11/16 test environments with substantial margins in several cases. When baselines remain competitive, margins are small. Uniform improvement across all ProcGen environments is not achieved even by [13], and similar patterns are standard in Atari benchmarks [7,8,9,10]. Hyper++ outperforms baselines in all Atari-5 games in addition to the ProcGen results.
>
> **Regarding HyperRL baseline numbers**: We discuss reproducibility issues in Appendix C.3. The primary obstacle is that the code of [13] is unseeded, making exact reproduction impossible. They also deviate from standard evaluation protocols (e.g., model not in eval mode). Their method uses SpectralNorm, which means that when not in eval mode, the approximation of the largest singular value is computed at test-time, leaking information into the training. The table below shows 5 runs on ProcGen BigFish with their code. The results do not match their reported performance, consistent with broader reproducibility issues in the field [15].
>
> |                          | Train           | Test            |
> | ------------------------ | --------------- | --------------- |
> | **Hyper+S-RYM (theirs)** | $14.52 \pm 1.3$ | $11.44 \pm 1.7$ |
> | **Hyper+S-RYM (ours)**   | $17.38 \pm 3.3$ | $11.88 \pm 2.7$ |
>
> Our evaluation protocol is more rigorous: we use more seeds, seed our runs, and utilize PyTorch's eval mode. This produces slightly conservative baseline numbers overall, making our improvements more robust, not less.
>
> ## Other W3: Analysis of latent representations
>
> > The paper lacks any analysis on what is the actual effect of each of their proposed addition and their effect on the learned latent representations, which is the critical main aspect of adding hyperbolic space to RL.
>
> This is a great question! As stated in our limitations and conclusion, our analysis adopts an optimization-centric view. We focus on the training dynamics of hyperbolic deep RL and the driving mechanisms by which it learns, rather than investigating specific hierarchical structures learned by the resulting latent representations. Analyzing the geometric properties would be highly interesting and a valuable direction for future research, but it exceeds the scope of this submission. Nevertheless, we can provide some insights into the latent space:
>
> - Section 4.1 theoretically shows that the proposed modifications prevent an unbounded growth of the embedding norms. Without RMSNorm, Proposition 4.2 does not hold, and large-norm embeddings can hinder gradient flow into the encoder. Figure 14 (Appendix E.3) empirically verifies this hypothesis.
> - We observed that agent performance did not consistently correlate with the level of "tree-likeness" as measured by $\delta$-hyperbolicity [16,17] in our experiments. This finding is consistent with recent research in graph learning, which has also raised questions about the use of this metric [18]. Therefore, we chose not to include this metric in our paper.
>
>
> ## Other W4: Code usability
>
> > While it is good that some code was shared with the submission, it is currently far from allowing/facilitating reproducibility for the reviewers and the public. For instance, the README is empty without installation/execution instructions. I quickly did some checks trying to install the dependencies with uv and launching the ."sh" files, but was not able to get experiments started. I think for the code to be actually useful to reviewers, a simple README with instructions should have been provided at the time of submission.
>
> We thank the reviewer for carefully checking the code and apologize for the omission of the README. We have now added a complete README to guide installation and usage, and after re-downloading the submitted code, we confirmed that all necessary configs and scripts are already provided to fully support reproducibility.

---

> ### Author Response · Authors · 2025-11-21
> **Author's rebuttal (5)**
>
> ## References
>
> [1] Moalla, Skander, et al. "No representation, no trust: connecting representation, collapse, and trust issues in ppo." Advances in Neural Information Processing Systems 37 (2024): 69652-69699.
>
> [2] Hu, Yuzheng, et al. "A Snapshot of Influence: A Local Data Attribution Framework for Online Reinforcement Learning." arXiv preprint arXiv:2505.19281 (2025).
>
> [3] Cobbe, Karl W., et al. "Phasic policy gradient." International Conference on Machine Learning. PMLR, 2021.
>
> [4] Aitchison, Matthew, Penny Sweetser, and Marcus Hutter. "Atari-5: Distilling the arcade learning environment down to five games." International Conference on Machine Learning. PMLR, 2023.
>
> [5] Obando Ceron, Johan, Marc Bellemare, and Pablo Samuel Castro. "Small batch deep reinforcement learning." Advances in Neural Information Processing Systems 36 (2023): 26003-26024.
>
> [6] Ceron, Johan Samir Obando, Aaron Courville, and Pablo Samuel Castro. "In value-based deep reinforcement learning, a pruned network is a good network." International Conference on Machine Learning. PMLR, 2024.
>
> [7] Sokar, Ghada, et al. "The dormant neuron phenomenon in deep reinforcement learning." International Conference on Machine Learning. PMLR, 2023.
>
> [8] Nikishin, Evgenii, et al. "Deep reinforcement learning with plasticity injection." Advances in Neural Information Processing Systems 36 (2023): 37142-37159.
>
> [9] Ceron, Johan Samir Obando, et al. "Mixtures of Experts Unlock Parameter Scaling for Deep RL." International Conference on Machine Learning. PMLR, 2024.
>
> [10] Lyle, Clare, et al. "Understanding plasticity in neural networks." International Conference on Machine Learning. PMLR, 2023.
>
> [11] Badia, Adrià Puigdomènech, et al. "Agent57: Outperforming the atari human benchmark." International conference on machine learning. PMLR, 2020.
>
> [12] Schrittwieser, Julian, et al. "Mastering atari, go, chess and shogi by planning with a learned model." Nature 588.7839 (2020): 604-609.
>
> [13] Cetin, Edoardo, et al. "Hyperbolic Deep Reinforcement Learning." The Eleventh International Conference on Learning Representations. 2023
>
> [14] Farebrother, Jesse, et al. "Stop Regressing: Training Value Functions via Classification for Scalable Deep RL." International Conference on Machine Learning. PMLR, 2024.
>
> [15] Agarwal, Rishabh, et al. "Deep reinforcement learning at the edge of the statistical precipice." Advances in neural information processing systems 34 (2021): 29304-29320.
>
> [16] Gromov, Mikhael. "Hyperbolic groups." In Essays in group theory, pp. 75-263. New York, NY: Springer New York, 1987.
>
> [17] Bonk, Mario, and Oded Schramm. "Embeddings of Gromov hyperbolic spaces." In Selected Works of Oded Schramm, pp. 243-284. New York, NY: Springer New York, 2011.
>
> [18] Katsman, Isay, and Anna Gilbert. "Shedding Light on Problems with Hyperbolic Graph Learning." Transactions on Machine Learning Research. 2025

---

> ### Author Response · Authors · 2025-11-26
>
> Dear Pwnf,
>
> Thank you for your thorough feedback. We have expanded our ProcGen ablations across the full benchmark, improved our Atari experiments, clarified why PPO and DDQN remain relevant baselines, and addressed your concerns regarding reproducibility. If any remaining issues prevent you from recommending acceptance, please let us know, and we will address them.

---

> ### Author Response · Authors · 2025-12-03
> **Results with a more recent baseline**
>
> During the discussion phase, we ran additional experiments with PPG, a more recent baseline than PPO. The table shows the test interquartile mean (IQM) on ProcGen using standard cleanRL [1] PPG hyperparameters:
>
>
> | Method         | IQM      |
> | -------------- | -------- |
> | Euclidean      | 0.47     |
> | Hyper+S-RYM    | 0.34     |
> | Hyper++ (Ours) | **0.49** |
>
> Compared to PPO, PPG introduces an auxiliary phase that combines value function training with a behavior cloning objective to prevent policy changes. Consistent with our PPO (Procgen) and DDQN (Atari) experiments, the results confirm that our theoretically justified method (Hyper++) outperforms previous work on hyperbolic RL (Hyper+S-RYM) by approximately 44%. These experiments clearly support the generality of Hyper++. Future work could investigate S-RYM's performance with PPG in more detail, but this is orthogonal to the main findings central to our proposed approach. Full per-environment results are in the revised manuscript (Tables 12 and 13).
>
> [1] Huang, Shengyi, et al. "Cleanrl: High-quality single-file implementations of deep reinforcement learning algorithms." JMLR, 2022.

---

### Official Review · Reviewer_sPEy · 2025-10-27

**Soundness:** 3
**Presentation:** 3
**Contribution:** 2
**Rating:** 4
**Confidence:** 3

**Summary:**

The paper proposes HYPER++, a modified hyperbolic PPO agent with: (i) a stable critic using categorical value loss, (ii) feature regularization to control embedding norms, and (iii) improved hyperbolic layer formulations. On ProcGen benchmarks, HYPER++ achieves more stable learning, better performance, and 30% faster training than previous hyperbolic RL methods.

**Strengths:**

- The paper is overall well-written and combines interesting ideas
- There is a quite strong theoretical section

**Weaknesses:**

None of the idea alone is truly novel and only the combination of several ideas. I therefore see the paper as largely empirical in scope. There is a 30% improvement on procgen as well as small improvements in 3 Atari games. However, the experimental setup is not fully convincing:
- relatively limited scope
- the interpretation of the improvements in practice are not fully clear

**Questions:**

- Since each component of HYPER++ (RMSNorm, scaling, categorical loss, hyperboloid model) are not fully novel, what new insight does their combination provide beyond improved stability? In other words, is there theoretical justification for why these specific components interact synergistically, or was the design largely empirical?
- Atari-3 and only 5M training steps is a very limited setup for the Atari benchmark. How do you know that it is representative of a larger set of environments (e.g. all ATARI games) and that the first 5M steps are representative of a longer training?
- How sensitive are the results to hyperparameter choices (e.g. NN architecture, etc.)
- The ablation results for HYPER++ in Table 1 are only reported on the 3 games with the highest improvements with the proposed architecture.

---

> ### Author Response · Authors · 2025-11-21
> **Author's rebuttal (1)**
>
> Dear sPEy,
>
> Thank you for your thoughtful and detailed feedback. We appreciate that you found the paper well-written and recognized the strength of our theoretical section. Your comments and remarks have been valuable in refining our experimental results. Below, we address each of your concerns directly. All changes based on your feedback are marked in **GREEN** in the revised manuscript; changes according to multiple reviewers are in **GRAY**.
>
> **Concrete steps we took based on your feedback**:
>
> - Added the synergies between our modifications to the first paragraph of Section 4 (green coloring)
> - Ran longer experiments on the Atari-5 benchmark (Section 5.3, Figure 9, gray coloring)
> - Expanded our existing ablation studies to include all ProcGen environments (Section 5.2, Figures 7 and 8, Appendix E.8, gray coloring)
> - Conducted additional ablation studies concerning the latent dimension and the $\alpha$ parameter of the learned scaling layer (Appendix E.9, green coloring)

---

> ### Author Response · Authors · 2025-11-21
> **Author's rebuttal (2)**
>
> ## Weaknesses: Scope, interpretation of improvements
>
> > relatively limited scope, the interpretation of the improvements in practice are not fully clear
>
> Our theoretical contribution establishes formal guarantees for gradient stabilization in hybrid hyperbolic RL architectures. Specifically, we prove that the commonly used SpectralNorm does not prevent unbounded norm and conformal factor growth in these architectures, thereby risking training instabilities. However, we then also prove that RMSNorm applied to the penultimate layer achieves dimension-independent bounds for activations with $f(0) = 0$ (covering ReLU, tanh, etc.), and that these bounds extend to both the Poincaré Ball and the Hyperboloid model. Through the gradient analysis in Section 3, the theoretical results on norm bounds directly connect to gradient dynamics. This provides the first rigorous characterization of training dynamics in Deep RL agents using hybrid hyperbolic networks.
>
> Our experimental design validates this theory across two fundamental and widely-used Deep RL paradigms: off-policy value-based (DDQN) and on-policy actor-critic (PPO). Furthermore, we use two established benchmarks: the full 16-game ProcGen suite and Atari-5, where games were selected via regression analysis to maximize representativeness of the 57-game benchmark [1]. We have substantially expanded our ablation study, now covering all ProcGen environments and more variants. These extended results confirm our core findings across the full experimental scope.
>
> Our theory directly predicts the empirical results. Figure 2 illustrates the effect of Hyper++. Our stabilization mechanism reduces gradient norms and variances (sub-figures g and h), thereby stabilizing core training dynamics and helping to maintain policy entropy while reducing PPO clipping. Our improvements on ProcGen, as shown in subfigure e, are a direct consequence of the improved training dynamics. Our ablation studies confirm the design choices derived from our theoretical analysis.
>
>
> ## Q1: Theoretical justification for component synergy
>
> > Since each component of HYPER++ (RMSNorm, scaling, categorical loss, hyperboloid model) are not fully novel, what new insight does their combination provide beyond improved stability? In other words, is there theoretical justification for why these specific components interact synergistically, or was the design largely empirical?
>
> The synergistic interactions between Hyper++'s components are based on a principled design derived from  Equation 3. Concretely, we identify the following terms as problematic:
>
> 1. The derivative of the MSE loss w.r.t. the prediction $\partial L / \partial v_{\mathbf{v}, r} (\mathbf{x}_H)$.
> 2. The Jacobian of the Hyperbolic layer w.r.t. its input $\partial v_{\mathbf{v}, r} (\mathbf{x}_{\mathrm H}) / \partial \mathbf{x}_H$.
> 3. The Jacobian of the exponential map $\partial \mathbf{x}_H / \partial \mathbf{x}_E$.
>
> We further analyze these terms in Equations 5 and 6, finding that gradients vanish or diverge as one moves away from the manifolds' origins in both models: The Poincaré Ball suffers from conformal factor divergence, while the Hyperboloid exponential map gradient suffers from exponential growth of the $\sinh(\cdot)$ and $\cosh(\cdot)$ terms in Equation 6. Each of the proposed components tackles a specific failure mode in Equation 3, and together they stabilize all gradient terms in Equation 3:
>
> 1. The categorical loss stabilizes the critic's learning signal by decoupling gradients from error magnitudes under non-stationary RL targets. This tackles instability in the first term of Equation 3.
> 2. The Hyperboloid removes conformal factor dependencies from the MLR layer gradient, tackling instability in the second term of Equation 3.
> 3. RMSNorm+Learnable scaling provably guarantees bounded gradients (Proposition 4.2, Corollary 4.3) without the drawback (loss of representational capacity) of manifold clipping [2] or SpectralNorm's global expressivity constraints. Crucially, learnable scaling prevents excessive loss of representational capacity in the latent space. This addresses instability in the third term of Equation 3.
>
> In summary, the synergy of our proposed modifications becomes clear when viewed through the lens of Equation 3. Together, our modifications stabilize training while preserving representational capacity. We have clarified this in the first paragraph of Section 4.

---

> ### Author Response · Authors · 2025-11-21
> **Author's rebuttal (3)**
>
> ## Q2: Atari experiments
>
> > Atari-3 and only 5M training steps is a very limited setup for the Atari benchmark. How do you know that it is representative of a larger set of environments (e.g. all ATARI games) and that the first 5M steps are representative of a longer training?
>
> For Atari, we expanded our experiments to Atari-5 at 10M steps (40M frames). These 5 games have the highest predictive power for median human-normalized score across the full Atari benchmark according to the Atari-5 paper [1] (regression $R^2 = 0.984$). Thus, these 5 games can be considered representative of the larger environment set. Running for less than 200M frames/50M steps is standard in recent works [3,4,5,6], and we follow the recommendations of [3,4] to run for 10M steps. Hyper++ improves on all 5 games, with substantial improvement over the Euclidean baseline and hyperbolic DDQN+S-RYM in IQM.
>
> | Game         |     Ours | Theirs | Euclidean |
> | ------------ | -------: | -----: | --------: |
> | Qbert        | **1.02** |   0.50 |      0.03 |
> | NameThisGame | **5.12** |   1.50 |      1.52 |
> | Phoenix      | **0.68** |   0.47 |      0.62 |
> | DoubleDunk   | **1.21** |  -1.93 |      0.12 |
> | BattleZone   | **0.74** |   0.58 |      0.52 |
> | **IQM**      | **1.07** |   0.32 |      0.45 |

---

> ### Author Response · Authors · 2025-11-21
> **Author's rebuttal (4)**
>
> ## Q3: Hyperparameter sensitivity and architectural choices
>
> > How sensitive are the results to hyperparameter choices (e.g. NN architecture, etc.)
>
> Please note that for DQN and PPO, we already use quite different network architectures: NatureCNN [7] and Impala-ResNet (large) [8]. NatureCNN uses three convolutional layers, followed by two linear layers. The Impala-ResNet (large) architecture is visualized in Figure 11 of our paper and consists of three blocks, each containing a convolutional layer, followed by two residual blocks.
>
> To understand the sensitivity of our approach to hyperparameter choices in even more detail, we conducted additional experiments. In particular, the table below shows results for our approach with different latent dimensions $d$ and different learned scaling factors $\alpha$. When not otherwise mentioned, $\alpha=0.95$. The table below shows the results, using the interquartile mean (IQM) as an aggregate metric. [9] report a test performance decrease when increasing the latent dimension from $d=32$ to $d=128$. In contrast, the performance of our method generally improves with intermediate latent sizes. However, on some games or when the latent size is too large, the agent overfits, resulting in decreased test performance. We find that using a lower $\alpha$ results in fluctuating performance for individual games, but may also slightly improve performance.
>
> | Environment          | d=32                    | d=64                     | d=128                    | d=512                   | d=64, $\alpha$=0.9             | d=64, $\alpha$=0.85            |
> | -------------------- | ----------------------- | ------------------------ | ------------------------ | ----------------------- | ----------------------- | ----------------------- |
> | starpilot            | $40.07 \pm 4.1$         | $\mathbf{42.49 \pm 2.6}$ | $41.49 \pm 2.0$          | $39.07 \pm 2.9$         | $39.55 \pm 4.1$         | $35.61 \pm 2.7$         |
> | leaper               | $4.10 \pm 1.6$          | $\mathbf{5.13 \pm 1.0}$  | $4.27 \pm 1.4$           | $4.35 \pm 1.6$          | $4.75 \pm 1.3$          | $4.90 \pm 1.0$          |
> | ninja                | $5.07 \pm 0.4$          | $\mathbf{5.68 \pm 0.4}$  | $5.68 \pm 0.4$           | $5.40 \pm 0.3$          | $5.45 \pm 0.4$          | $5.68 \pm 0.9$          |
> | chaser               | $6.42 \pm 0.7$          | $7.27 \pm 0.9$           | $\mathbf{7.30 \pm 0.7}$  | $6.63 \pm 0.6$          | $6.74 \pm 0.9$          | $6.40 \pm 1.2$          |
> | coinrun              | $8.60 \pm 0.4$          | $8.72 \pm 0.4$           | $8.68 \pm 0.4$           | $\mathbf{8.82 \pm 0.3}$ | $8.67 \pm 0.2$          | $8.67 \pm 0.4$          |
> | bossfight            | $9.23 \pm 1.2$          | $9.40 \pm 0.7$           | $8.94 \pm 1.0$           | $\mathbf{9.55 \pm 0.7}$ | $9.00 \pm 0.7$          | $9.43 \pm 0.9$          |
> | dodgeball            | $9.02 \pm 1.0$          | $9.41 \pm 1.2$           | $9.15 \pm 1.9$           | $7.16 \pm 2.3$          | $\mathbf{9.62 \pm 1.4}$ | $9.07 \pm 1.5$          |
> | caveflyer            | $\mathbf{4.33 \pm 0.9}$ | $4.12 \pm 0.4$           | $3.98 \pm 0.5$           | $4.08 \pm 0.5$          | $3.93 \pm 0.6$          | $3.96 \pm 0.6$          |
> | heist                | $2.67 \pm 1.1$          | $2.28 \pm 1.0$           | $2.42 \pm 0.5$           | $2.60 \pm 0.6$          | $\mathbf{2.73 \pm 0.7}$ | $2.20 \pm 0.4$          |
> | fruitbot             | $27.65 \pm 1.8$         | $28.28 \pm 0.9$          | $\mathbf{28.81 \pm 0.8}$ | $28.75 \pm 0.8$         | $27.87 \pm 0.5$         | $28.01 \pm 1.5$         |
> | maze                 | $5.68 \pm 0.4$          | $5.17 \pm 0.8$           | $\mathbf{5.70 \pm 1.2}$  | $5.57 \pm 0.4$          | $5.40 \pm 0.7$          | $5.48 \pm 0.7$          |
> | miner                | $\mathbf{7.76 \pm 0.6}$ | $7.21 \pm 0.7$           | $6.63 \pm 0.9$           | $6.46 \pm 0.7$          | $7.11 \pm 1.1$          | $7.44 \pm 0.4$          |
> | climber              | $5.10 \pm 0.9$          | $5.86 \pm 1.2$           | $\mathbf{6.54 \pm 1.0}$  | $6.42 \pm 1.1$          | $6.20 \pm 0.9$          | $5.70 \pm 0.6$          |
> | jumper               | $5.15 \pm 0.4$          | $5.30 \pm 0.4$           | $5.22 \pm 0.6$           | $5.90 \pm 0.4$          | $5.32 \pm 0.7$          | $\mathbf{5.98 \pm 0.4}$ |
> | plunder              | $6.40 \pm 1.0$          | $5.82 \pm 1.6$           | $6.17 \pm 1.2$           | $5.85 \pm 0.7$          | $\mathbf{6.71 \pm 0.6}$ | $6.01 \pm 0.7$          |
> | bigfish              | $20.77 \pm 2.0$         | $20.01 \pm 4.8$          | $\mathbf{20.96 \pm 2.4}$ | $19.84 \pm 4.2$         | $19.55 \pm 2.9$         | $20.89 \pm 3.3$         |
> | **IQM (normalized)** | $0.38$                  | $0.41$                   | $\mathbf{0.42}$          | $0.39$                  | $0.40$                  | $0.

---

> ### Author Response · Authors · 2025-11-21
> **Author's rebuttal (5)**
>
> ## Q4: Ablation breadth
>
> > The ablation results for HYPER++ in Table 1 are only reported on the 3 games with the highest improvements with the proposed architecture.
>
> We report ablations across all 16 ProcGen games in the table below using the interquartile mean (IQM) as an aggregate metric. Hyper++ (IQM=0.4) outperforms all ablated variants. Without RMSNorm, performance collapses (IQM=0.00), empirically verifying Proposition 4.2. Table 1 in the paper highlighted the three games where component contributions are most visible. We have updated our paper (Figures 7 and 8) accordingly and improved the discussion.
>
> | Environment          | Ours                     | Ours+Poincaré           | Ours w/o Scaling | Ours+MSE Loss            | Ours w/o RMSNorm |
> | -------------------- | ------------------------ | ----------------------- | ---------------- | ------------------------ | ---------------- |
> | bigfish              | $\mathbf{21.56 \pm 3.5}$ | $12.52 \pm 3.8$         | $15.02 \pm 2.3$  | $16.47 \pm 4.6$          | $1.05 \pm 0.1$   |
> | dodgeball            | $\mathbf{8.91 \pm 0.7}$  | $5.61 \pm 1.1$          | $5.28 \pm 1.0$   | $5.42 \pm 1.3$           | $1.59 \pm 0.3$   |
> | miner                | $\mathbf{7.42 \pm 1.2}$  | $7.02 \pm 0.5$          | $6.44 \pm 1.4$   | $6.23 \pm 1.0$           | $1.62 \pm 0.4$   |
> | fruitbot             | $\mathbf{28.62 \pm 0.9}$ | $28.04 \pm 1.3$         | $27.76 \pm 1.3$  | $28.25 \pm 1.2$          | $-1.69 \pm 0.2$  |
> | ninja                | $\mathbf{5.57 \pm 0.6}$  | $5.32 \pm 0.9$          | $4.70 \pm 0.5$   | $4.78 \pm 0.2$           | $3.28 \pm 0.5$   |
> | leaper               | $\mathbf{4.62 \pm 1.0}$  | $3.58 \pm 0.9$          | $4.42 \pm 1.1$   | $2.47 \pm 0.2$           | $3.30 \pm 0.5$   |
> | bossfight            | $\mathbf{9.51 \pm 0.7}$  | $8.60 \pm 1.0$          | $8.46 \pm 0.8$   | $9.50 \pm 0.7$           | $0.78 \pm 0.3$   |
> | coinrun              | $\mathbf{8.75 \pm 0.4}$  | $8.58 \pm 0.4$          | $8.75 \pm 0.5$   | $7.75 \pm 0.7$           | $5.25 \pm 0.6$   |
> | starpilot            | $39.85 \pm 2.5$          | $37.34 \pm 2.4$         | $39.66 \pm 3.5$  | $\mathbf{40.14 \pm 2.8}$ | $2.76 \pm 0.3$   |
> | chaser               | $6.75 \pm 0.8$           | $\mathbf{7.08 \pm 1.2}$ | $6.94 \pm 0.6$   | $5.84 \pm 0.8$           | $0.65 \pm 0.0$   |
> | jumper               | $5.37 \pm 0.3$           | $\mathbf{5.73 \pm 0.2}$ | $5.38 \pm 0.5$   | $5.52 \pm 0.4$           | $2.50 \pm 0.3$   |
> | maze                 | $5.58 \pm 1.5$           | $5.40 \pm 0.7$          | $5.87 \pm 0.6$   | $\mathbf{6.18 \pm 0.8}$  | $5.22 \pm 0.4$   |
> | climber              | $4.95 \pm 1.1$           | $5.14 \pm 1.2$          | $4.13 \pm 1.3$   | $\mathbf{5.79 \pm 0.3}$  | $2.47 \pm 0.4$   |
> | caveflyer            | $3.91 \pm 0.2$           | $4.05 \pm 0.7$          | $4.85 \pm 0.7$   | $\mathbf{5.41 \pm 0.7}$  | $3.62 \pm 0.7$   |
> | heist                | $2.43 \pm 0.4$           | $2.35 \pm 0.7$          | $2.00 \pm 0.6$   | $\mathbf{4.23 \pm 0.7}$  | $2.80 \pm 0.4$   |
> | plunder              | $5.08 \pm 0.7$           | $6.88 \pm 1.6$          | $6.16 \pm 1.0$   | $\mathbf{7.59 \pm 1.3}$  | $4.44 \pm 0.3$   |
> | **IQM (normalized)** | $\mathbf{0.40}$          | $0.34$                  | $0.33$           | $0.33$                   | $0.00$           |
>
> ## References
>
> [1] Aitchison, Matthew, Penny Sweetser, and Marcus Hutter. "Atari-5: Distilling the arcade learning environment down to five games." International Conference on Machine Learning. PMLR, 2023.
>
> [2] Guo, Yunhui, et al. "Clipped hyperbolic classifiers are super-hyperbolic classifiers." Proceedings of the IEEE/CVF Conference on Computer Vision and Pattern Recognition. 2022.
>
> [3] Obando Ceron, Johan, Marc Bellemare, and Pablo Samuel Castro. "Small batch deep reinforcement learning." Advances in Neural Information Processing Systems 36 (2023): 26003-26024.
>
> [4] Ceron, Johan Samir Obando, Aaron Courville, and Pablo Samuel Castro. "In value-based deep reinforcement learning, a pruned network is a good network." International Conference on Machine Learning. PMLR, 2024.
>
> [5] Sokar, Ghada, et al. "The dormant neuron phenomenon in deep reinforcement learning." International Conference on Machine Learning. PMLR, 2023.
>
> [6] Moalla, Skander, et al. "No representation, no trust: connecting representation, collapse, and trust issues in ppo." Advances in Neural Information Processing Systems 37 (2024): 69652-69699.
>
> [7] Mnih, Volodymyr, et al. "Human-level control through deep reinforcement learning." nature 518.7540 (2015): 529-533.
>
> [8] Espeholt, Lasse, et al. "Impala: Scalable distributed deep-rl with importance weighted actor-learner architectures." International conference on machine learning. PMLR, 2018.
>
> [9] Cetin, Edoardo, et al. "Hyperbolic Deep Reinforcement Learning." The Eleventh International Conference on Learning Representations. 2023

---

> ### Author Response · Authors · 2025-11-26
>
> Dear sPEy,
>
> Thank you for your detailed feedback. We have clarified the synergies between our components, improved our Atari experiments, and substantially expanded our ProcGen ablations to cover the full 16-game benchmark with varying latent sizes and scaling parameters.
>
> We believe these revisions, particularly the comprehensive ProcGen evaluation, address your concerns about the experimental setup. If any remaining issues prevent you from recommending acceptance, please let us know, and we will address them.

---

### Official Review · Reviewer_ZmLs · 2025-10-28

**Soundness:** 3
**Presentation:** 2
**Contribution:** 3
**Rating:** 4
**Confidence:** 2

**Summary:**

The paper aims to stabilize the learning process of PPO agents with hyperbolic representation. Authors provide empirical (Figure 2) and theoretical evidence (Section 3.2-3.4) that the instabilities arise in unstable gradients induced by nonstationarities in RL, large euclidean feature norms and exploding conformal factors in Poincare ball formulation of the hyperbolic space.

To this end, authors make three improvements to the naive Hyper method:
1. Categorical value function (HL-Gauss) to stabilize gradients
2. RMSNorm to prevent feature norm growth
3. Hyperboloid model (in replacement of Poincare model) to remove the conformal factor

This results in a superior performance in 4 Procgen and 3 Atari environments, compared to prior work (Hyper + S-RYM) and PPO with default Euclidean space.

**Strengths:**

- **Rich theoretical results**. Thorough analysis of the feature norms and gradients for hyperboloid MLR seems to be a contribution by itself. It also gives enough justification to the components such as RMSNorm and Hyperboloid model, which are also verified in their ablation studies.
- **Careful implementation details**. When replacing SpectralNorm with RMSNorm, authors also point out the exponential shrinkage of the hyperbolic space after normalization and proposes to rescale the features accordingly.

**Weaknesses:**

- **Lack of motivation for HL-Gauss**. Although all other components seem to be well-motivated, there are no sufficient justifications or observations that lead to the addition of categorical loss. Indeed, the benefits of categorical loss is not limited to hyperbolic DRL but for most DRL methods in general [1,2,3].
- **Limited evaluation setup**. The proposed method is verified in 4 ProcGen and 3 Atari environments, which is significantly smaller compared to the prior work by [4]. I suggest expanding the benchmark size, or at least incorporate environments where prior work (PPO + S-RYM) has degraded performance compared to PPO, such as `heist`, `caveflyer` and `ninja`.

Overall, I think the results are a bit weak to confidently accept this paper. However, I may be overlooking the theoretical value of this paper, as I do not have the sufficient theoretical backgrounds for it. If the authors are able to provide more empirical results, I will positively consider raising my score.

[1] FastTD3: Simple, Fast, and Capable Reinforcement Learning for Humanoid Control., Seo et al., ArXiv'25.

[2] Bigger, Regularized, Categorical: High-Capacity Value Functions are Efficient Multi-Task Learners., Nauman et al., ArXiv'25.

[3] Hyperspherical Normalization for Scalable Deep Reinforcement Learning., Lee et al., ICML'25.

[4] Hyperbolic Deep Reinforcement Learning., Cetin et al., ArXiv'22.

**Questions:**

- Have you tried C51-style categorical distribution instead of HL-Gauss, as a lot of works in RL tend to adapt the former [1,2,3] instead of HL-Gauss?
- Line 60-63: I think claiming that "PPO with a hybrid Euclidean–hyperbolic encoder is the prevalent architecture in deep RL" is an overstatement unless it is supported by a number of citations.

[1] FastTD3: Simple, Fast, and Capable Reinforcement Learning for Humanoid Control., Seo et al., ArXiv'25.

[2] Bigger, Regularized, Categorical: High-Capacity Value Functions are Efficient Multi-Task Learners., Nauman et al., ArXiv'25.

[3] Hyperspherical Normalization for Scalable Deep Reinforcement Learning., Lee et al., ICML'25.

---

> ### Author Response · Authors · 2025-11-21
> **Author's rebuttal (1)**
>
> Dear ZmLs,
>
> Thank you for your detailed and thoughtful feedback. We are pleased that you recognized the depth of our theoretical analysis, particularly regarding feature norms and gradients for hyperboloid multinomial logistic regression. We also appreciated your careful attention to implementation details, such as the exponential loss of capacity in latent space. We enjoyed your recognition of how our theoretical results directly justify and motivate our architectural choices. Below, we address each of your raised concerns. All changes based on your feedback are marked in **BLUE** in the revised manuscript; changes according to multiple reviewers are in **GRAY**.
>
> **Concrete steps we took based on your feedback**:
>
> - Adjusted our claim related to hybrid hyperbolic architectures in Deep RL (Section 1, blue coloring)
> - Improved our geometrical motivation for the categorical loss in the paper (Section 4, blue coloring)
> - Ran longer experiments on more Atari games (Section 5.3, Figure 9, gray coloring)
> - Conducted an ablation study with the C-51 value function (Section 5.2, Figure 7, Appendix E.8, gray coloring)

---

> ### Author Response · Authors · 2025-11-21
> **Author's rebuttal (2)**
>
> ## W1: Lack of motivation for HL-Gauss
>
> > Although all other components seem to be well-motivated, there are no sufficient justifications or observations that lead to the addition of categorical loss. Indeed, the benefits of categorical loss is not limited to hyperbolic DRL but for most DRL methods in general.
>
> Our use of categorical losses is motivated by how hyperbolic layers work. Unlike Euclidean networks that naturally output scalar values, hyperbolic multinomial logistic regression (MLR) layers produce classification scores, making a categorical value function the natural choice. While categorical losses benefit general Deep RL methods by handling nonstationarity and bounding gradients [1], their use in Hyper++ is motivated by a more fundamental issue: architectural alignment between the loss function and hyperbolic layer design.
>
> Concretely, in standard Euclidean Deep RL, the value function is approximated using an unbounded linear layer that directly outputs real-valued predictions. Directly optimizing these values with MSE is a natural choice. In contrast, hyperbolic networks differ fundamentally in their output structure. For linear layers, the outputs must remain on the hyperbolic manifold, which limits the range of feasible values and renders real-valued regression impractical. Instead, [2] use hyperbolic MLR layers (Section 2.2, Appendix B.3), which compute signed distances to margin hyperplanes in the hyperbolic space. These distance-based scores are inherently classification-oriented, as each output represents how far a point lies from a decision boundary. Applying MSE regression to these classification-native distance scores creates a mismatch: we would be using a regression objective on representations designed for classification. The categorical loss resolves this by framing value estimation as classification over discrete value bins, directly aligning with how hyperbolic MLR layers naturally operate. This creates a principled match between the geometric structure of hyperbolic representations and the optimization objective. Our ablation in Table 1 validates this design. We have updated our paper to clarify this motivation.
>
>
> ## W2: Limited evaluation setup
>
> > The proposed method is verified in 4 ProcGen and 3 Atari environments, which is significantly smaller compared to the prior work. I suggest expanding the benchmark size, or at least incorporate environments where prior work (PPO + S-RYM) has degraded performance compared to PPO, such as heist, caveflyer and ninja
>
> While we agree with the reviewer that additional experiments could further strengthen the empirical evaluation, we want to emphasize that our evaluation is in line with well-perceived current works. In particular, we evaluate on the full ProcGen benchmark (16 environments). Figure 5 aggregates all 16. Tables 7 and 8 (Appendix E.2) contain heist, caveflyer, and ninja. Figure 6 shows only four environments, which are identical to those in Figure 7 of [2].
>
> For Atari, we expanded our experiments to Atari-5 at 10M steps (40M frames). Running for less than 50M steps is consistent with recent works [3,4,5,6]. Atari-5 comprises the 5 games with the highest predictive power for median human-normalized score across the full benchmark (regression $R^2 = 0.984$) [7]. Hyper++ is better on all games while substantially improving IQM performance over the Euclidean baseline and hyperbolic DDQN+S-RYM.
>
> | Game         |     Ours | Theirs | Euclidean |
> | ------------ | -------: | -----: | --------: |
> | Qbert        | **1.02** |   0.50 |      0.03 |
> | NameThisGame | **5.12** |   1.50 |      1.52 |
> | Phoenix      | **0.68** |   0.47 |      0.62 |
> | DoubleDunk   | **1.21** |  -1.93 |      0.12 |
> | BattleZone   | **0.74** |   0.58 |      0.52 |
> | **IQM**      | **1.07** |   0.32 |      0.45 |

---

> ### Author Response · Authors · 2025-11-21
> **Author's rebuttal (3)**
>
> ## Q1: C51 instead of HL-Gauss
>
> > Have you tried C51-style categorical distribution instead of HL-Gauss, as a lot of works in RL tend to adapt the former instead of HL-Gauss?
>
> We attempted only HL-Gauss for the submission due to the superior empirical results presented in [1]. For the rebuttal, we conducted an ablation using the C51 categorical distribution shown below. The HL-Gauss value function outperforms the C51 value function substantially (IQM=0.40 vs. IQM=0.27).
>
> | Environment          | Ours                     | Ours + Poincaré         | Ours w/o Scaling | Ours + MSE Loss          | Ours w/o RMSNorm | Ours + C51      |
> | -------------------- | ------------------------ | ----------------------- | ---------------- | ------------------------ | ---------------- | --------------- |
> | bigfish              | $\mathbf{21.56 \pm 3.5}$ | $12.52 \pm 3.8$         | $15.02 \pm 2.3$  | $16.47 \pm 4.6$          | $1.05 \pm 0.1$   | $14.77 \pm 4.7$ |
> | dodgeball            | $\mathbf{8.91 \pm 0.7}$  | $5.61 \pm 1.1$          | $5.28 \pm 1.0$   | $5.42 \pm 1.3$           | $1.59 \pm 0.3$   | $5.70 \pm 1.3$  |
> | miner                | $\mathbf{7.42 \pm 1.2}$  | $7.02 \pm 0.5$          | $6.44 \pm 1.4$   | $6.23 \pm 1.0$           | $1.62 \pm 0.4$   | $4.40 \pm 1.4$  |
> | fruitbot             | $\mathbf{28.62 \pm 0.9}$ | $28.04 \pm 1.3$         | $27.76 \pm 1.3$  | $28.25 \pm 1.2$          | $-1.69 \pm 0.2$  | $27.21 \pm 1.3$ |
> | ninja                | $\mathbf{5.57 \pm 0.6}$  | $5.33 \pm 0.8$          | $4.70 \pm 0.5$   | $4.78 \pm 0.2$           | $3.28 \pm 0.5$   | $5.35 \pm 0.4$  |
> | leaper               | $\mathbf{4.62 \pm 1.0}$  | $3.58 \pm 0.9$          | $4.42 \pm 1.1$   | $2.47 \pm 0.2$           | $3.30 \pm 0.5$   | $4.20 \pm 1.4$  |
> | bossfight            | $\mathbf{9.51 \pm 0.7}$  | $8.60 \pm 1.0$          | $8.46 \pm 0.8$   | $9.50 \pm 0.7$           | $0.78 \pm 0.3$   | $8.68 \pm 0.9$  |
> | coinrun              | $\mathbf{8.75 \pm 0.4}$  | $8.58 \pm 0.4$          | $8.75 \pm 0.5$   | $7.75 \pm 0.7$           | $5.25 \pm 0.6$   | $8.67 \pm 0.3$  |
> | starpilot            | $39.85 \pm 2.5$          | $37.34 \pm 2.4$         | $39.66 \pm 3.5$  | $\mathbf{40.14 \pm 2.8}$ | $2.76 \pm 0.3$   | $33.68 \pm 4.7$ |
> | chaser               | $6.75 \pm 0.8$           | $\mathbf{7.08 \pm 1.2}$ | $6.94 \pm 0.6$   | $5.84 \pm 0.8$           | $0.65 \pm 0.0$   | $3.99 \pm 0.5$  |
> | jumper               | $5.37 \pm 0.3$           | $\mathbf{5.73 \pm 0.2}$ | $5.38 \pm 0.5$   | $5.52 \pm 0.4$           | $2.50 \pm 0.3$   | $5.23 \pm 0.6$  |
> | maze                 | $5.58 \pm 1.5$           | $5.40 \pm 0.7$          | $5.87 \pm 0.6$   | $\mathbf{6.18 \pm 0.8}$  | $5.22 \pm 0.4$   | $5.22 \pm 0.9$  |
> | climber              | $4.95 \pm 1.1$           | $5.14 \pm 1.2$          | $4.13 \pm 1.3$   | $\mathbf{5.79 \pm 0.3}$  | $2.47 \pm 0.4$   | $3.46 \pm 0.5$  |
> | caveflyer            | $3.91 \pm 0.2$           | $4.05 \pm 0.7$          | $4.85 \pm 0.7$   | $\mathbf{5.41 \pm 0.7}$  | $3.62 \pm 0.7$   | $3.87 \pm 0.3$  |
> | heist                | $2.43 \pm 0.4$           | $2.35 \pm 0.7$          | $2.00 \pm 0.6$   | $\mathbf{4.44 \pm 0.6}$  | $2.80 \pm 0.4$   | $2.55 \pm 0.9$  |
> | plunder              | $5.08 \pm 0.7$           | $6.88 \pm 1.6$          | $6.16 \pm 1.0$   | $\mathbf{7.59 \pm 1.3}$  | $4.44 \pm 0.3$   | $5.94 \pm 0.6$  |
> | **IQM (normalized)** | $\mathbf{0.40}$          | $0.34$                  | $0.33$           | $0.33$                   | $0.00$           | $0.27$          |
>
>
> ## Q2: Clarification of claim
>
> > Line 60-63: I think claiming that "PPO with a hybrid Euclidean–hyperbolic encoder is the prevalent architecture in deep RL" is an overstatement unless it is supported by a number of citations.
>
> To the best of our knowledge, [2] and [8] are the only studies that have implemented hyperbolic representations for online RL training to date, and both employ a hybrid architecture. Therefore, we made this claim.  We agree with the reviewer that it could be too strong, and have toned it down to “commonly used” in the updated version of our paper.

---

> ### Author Response · Authors · 2025-11-21
> **Author's rebuttal (4)**
>
> ## References
>
> [1] Farebrother, Jesse, et al. "Stop Regressing: Training Value Functions via Classification for Scalable Deep RL." International Conference on Machine Learning. PMLR, 2024.
>
> [2] Cetin, Edoardo, et al. "Hyperbolic Deep Reinforcement Learning." The Eleventh International Conference on Learning Representations. 2023
>
> [3] Obando Ceron, Johan, Marc Bellemare, and Pablo Samuel Castro. "Small batch deep reinforcement learning." Advances in Neural Information Processing Systems 36 (2023): 26003-26024.
>
> [4] Ceron, Johan Samir Obando, Aaron Courville, and Pablo Samuel Castro. "In value-based deep reinforcement learning, a pruned network is a good network." International Conference on Machine Learning. PMLR, 2024.
>
> [5] Sokar, Ghada, et al. "The dormant neuron phenomenon in deep reinforcement learning." International Conference on Machine Learning. PMLR, 2023.
>
> [6] Moalla, Skander, et al. "No representation, no trust: connecting representation, collapse, and trust issues in ppo." Advances in Neural Information Processing Systems 37 (2024): 69652-69699.
>
> [7] Aitchison, Matthew, Penny Sweetser, and Marcus Hutter. "Atari-5: Distilling the arcade learning environment down to five games." International Conference on Machine Learning. PMLR, 2023.
>
> [8] Salemohamed, Omar, et al. "Hyperbolic Deep Reinforcement Learning for Continuous Control." Tiny Papers@ ICLR. 2023.

---

> ### Author Response · Authors · 2025-11-26
>
> Dear Reviewer ZmLs,
>
> Thank you for your constructive feedback. We have added our geometric motivation for HL-Gauss, clarified and expanded our Atari experiment choices, and included a C51 ablation study as you suggested.
>
> We believe these additions address your concerns about motivation and empirical breadth. You mentioned you would positively consider raising your score if we provided more empirical results. We hope our expanded experiments meet this criterion. If you have remaining questions, please let us know.

---

### Official Review · Reviewer_Hm1V · 2025-10-31

**Soundness:** 3
**Presentation:** 3
**Contribution:** 3
**Rating:** 6
**Confidence:** 2

**Summary:**

This paper studies why deep RL agents with hyperbolic latent spaces are hard to optimize and proposes HYPER++, a PPO-based agent that stabilizes training and improves performance on ProcGen and a small Atari subset. The analysis traces instability to (i) exploding/vanishing gradients induced by the conformal factor and exponential map in the Poincaré ball, (ii) exponential growth in the Hyperboloid exponential Jacobian for large Euclidean feature norms, and (iii) non-stationary critics. The method has three ingredients: (1) replace value regression with a categorical value loss; (2) add RMSNorm at the last Euclidean layer plus a learned feature-scaling that caps norms in an optimization-friendly way; and (3) prefer the Hyperboloid model for the final layers. Empirically, HYPER++ improves median/IQM/mean normalized test rewards on ProcGen with better update-KL/clip-fraction metrics and ~30% wall-clock reductions versus prior hyperbolic agents, and transfers to DDQN on Atari-3.

**Strengths:**

**Clear diagnosis of instability.** The chain-rule decomposition (Eq. 3) and analysis of the conformal-factor gradient (Eq. 4) and HNN++ MLR derivative (Eq. 5) convincingly explain Poincaré instability; the Hyperboloid exponential Jacobian (Eq. 6) highlights a distinct failure mode—large Euclidean norms—justifying feature-norm control. RMSNorm placed only at the last Euclidean layer, plus a learnable radius cap, maintains capacity without per-layer SpectralNorm overhead; the bound in Prop. 4.2 ties directly to conformal-factor control.

**Well-motivated critic choice.** Switching to a categorical value loss is compatible with distributional RL’s stability theory and practice, and the ablation (+MSE) shows degradation on noisy-reward games.

**Comprehensive PPO metrics.** Reporting entropy, update-KL, clip-fraction, and gradient norms is great and aligns with best-practice PPO diagnostics.

**Empirical evidence.** Consistent ProcGen gains across aggregation metrics and competitive Atari-3 results demonstrate utility; reductions in wall-clock (~30%) are practically meaningful.

**Weaknesses:**

**Breadth of algorithms.** Off-policy validation is limited to DDQN. I think modern strong baselines (e.g., SAC, DrQ-v2) would better test generality. Ablations on Euclidean + categorical critic and Euclidean + RMSNorm controls would isolate which pieces help independent of hyperbolic geometry.

**Scope of environments.** ProcGen is appropriate, but analysis claims about hierarchy/tree-like structure would be stronger with goal-conditioned or hierarchical RL tasks (MiniGrid, Crafter, options) and long-horizon domains.

**Theory tightness and assumptions.** Some proofs rely on Lipschitz constants and dimension-independent bounds; clarity on tightness and behavior under non-1-Lipschitz activations would help. A formal link from stability bounds to PPO trust-region behavior (e.g., expected update-KL control) would strengthen the story but is not necessary.

**Questions:**

1. **Categorical critic details.** Which distributional parameterization is used (e.g., HL-Gauss vs. C51-style fixed supports)? How are supports chosen and are they shared across tasks? Could you report critic loss variance to corroborate stability claims?

2. **RMSNorm placement.** Did you try RMSNorm earlier in the encoder, or combining with SpectralNorm only on the last layer? A small table contrasting capacity vs. stability would help.

3. **Trust-region proxies.** Beyond update-KL and clip-fraction, do you observe changes in policy-entropy across states not in the batch (to quantify unseen-state interference)?

4. **Off-policy breadth.** Any results with SAC/DrQ (image-based), and do your findings about feature-norm control translate when using target networks and Polyak averaging?

5. **Comparisons to Euclidean + categorical.** To isolate hyperbolic benefits, could you add Euclidean + categorical critic and Euclidean + RMSNorm baselines?

---

> ### Author Response · Authors · 2025-11-21
> **Author's rebuttal (1)**
>
> Dear Hm1V,
>
> Thank you for your thorough and insightful feedback. We appreciate that you found our diagnosis of instability clear, our categorical value loss well-motivated, our PPO diagnostics comprehensive, and our empirical validation convincing. We have carefully considered your suggestions and believe they significantly strengthen our paper. Below, we address each of the points you raised. All changes based on your feedback are marked in **BROWN** in the revised manuscript; changes according to multiple reviewers are in **GRAY**.
>
> **Concrete steps we took based on your feedback**:
>
> - Ran an ablation with Polyak averaging for DDQN (Appendix E.6, brown coloring)
> - Ran an ablation where we replace RMSNorm with SpectralNorm in our method (Section 5.2, Figure 7, Appendix E.8, gray coloring)
> - Tracked trust-region statistics for data not from the current batch (Appendix E.7, brown coloring)
> - Added an analysis of loss variance and its effect on gradients (Appendix E.4, brown coloring)
> - Added ablation studies for Euclidean+RMSNorm and Euclidean+HL-Gauss (Section 5.2, Figure 8, Appendix E.8, gray coloring)

---

> ### Author Response · Authors · 2025-11-21
> **Author's rebuttal (2)**
>
> ## W1: Breadth of off-policy algorithms
>
> > Off-policy validation is limited to DDQN. I think modern strong baselines (e.g., SAC, DrQ-v2) would better test generality. Ablations on Euclidean + categorical critic and Euclidean + RMSNorm controls would isolate which pieces help independent of hyperbolic geometry.
>
> Our evaluation spans two fundamental RL paradigms: off-policy value-based (DDQN) and on-policy actor-critic (PPO). This cross-paradigm validation demonstrates architectural generalization more directly than testing algorithmic variants within discrete control (Rainbow, C51), which share the same value function architecture and therefore do not test different representational capacities. Hyperbolic continuous control (SAC, DrQ-v2) requires a fundamental technical extension: hyperbolic policy distributions and manifold-aware action spaces. Even preliminary work [1] uses hyperbolic features but reverts to Euclidean policies. Therefore, hyperbolic continuous control represents orthogonal future work, which falls beyond the scope of this rebuttal.
>
> Regarding Polyak averaging: we already use target networks in DDQN. New results with Polyak averaging (Appendix E.6, NameThisGame) show that our method is robust to the choice of the target network update. The table below shows the final scores.
>
> |                 | Replacement | Polyak   |
> | --------------- | ----------- | -------- |
> | **Ours**        | 11427.83    | 11164.88 |
> | **Euclidean**   | 5009.71     | 5651.24  |
> | **Hyper+S-RYM** | 4959.04     | 5164.92  |
>
>
> ## W2: Scope of environments
>
> > ProcGen is appropriate, but analysis claims about hierarchy/tree-like structure would be stronger with goal-conditioned or hierarchical RL tasks (MiniGrid, Crafter, options) and long-horizon domains.
>
> We focus on Atari and ProcGen because they are standard benchmarks for discrete control from images, allowing direct comparison with [2]. Hierarchical structure is ubiquitous in control tasks and does not require explicit goal-conditioning or option frameworks. For example, in ProcGen BigFish, the player eats progressively larger fish, creating a dynamic hierarchy where: (a) transitions become irreversible as the player grows, and (b) new actions become available (eating previously too large fish). This structure may partially explain the substantial gains observed by both [2] and our method in this environment. Similar temporal and state-based hierarchies exist across ProcGen (e.g., key collection before door opening, enemy defeat sequences) and Atari (e.g., level progression, power-up acquisition).
> MiniGrid and Crafter have sparse rewards, which introduces exploration efficiency as a confounding effect when evaluating representation quality. Our current results isolate the representational benefit of hyperbolic geometry in standard dense-reward settings where hierarchy still matters, but exploration does not confound the analysis.
>
>
> ## W3: Theory tightness and assumptions
>
> > Some proofs rely on Lipschitz constants and dimension-independent bounds; clarity on tightness and behavior under non-1-Lipschitz activations would help. A formal link from stability bounds to PPO trust-region behavior (e.g., expected update-KL control) would strengthen the story but is not necessary.
>
> Lemma 4.1 and Proposition 4.2 state tight worst-case upper bounds which hold for any Lipschitz continuous activation function $f$. If the activation function additionally satisfies $f(0)=0$ the bounds are fully dimension-independent and simplify to  $\lVert \hat{x} \rVert_2 < L$
> and $\lambda_{\text{exp}_{\bar{0}}(\hat{x})}<2\text{cosh}^2(\sqrt{c}L)$. This holds for $1$-Lipschitz activation functions (Tanh, ReLU) as well as non-$1$-Lipschitz ones (Leaky ReLU, ELU, Swish, GELU) [3]. For Lipschitz-activation functions that do not satisfy $f(0)=0$ (e.g. sigmoid, softplus, softmax) we get the original non-simplified bounds (equation 9), whose terms are dominated by the Lipschitz constant $L$ for sufficiently large $d$. The upper bound is attained if and only if $x$ is in the subspace spanned by the right-singular vectors corresponding to the largest singular value and $f(Wx+b)=c \cdot f(0)$ with $c\in[0,1]$ holds. Establishing a formal link between representation stability and the PPO trust region remains an open problem, even for Euclidean architectures, where only observations of empirical correlations exist [4].

---

> ### Author Response · Authors · 2025-11-21
> **Author's rebuttal (3)**
>
> ## Q1: Categorical critic details
>
> > Which distributional parameterization is used (e.g., HL-Gauss vs. C51-style fixed supports)? How are supports chosen and are they shared across tasks? Could you report critic loss variance to corroborate stability claims?
>
> We use HL-Gauss with default parameters specified in [5].
>
> Min/max reward: $\pm$10
> Number of bins: 51
> Smoothing ratio: 0.75
>
> The smoothing ratio defines how spread out the Gaussian distribution around the target is. A smoothing ratio of 0.75 corresponds to distributing probability mass to approximately 6 bins. We use these hyperparameters for all agents and environments.
>
> We added an analysis of the MSE and HL-Gauss variances in Appendix E.4 of our paper. The HL-Gauss loss produces larger loss values, and consequently, also a larger loss variance. However, the stability claims of our paper center on gradient dynamics, particularly those of the penultimate layer (Section 3). Distributional losses decouple gradient magnitude from error magnitude. Therefore, loss variance doesn't reflect optimization stability. HL-Gauss yields smaller L1 and L2 gradient norms with lower variance compared to MSE, measured on the penultimate encoder layer. This gradient behavior is more indicative of training stability than loss variance.
>
>
> ## Q2: RMSNorm placement
>
> > Did you try RMSNorm earlier in the encoder, or combining with SpectralNorm only on the last layer? A small table contrasting capacity vs. stability would help.
>
> Section 3 and Proposition 4.2 suggest that using RMSNorm on the penultimate layer is sufficient for stable training and aligns with existing works [6]. Applying RMSNorm to earlier convolutional layers does not yield performance gains [7], and we found no prior work applying RMSNorm to ResNet architectures. We conducted additional experiments with SpectralNorm (SN) on the penultimate layer for this rebuttal. The table below shows that training with SN fails, validating Lemma 4.1's theoretical prediction that SN alone is insufficient for stable learning.
>
> | Environment          | Ours                     | Ours+SpectralNorm (Penultimate) |
> | -------------------- | ------------------------ | ------------------------------- |
> | starpilot            | $\mathbf{39.85 \pm 2.5}$ | $2.58 \pm 0.3$                  |
> | fruitbot             | $\mathbf{28.62 \pm 0.9}$ | $-1.65 \pm 0.3$                 |
> | bigfish              | $\mathbf{21.56 \pm 3.5}$ | $0.96 \pm 0.1$                  |
> | bossfight            | $\mathbf{9.51 \pm 0.7}$  | $0.59 \pm 0.3$                  |
> | dodgeball            | $\mathbf{8.91 \pm 0.7}$  | $1.61 \pm 0.2$                  |
> | chaser               | $\mathbf{6.75 \pm 0.8}$  | $0.64 \pm 0.0$                  |
> | miner                | $\mathbf{7.42 \pm 1.2}$  | $1.33 \pm 0.2$                  |
> | coinrun              | $\mathbf{8.75 \pm 0.4}$  | $5.40 \pm 0.5$                  |
> | jumper               | $\mathbf{5.37 \pm 0.3}$  | $2.60 \pm 0.4$                  |
> | climber              | $\mathbf{4.95 \pm 1.1}$  | $2.39 \pm 0.8$                  |
> | ninja                | $\mathbf{5.57 \pm 0.6}$  | $3.40 \pm 0.2$                  |
> | leaper               | $\mathbf{4.62 \pm 1.0}$  | $3.27 \pm 0.3$                  |
> | plunder              | $\mathbf{5.08 \pm 0.7}$  | $4.45 \pm 0.3$                  |
> | maze                 | $\mathbf{5.58 \pm 1.5}$  | $5.27 \pm 0.5$                  |
> | caveflyer            | $3.91 \pm 0.2$           | $\mathbf{3.95 \pm 0.5}$         |
> | heist                | $2.43 \pm 0.4$           | $\mathbf{3.27 \pm 0.3}$         |
> | **IQM (normalized)** | $\mathbf{0.40}$          | $0.00$                          |
>
>
> ## Q3: Trust-region proxies
>
> > Beyond update-KL and clip-fraction, do you observe changes in policy-entropy across states not in the batch (to quantify unseen-state interference)?
>
> We track the update KL divergence on randomly sampled on-policy states not in the current batch (Appendix E.7), directly quantifying policy changes on unseen states. Off-batch KL tracks closely with on-batch KL, evolving similarly with moderately higher variance. The clipping fraction is virtually identical in both settings.
>
>
> ## Q4: Off-policy breadth
>
> > Any results with SAC/DrQ (image-based), and do your findings about feature-norm control translate when using target networks and Polyak averaging?
>
> Answered with Weakness 1.

---

> ### Author Response · Authors · 2025-11-21
> **Author's rebuttal (4)**
>
> ## Q5: Comparisons to Euclidean + categorical
>
> > To isolate hyperbolic benefits, could you add Euclidean + categorical critic and Euclidean + RMSNorm baselines?
>
> We added Euclidean+HL-Gauss and Euclidean+RMSNorm ablations (table below). HL-Gauss improves performance on 6/16 environments over the Euclidean baseline; RMSNorm improves 11/16. Hyper++ outperforms all Euclidean configurations in 12/16 environments. In environments where individual components help (e.g., BigFish, StarPilot, DodgeBall), Hyper++ substantially exceeds their individual contributions. This suggests that hyperbolic geometry is independently beneficial for training, but it requires gradient stabilization.
>
> | Environment          | Ours                     | Euclidean+RMSNorm       | Euclidean+HL-Gauss | Euclidean               |
> | -------------------- | ------------------------ | ----------------------- | ------------------ | ----------------------- |
> | bigfish              | $\mathbf{21.56 \pm 3.5}$ | $3.71 \pm 2.9$          | $7.60 \pm 3.6$     | $3.60 \pm 3.4$          |
> | starpilot            | $\mathbf{39.85 \pm 2.5}$ | $25.61 \pm 2.9$         | $32.12 \pm 7.1$    | $30.44 \pm 3.1$         |
> | dodgeball            | $\mathbf{8.91 \pm 0.7}$  | $2.74 \pm 0.9$          | $2.92 \pm 1.3$     | $1.79 \pm 0.3$          |
> | miner                | $\mathbf{7.42 \pm 1.2}$  | $5.25 \pm 2.0$          | $4.10 \pm 2.1$     | $7.15 \pm 0.6$          |
> | chaser               | $\mathbf{6.75 \pm 0.8}$  | $4.67 \pm 2.1$          | $4.24 \pm 1.1$     | $4.55 \pm 1.0$          |
> | leaper               | $\mathbf{4.62 \pm 1.0}$  | $3.37 \pm 1.1$          | $3.88 \pm 0.8$     | $3.58 \pm 1.3$          |
> | coinrun              | $\mathbf{8.75 \pm 0.4}$  | $7.63 \pm 1.2$          | $8.05 \pm 0.2$     | $6.13 \pm 1.2$          |
> | fruitbot             | $\mathbf{28.62 \pm 0.9}$ | $26.82 \pm 1.3$         | $28.17 \pm 0.9$    | $27.88 \pm 0.7$         |
> | ninja                | $\mathbf{5.57 \pm 0.6}$  | $5.13 \pm 0.4$          | $3.20 \pm 1.1$     | $4.65 \pm 0.5$          |
> | bossfight            | $\mathbf{9.51 \pm 0.7}$  | $9.19 \pm 0.8$          | $6.10 \pm 1.3$     | $9.50 \pm 0.8$          |
> | plunder              | $\mathbf{5.08 \pm 0.7}$  | $4.83 \pm 0.5$          | $4.33 \pm 1.1$     | $4.71 \pm 0.5$          |
> | jumper               | $\mathbf{5.37 \pm 0.3}$  | $5.37 \pm 0.1$          | $5.03 \pm 0.6$     | $5.08 \pm 0.5$          |
> | caveflyer            | $3.91 \pm 0.2$           | $4.45 \pm 0.5$          | $3.99 \pm 0.5$     | $\mathbf{5.22 \pm 0.3}$ |
> | maze                 | $5.58 \pm 1.5$           | $\mathbf{6.92 \pm 0.3}$ | $5.35 \pm 1.0$     | $6.43 \pm 0.5$          |
> | climber              | $4.95 \pm 1.1$           | $\mathbf{6.43 \pm 0.4}$ | $4.23 \pm 1.3$     | $5.89 \pm 0.7$          |
> | heist                | $2.43 \pm 0.4$           | $\mathbf{3.92 \pm 0.9}$ | $2.28 \pm 0.6$     | $3.68 \pm 0.5$          |
> | **IQM (normalized)** | $\mathbf{0.40}$          | $0.28$                  | $0.20$             | $0.26$                  |
>
>
> ## References
>
> [1] Salemohamed, Omar, et al. "Hyperbolic Deep Reinforcement Learning for Continuous Control." Tiny Papers@ ICLR. 2023.
>
> [2] Cetin, Edoardo, et al. "Hyperbolic Deep Reinforcement Learning." The Eleventh International Conference on Learning Representations. 2023
>
> [3] https://openreview.net/pdf?id=pRZ0RKl11f
>
> [4] Moalla, Skander, et al. "No representation, no trust: connecting representation, collapse, and trust issues in ppo." Advances in Neural Information Processing Systems 37 (2024): 69652-69699.
>
> [5] Farebrother, Jesse, et al. "Stop Regressing: Training Value Functions via Classification for Scalable Deep RL." International Conference on Machine Learning. PMLR, 2024.
>
> [6] Gallici, Matteo, et al. "Simplifying Deep Temporal Difference Learning." The Thirteenth International Conference on Learning Representations. 2025
>
> [7] Zhang, Biao, and Rico Sennrich. "Root mean square layer normalization." Advances in neural information processing systems 32 (2019).

---

> ### Author Response · Authors · 2025-11-26
>
> Dear Reviewer Hm1V,
>
> Thank you for your thorough review. We have expanded our ablations on ProcGen and Atari, added loss/gradient variance experiments, clarified our theoretical results, and answered your questions about categorical critic details, RMSNorm placement, and off-policy breadth.
>
> We believe our revisions address your concerns. If you have any remaining questions or would like further clarification on any aspect of our responses, please let us know.

---

### Author Response · Authors · 2025-12-03
**General response**

Dear AC and Reviewers,
We would like to thank you, and especially the new AC, for your time and effort during this particularly stressful review process. We are glad to see that the reviewers recognize our core contributions:

1. **Strong theoretical results** [ZmLs, sPEy], specifically our chain-rule analysis of the penultimate layer gradients, our insights into Poincaré MLR conformal factor instability, and the impact of feature norms on exponential map gradients.
2. A **carefully-designed approach** [Hm1V, sPEy, Pwnf] that is grounded in the challenges of hyperbolic RL. The paper was described as well-written, with ideas that are effectively presented.

Below, we summarize the reviewers' main concerns and our responses.

## Evaluation Scope [Hm1V, ZmLs, sPEy, Pwnf]

We addressed concerns regarding benchmark breadth:

- **ProcGen**: We clarified that our main results already cover the full 16-game benchmark, not a subset (Tables 7-8).
- **Atari**: We expanded our experiments to Atari-5 (40M frames [1,2]). [3] selected Atari-5 by evaluating all possible 5-game subsets and choosing the one that best represents the full benchmark (regression $R^2 =0.984$). Hyper++ (1.07 IQM) strongly outperforms Euclidean (0.45) and prior hyperbolic methods (0.32).

Our revised experiments comprehensively address the concerns about the scope of our existing evaluation [ZmLs, sPEy, Pwnf]. The results confirm that Hyper++ outperforms baselines for both on-policy (PPO, PPG) and off-policy (DDQN) algorithms across two diverse discrete control benchmarks. We note that extending our work to hyperbolic continuous control (SAC/DrQ-v2) [Hm1V] would require fundamental theoretical extensions (bounded hyperbolic policy distributions) and is orthogonal to this work.

## Theoretical Grounding & Component Synergy [Hm1V, sPEy]

Component synergy is derived directly from the penultimate layer Jacobian (Equation 3) [sPEy]. Each component targets a specific computationally or statistically problematic part of the gradient derived via the chain rule:

- **Categorical loss (HL-Gauss)**: Stabilizes the prediction gradients.
- **Hyperboloid parameterization**: Eliminates conformal factor divergence in the hyperbolic MLR Jacobian.
- **RMSNorm + scaling**: Ensures well-behaved exponential map Jacobians.

**Proposition 4.2** formalizes the impact of RMSNorm on the exponential map and conformal factor for arbitrary Lipschitz-continuous activation functions [Hm1V]. Our clarifications and manuscript revisions address all reviewer comments related to our theory.

## Ablation Studies [Hm1V, sPEy, Pwnf, ZmLs]

Following the reviewers' suggestions, we expanded our ablations to isolate the benefits of hyperbolic representations and strengthen our claims. We added the following experiments:

- **Existing ablations**: The originally presented ablations now cover all ProcGen environments [sPEy, Pwnf].
- **Euclidean results**: Hyper++ outperforms strong Euclidean baselines (Euclidean+RMSNorm, Euclidean+HL-Gauss, Euclidean+Ours), demonstrating the benefits of hyperbolic geometry [Hm1V].
- **SpectralNorm**: SpectralNorm ablations (penultimate layer, Euclidean encoder) fail to learn [Hm1V, Pwnf].
- **Robustness**: Ablations over latent dimension and scaling $\alpha$ show Hyper++'s hyperparameter robustness [sPEy].
- **C-51**: Hyper++ with HL-Gauss outperforms Hyper++ with C-51 [ZmLs].

*To summarize: we added all the additional ablations requested by the reviewers.*

## RL Algorithms  & Reproducibility [Pwnf]

Reviewer Pwnf raised concerns about our used algorithms. PPO and DDQN are standard baselines for research on RL training dynamics, as established in recent literature [4,5,6,7]. To further validate generality, we conducted additional PPG experiments during the discussion phase. The PPG experiments confirm that Hyper++ outperforms existing hyperbolic RL agents.

Crucially, all baselines are evaluated with fixed seeds and use of `eval` mode to ensure reproducibility. Our code includes a comprehensive `README` for full reproducibility.

## References

[1] Ceron, Johan Samir Obando, et al. "In value-based deep reinforcement learning, a pruned network is a good network." ICML, 2024.

[2] Sokar, Ghada, et al. "The dormant neuron phenomenon in deep reinforcement learning." ICML, 2023.

[3] Aitchison, Matthew, et al. "Atari-5: Distilling the arcade learning environment down to five games." ICML, 2023.

[4] Moalla, Skander, et al. "No representation, no trust: connecting representation, collapse, and trust issues in ppo." NeurIPS, 2024.

[5] Hu, Yuzheng, et al. "A Snapshot of Influence: A Local Data Attribution Framework for Online Reinforcement Learning." NeurIPS, 2025.

[6] Sokar, Ghada, et al. "The dormant neuron phenomenon in deep reinforcement learning." ICML, 2023.

[7] Nikishin, Evgenii, et al. "Deep reinforcement learning with plasticity injection." NeurIPS, 2023.

[8] Cetin, Edoardo, et al. "Hyperbolic Deep Reinforcement Learning." ICLR, 2023.

---

### Meta-Review · Area_Chair_JYSV · 2026-01-07

**Summary:**

This paper tackles some of the optimization issues arising from hyperbolic feature spaces for RL and proposes Hyper++ to address these. Specifically, they propose using categorical value losses, feature norm regularization, and optimization-friendly hyperbolic layers. The authors provide both theoretical and empirical justification for their proposal.

Overall I found the paper and author responses convincing, and as such I am suggesting accepting this work.

**Reviewer Concerns:**

I highlight the main concerns raised by reviewers.

## Hm1V
- W1 (Breadth of algorithms). The authors added extra experiments with PPG, which also supports the effectiveness of Hyper++
- W2 (Scope of environments). Here the authors provide a somewhat hand-wavy response for why ProcGen is adequate enough for their purposes. I would have liked to see more experiments on MiniGrid, for example, which I believe does share similar characteristics. This, however, is not enough of a criticism to prevent me from recommending acceptance. The authors did provide extra experiments on Atari (as requested by other reviewers as well).
- W3 (Theory tightness and assumptions). This was adequately addressed by the authors.
- Questions. The reviewer posed a number of questions that would help clarify the understanding of Hyper++. The authors did a great job addressing these suggestions by running the experiments and adding the results to the revision.

## ZmLs
- W1 (Lack of motivation for HL-Gauss). This was very well addressed by the authors, as they point out that, aside from them being shown to lead to generally more performant agents, "their use in Hyper++ is motivated by a more fundamental issue: architectural alignment between the loss function and hyperbolic layer design."
- W2 (Limited evaluation setup). The authors ran on extra Atari environments and presented results on all ProcGen envs, so this is adequately addressed.
- Q1 (C51 instead of HL-Gauss). The authors ran the suggested experiments and provided the results, which are consistent with their claims. This concern was well addressed.

## sPEy
- W1 (relatively limited scope and the interpretation of the improvements in practice are not fully clear). I don't think these were very reasonable concerns from the reviewer, but the authors addressed them very well.
- Q1 (Theoretical justification for component synergy). This concern was excellently addressed by the authors, as they provided clear theoretical justification for each of the components in Hyper++.
- Q2 (Atari experiments). The authors addressed this well, providing extra experiments on Atari-5.
- Q3 and Q4 (How sensitive are the results to hyperparameter choices? and Ablation Breadth). The authors expanded their ablation studies to include all ProcGen environments, which address these questions well.

## Pwnf
- W1 (Validity of PPO). The authors ran extra experiments with PPG, which I feel adequately addresses the reviewer concern.
- W2 (The evaluation for DDQN seems to be constrained to only three handpicked environments from the whole suite.). While running on the whole suite is computationally prohibitive, the authors did expand to the full Atari-5 suite, which I feel adequately addresses the reviewer's concern.
- W3 (ablation study is only focused on three Procgen environments). The authors extended ablation studies to all ProcGen environments.
- W4 (Consistency of results). The authors provide a good response to this concern, even raising the issue of reproducibility with the original HyperRL paper.
- W5 (Analysis of latent representations). The authors provide a reasonable response to this concern. Although not fully satisficing the reviewer's request, I do not believe it is enough to reduce the acceptance recommendation.

**Reviewer Scores:**

- **Hm1V:** Currently at 6, likely to stay supportive of acceptance but not sure if they would have increased to 8.
- **ZmLs:** Currently at 4, but likely to increase to acceptance given that all concerns were addressed.
- **sPEy:** Currently at 4, but likely to increase to acceptance given that all concerns were addressed.
- **Pwnf:** Currently at 4, but likely to increase to acceptance given that all concerns were addressed.

---

### Decision · Program_Chairs · 2026-01-26

Accept (Poster)